# Semantically Aligned Task Decomposition in Multi-Agent Reinforcement Learning

## Abstract

The difficulty of appropriately assigning credit is particularly heightened in cooperative MARL with sparse reward, due to the concurrent time and structural scales involved. Automatic subgoal generation (ASG) has recently emerged as a viable MARL approach inspired by utilizing subgoals in intrinsically motivated reinforcement learning. However, end-to-end learning of complex task planning from sparse rewards without prior knowledge, undoubtedly requires massive training samples. Moreover, the diversity-promoting nature of existing ASG methods can lead to the "over-representation" of subgoals, generating numerous spurious subgoals of limited relevance to the actual task reward and thus decreasing the sample efficiency of the algorithm. To address this problem and inspired by the disentangled representation learning, we propose a novel "disentangled" decision-making method, **S**emantically **A**ligned task decomposition in **MA**RL (**SAMA**), that prompts pretrained language models with chain-of-thought that can suggest potential goals, provide suitable goal decomposition and subgoal allocation as well as self-reflection-based replanning. Additionally, SAMA incorporates language-grounded MARL to train each agent's subgoal-conditioned policy. SAMA demonstrates considerable advantages in sample efficiency compared to state-of-the-art ASG methods, as evidenced by its performance on two challenging sparse-reward tasks, `Overcooked` and `MiniRTS`.

## 1 Introduction

The challenge of *credit assignment* is notably exacerbated in cooperative multi-agent reinforcement learning (MARL) with sparse-reward compared to the single-agent context (Hernandez-Leal et al., 2019; Zhou et al., 2020; Li et al., 2022). This issue arises not only on temporal scales, as observed in assigning sparse credits within a trajectory, but also in structural scales, regarding credit assignment among agents (Agogino & Tumer, 2004). *Value decomposition* serves as the primary paradigm for addressing structural credit assignment (Devlin et al., 2014; Nguyen et al., 2018; Foerster et al., 2018; Sunehag et al., 2018; Rashid et al., 2020b; Son et al., 2019; Rashid et al., 2020a; Wang et al., 2021a; Böhmer et al., 2020; Kang et al., 2022). Consequently, most studies tackling temporal and structural credit assignment problems follow the "`value decomposition+x`" framework.

In directly learning of assigning credits along a trajectory in sparse environments, the primary challenge lies in the inherent difficulty each agent faces while attempting to acquire a substantial number of beneficial trajectories through random exploration (Mesnard et al., 2021). One prevalent example of "`x`" involves learning with efficient exploration, which shows efficacy in task selection (Mahajan et al., 2019; Yang et al., 2020; Zheng et al., 2021; Li et al., 2021a; Zhang et al., 2021; Gupta et al., 2021). Nevertheless, solely employing exploration still falls short of ascertaining which actions yield rare non-zero rewards (Jeon et al., 2022; Hao et al., 2023). Subgoal-based methodologies have recently emerged as a viable alternative inspired by approaches utilizing subgoals in single-agent RL (Kulkarni et al., 2016; Bellemare et al., 2016; Pathak et al., 2017; Burda et al., 2019; Ecoffet et al., 2021; Guo et al., 2022) (also referred to as intrinsically motivated RL (Colas et al., 2022)). Intuitively, these methods decompose a task into a series of goals while concurrently breaking down each goal into subgoals that require completion by agents. Both time and structural scales could receive dense and goal-directed rewards, which mitigats the credit assignment dilemma.

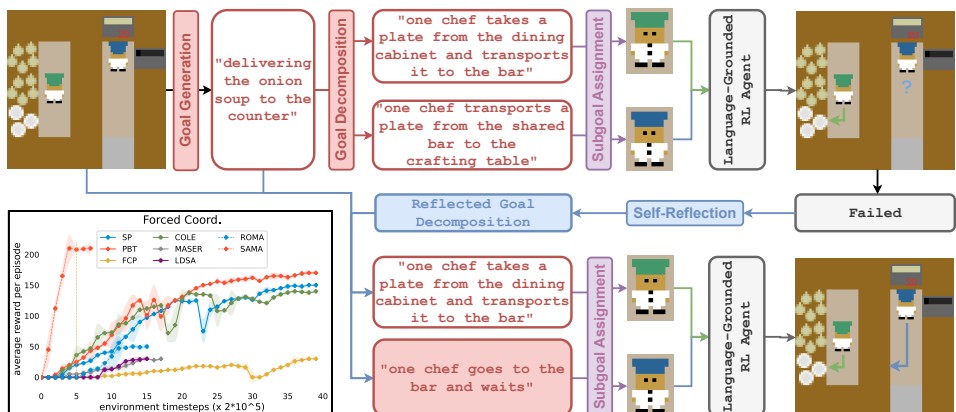

Figure 1: SAMA's proficiency progression and experimental trials on the **Forced Coordination** within `Overcooked`. SAMA necessitates merely approximately 10% of the training instances to approach the SOTA performance. In executing the task, SAMA initially prompts the PLM to generate goals, decompose the goal to subgoals, and apportion subgoals based on the individual agent's status, semantically aligned. Next, the cultivated language-grounded MARL policy adheres to the designated subgoals and engages with the environment. In the event of unsuccessful sub-objective execution, SAMA prompts the PLM again to re-strategize, drawing upon the self-reflection mechanism.

While promising, four primary problems must be addressed within subgoal-based MARL: **a)** sequential goal generation; **b)** rational decomposition of goals into subgoals; **c)** accurate subgoal allocation to agents; and **d)** agents' efficient achievement of subgoals. Early solutions for the above issues primarily rely on artificially predefined rules (Becht et al., 1999; Lhaksmana et al., 2018; Min et al., 2018; Grote et al., 2020), which limits their dynamism and adaptability in new environments. Consequently, *automatic subgoal generation* (ASG) serves as a more practical implementation of "x", permitting application to various tasks without relying on domain knowledge.

Predominant ASG methods typically encapsulate solutions for the four identified problems within a two-stage, end-to-end learning procedure. Initially, subgoals for each agent's completion are generated, followed by the learning of policies that facilitate the attainment of these subgoals. These approaches can be broadly classified into two categories depending on the representation of subgoals. Some methods employ embeddings for encoding subgoals, allowing their generation to be based on local observations (Wang et al., 2020; 2021b), local message-passing (Li et al., 2021b; Shao et al., 2022), or selection from a pre-generated subgoal set using global information (Yang et al., 2022a). Subsequently, goal-conditioned policies are learned through standard reward maximization. In contrast, other techniques implicitly transform subgoals into intrinsic rewards, optimizing policies for subgoal completion by maximizing the shaped reward (Phan et al., 2021; Jeon et al., 2022; Nguyen et al., 2022; Yang et al., 2022b; Li et al., 2023a).

Nevertheless, the end-to-end learning of complex task planning **(a–d)**, with sparse rewards and without a priori knowledge, undoubtedly necessitates massive training samples. Furthermore, existing ASG methods often incorporate representation learning techniques promoting diversity to learn effective subgoal embeddings or intrinsic rewards. Given the rich sources of novelty in complex environments (Burda et al., 2019), this may result in the "over-representation" of subgoals, generating numerous redundant subgoals of limited relevance to the task reward and decreasing the algorithm's sample efficiency. Contrarily, humans do not uniformly explore goal spaces; instead, they rely on commonsense to generate plausibly functional goals (Du et al., 2023). Take the `Overcooked` (Carroll et al., 2019) as an example, humans immediately deduce that ingredient preparation must precede cooking, while utilizing utensils is necessary for food handling prior to serving.

In the realm of disentangled representation learning (DRL), the process exists to isolate the underlying factors of variation into variables with semantic significance. It leads to the acquisition of representations that mimic the comprehension process of humans grounded in commonsense (Bengio et al., 2013; Wang et al., 2022b). As a semantically aligned learning scheme, DRL has evinced its efficacy in enhancing model effectiveness, explainability, controllability, robustness, and generalization capacity,

all of which are indispensable for general decision-making. In contrast to DRL, which necessitates the learning of semantically aligned and disjointed representations, we switch from the "`value decomposition+x`" framework, and proffer an approach to *disentangled* decision-making. The proposed method leverages pre-trained language models (PLMs) as a trove of priors and common-sense for the automatic generation of *semantically aligned and disjointed* subgoals in cooperative MARL with sparse reward. As probabilistic models trained on extensive open-world text corpora, PLMs encapsulate rich data on human knowledge (Bommasani et al., 2021; Yang et al., 2023).

The proposed **S**emantically **A**ligned task decomposition in **MARL** (**SAMA**), employs PLM prompting with chain-of-thought (Wei et al., 2022) (CoT) to suggest potential goals and provide suitable goal decomposition and subgoal allocation. The PLM-based task planning is based on the current environment state, the agents' current local contexts, and the language task manuals (see Figure 1). SAMA incorporates language-grounded MARL (Hanjie et al., 2021; Ding et al., 2023) to train each agent's goal-conditioned policy, enabling the agents to comprehend the natural language subgoals produced by PLMs. Additionally, acknowledging current PLM limitations, such as hallucinations, SAMA leverages the *self-reflection* mechanism (Yao et al., 2023; Shinn et al., 2023; Madaan et al., 2023; Chen et al., 2023) to prompt PLM re-planning of tasks when errors occur.

In summary, this paper's main contributions are three-fold: 1) the introduction of a novel algorithmic framework, SAMA, that implements disentangled, commonsense-driven automatic subgoal generation by prompting PLMs to alleviate the credit assignment problem in MARL; 2) the incorporation of a language-grounding mechanism, that enables each agent to learn an MARL policy conditioned on a natural language subgoal for efficient MARL and PLM cooperation; and 3) the demonstration of SAMA's considerable advantage in sample efficiency compared to state-of-the-art subgoal-based MARL methods, as evidenced by performance on `Overcooked` and `MiniRTS`.

## 2 PROBLEM FORMULATION

This paper examines a fully cooperative multi-agent setting which can be characterized by a decentralized, partially observable Markov decision process (DEC-POMDP, Peshkin et al. 2000; Bernstein et al. 2002) $\langle \mathcal{I}, \mathcal{S}, \{\mathcal{A}_i\}_{i=1}^N, \{\mathcal{O}_i\}_{i=1}^N, \mathcal{P}, \mathcal{E}, \{\mathcal{R}_i\}_{i=1}^N \rangle$, wherein $\mathcal{I}$ signifies the domain of $N$ agents. The environmental state is denoted by $s \in \mathcal{S}$. Agent $i$ is solely able to access a localized observation $o_i \in \mathcal{O}_i$ following the emission function $\mathcal{E}(o_i \mid s)$. At each discrete timestep, an individual agent $i$ opts for an action $a_i \in \pi_i(a \mid o_i)$, originating a jointly executed action $\boldsymbol{a} = \langle a_1, \ldots, a_n \rangle \in \times \mathcal{A}_i$. This action consequently leads to the subsequent state $s'$ according to the transition function $P(s' \mid s, \boldsymbol{a})$ and procures a mutually experienced reward $r = \mathcal{R}(s, \boldsymbol{a})$.

We endeavor to develop an agent capable of addressing arbitrary long-horizon sparse reward tasks utilizing linguistic task manuals. To achieve this, we contemplate employing planning, amalgamating language-grounded MARL policies tailored to fulfill subgoals with a planner that proposes semantically aligned goals, furnishes appropriate goal decomposition, and apportions subgoal allocations, taking into account the environment's current state, agents' extant local contexts, and linguistic task manuals. We presume that task planning adheres to Markov properties, implying that the planner only necessitates completing the planning above operations under the environment's current contexts.

Explicitly, we employ a PLM planner, i.e., GPT-3.5 (OpenAI, 2022) and GPT-4 (OpenAI, 2023), to leverage its encoded commonsense knowledge. The higher-level PLM planner recommends the next semantically aligned goal the multi-agent system is required to achieve, $g_k$, at each round $k$, endeavoring to parse it into $N$ distinct subgoals $\langle g_k^1, \cdots, g_k^N \rangle$. Subsequently, the PLM planner individually assigns the $N$ subdivided subgoals to the corresponding $N$ lower-level agents. For each agent and its correlated subgoal $g_{t_k}^i$ and the task manual $z$ at each round $k$, a language-grounded MARL policy $\pi\left(a_{t_k}^i \mid o_{t_k}^i, z, g_k^i\right)$ samples an action $a_{t_k}^i$ contingent upon the current local observation $o_{t_k}^i$, where $t_k$ represents the timesteps in round $k$. Since completing a multi-agent task requires achieving multiple transition goals, an episode contains multiple rounds.

## 3 SEMANTICALLY ALIGNED TASK DECOMPOSITION

This section initially provides an outline (Figure 2) of the proposed SAMA, which comprises three components: task decomposition, self-reflection, and language-grounded MARL. The task

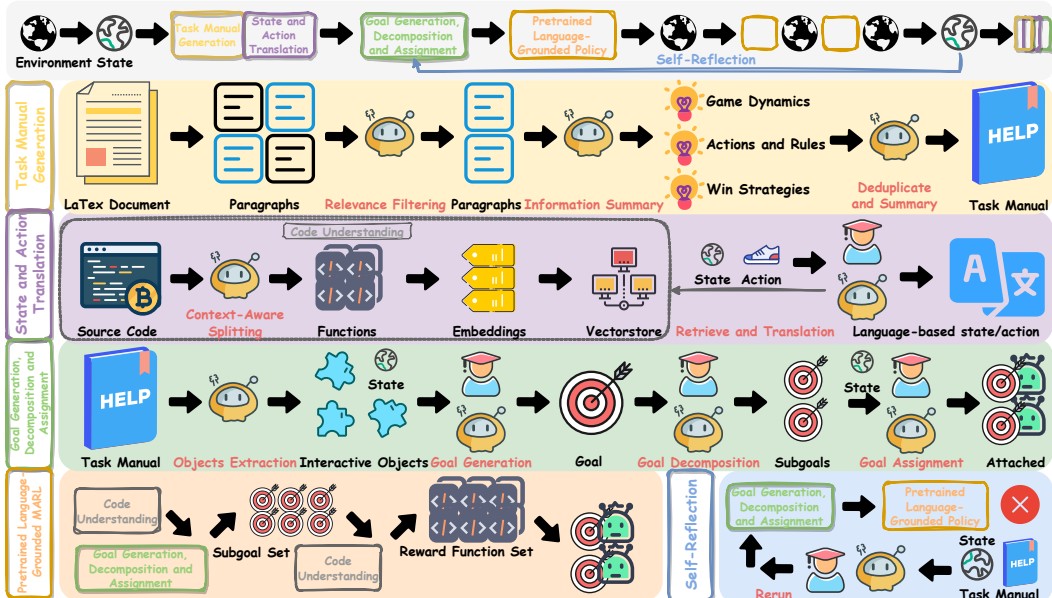

Figure 2: Illustration of task planning procedure and core modules of SAMA. SAMA uses a hierarchical framework for task planning. The higher-level is based on goal generation, decomposition and subgoal assignment of PLMs (🧠), and the lower-level is based on subgoal achievement of language-grounded MARL agent. In order to complete the upper-level tasks, SAMA needs to generate task manuals and translate states and actions based on PLMs in advance. In order to complete lower-level tasks, SAMA also needs to generate diversified subgoals and design reward functions based on PLMs in advance. SAMA also introduces a self-reflection mechanism to improve task planning success rate. The SAMA framework is highly automated and requires only a few-shot example designs (🧑), which greatly improves SAMA's generalization ability for different tasks.

decomposition accomplishes credit assignment on both temporal and structural scales; language-grounded MARL builds upon task decomposition, interacting with the environment to achieve a series of transition goals; the self-reflection, grounded in a trial-and-error paradigm akin to RL, assimilates interaction outcomes to optimize the task decomposition process. In Section 3.1, we will discuss the prerequisite preprocessing stages for the seamless functioning of SAMA's three core components. In Section 3.2, we will elaborate on realizing semantically aligned task decomposition by employing multi-stage and multi-faceted prompting on PLM; In Section 3.3, we examine the training of language-grounded MARL agents, ensuring the agent's policy corresponds closely to language objectives while promoting collaboration among agents. In Section 3.4, we expound on the self-reflection mechanism's optimization of the task decomposition process, encompassing goal generation, decomposition, and assignment, making it more effective based on the outcome of language-grounded policy execution. All prompts are shown in Appendix F.

## 3.1 PREPROCESSING

As will be elucidated, task decomposition and self-reflection are accomplished via PLM prompting based on environmental states or history, agents' local contexts, and task manuals, all represented in natural language. Nevertheless, linguistic task manuals still need to be made available for general multi-agent tasks. Furthermore, except for a handful of text-based tasks, typical multi-agent environment interfaces cannot provide text-based representations of environmental or agent state information and action information. Consequently, task, state, and action translation is essential. We employ PLM for translation to maximize SAMA's adaptability.

**Task Manual Generation.** Drawing inspiration from SPRING (Wu et al., 2023), we generate the language task manuals directly from the LaTeX code of the original paper of the multi-task. We illustrate the process in the yellow line of Figure 2. Initially, we compose gameplay-specific prompts

and obtain the task manual by directing the PLM with prompts for the LaTeX file. As a substantial portion of the paper is unrelated to gameplay, we apply the relevant prompt set $Q_{\text{rel}}$ to determine relevance and the game prompt collection $Q_{\text{game}}$ to summarize gameplay and action space pertinent information. We then dissect the entire LaTeX file into paragraphs $\{\mathcal{S}_{\text{para}}^i\}$ to avoid surpassing input length constraints for most PLMs. For each paragraph $\mathcal{S}_{\text{para}}^i$, we filter paragraphs for relevance and retain only those deemed relevant by at least one prompt from $Q_{\text{rel}}$. We designate $P_{\text{rel}}$ as the set of relevant paragraphs. Subsequently, for each prompt in $Q_{\text{game}}$ and each paragraph in $P_{\text{rel}}$, we instruct the PLM to generate corresponding answers. Ultimately, we prompt the PLM to deduplicate and summarize all answers to generate the final language task manual.

**State and Action Translation.** As states (encompassing the environment condition and the agent's local context) and actions are characterized by code instead of paper, we utilize environment code files for state and action translation. However, given the non-contiguous semantics of code files, we cannot segment and process them individually like LaTeX files. Fortunately, some successful code understanding applications exist, such as Github Copilot, Code interpreter, and the open-source implementation of LangChain (Chase, 2022). Thus, we execute state translation based on LangChain (purple line in Figure 2). Explicitly, LangChain first segregates the entire code project into multiple documents according to code functions and classes, utilizing context-aware splitting. Each document is then encoded and stored in a vector database. Eventually, PLM is augmented with the capacity to respond to relevant questions by retrieval from the database. The inquiry, or prompt, consists of the state and translation instruction.

## 3.2 SEMANTICALLY ALIGNED TASK DECOMPOSITION

We can accomplish semantically aligned task decomposition by prompting the PLM upon obtaining the text-based task manual alongside the environment and agent state translation. The entire process may be broadly classified into 3 stages (green line in Figure 2): goal generation, decomposition, and subgoal assignment. Exploring a goal space non-uniformly, even with meaningful semantics, can be prohibitively costly in specific tasks. This occurs due to the open-ended tasks, rendering the goals highly abstract. Consequently, this paper centers on more straightforward tasks for preliminary investigation. Similar to most multi-agent environments and extant literature on PLM-based decision-making (Du et al., 2023; Wang et al., 2023), SAMA primarily concentrates on addressing tasks necessitating continuous interactions with particular objects in the environment.

**Goal Generation and Decomposition.** As task completion demands interactions with environmental objects, transition goals must comprise objects. Before goal generation, we prompt the PLM to extract interactive objects from the task manual similar with (Varshney et al., 2023). Subsequently, by constructing few-shot examples, we prompt the PLM to generate rational transition goals and decompose them into subgoals equivalent to the number of agents predicated on the extracted objects and current environmental state information, thereby effectuating temporal scale credit assignment. It is important to note that object extraction is not obligatory, as goals can be generated based solely on the task manual. Nevertheless, our experiments demonstrate that object extraction considerably constrains the goal space produced by the PLM, enhancing the training efficiency of language-grounded MARL and subsequently augmenting the final algorithm performance.

**Subgoal Assignment.** Following the transition goal's decomposition into several subgoals equal to the agent count, we ultimately prompt the PLM to establish a one-to-one correspondence between the subgoals and the agents, relying on the task manual, decomposed subgoals, current environmental, and agent state. This culminates in the completion of the structural scale credit assignment.

## 3.3 LANGUAGE-GROUNDED MARL

Following the subgoal assignment, a low-level policy is required to efficiently guide the agent in accomplishing the designated subgoal. In light of the constraints imposed by pre-trained language models (PLMs) on low-level decision-making (Du et al., 2023; Wang et al., 2023), we employ a goal-conditioned MARL policy for this purpose. However, conventional MARL tasks predominantly operate in non-textual observation spaces reliant on image- or numeric-derived information. Therefore, MARL agents must establish a connection between text-based objectives and non-textual observations (or trajectories) to fulfill subgoals effectively.

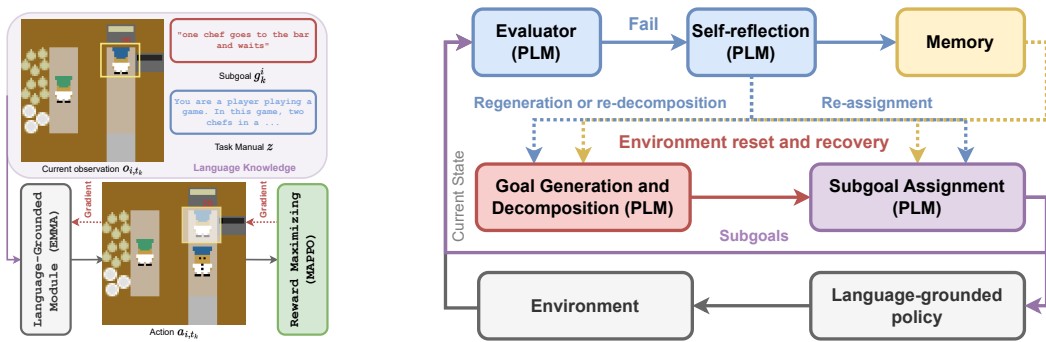

Figure 3: **L:** Language-grounded MARL (blue agent viewpoint); **R:** One self-reflection trail.

To achieve this, we incorporate the language-grounded MARL for training the goal-conditioned policy (Figure 3 Left), which facilitates the final stage of task planning—specifically, the efficient attainment of subgoals based on natural language. Referring to Section 2, upon completion of the subgoal assignment, agent $i$ receives the instruction manual $z$ and a language-based subgoal $g_k^i$ at iteration $k$. Concurrently, at each time step $t_k$, agent $i$ acquires an observation $o_{i,t_k}$ and produces a distribution over action $\pi\left(a_{i,t_k} \mid o_{i,t_k}, z, g_k^i\right)$. The essence of language-grounded MARL involves parameterizing policy $\pi$ by incorporating a language grounding module to generate a language-based representation $X = \text{Ground}(o_{i,t_k}, z, g_k^i)$, reflecting the relationship between the goal, manual, and observation. Agents endeavor to comprehend game dynamics and general subgoals by interacting with environmental entities, ultimately updating the policy parameters using a standard MARL algorithm (such as MAPPO; Yu et al. 2022).

However, two primary challenges must be addressed when integrating language-grounded techniques into the SAMA framework: 1) Online subgoal generation is computationally expensive, as continuous PLM queries during language-grounded MARL training lead to significant temporal and financial costs. 2) A reward function capable of indicating subgoal completion is lacking. In tasks with sparse rewards, rewards are assigned only upon completion, such as onion soup delivery in `Overcooked` or team victory in `MiniRTS`. We discuss the respective solutions below (orange line in Figure 2).

**Training Dataset Generation.** To diminish the cost associated with PLM queries and reduce language-grounded MARL training durations, we generate the training dataset in advance. The training dataset for language grounding is consist of many subgoals and corresponding reward functions. For subgoals, the technology employed in *state and action translation* from Section 3.1 is reused to convert PLM prompt content from "translation" tasks to the generation of a diverse set of states, $\mathcal{D}_{\text{state}}$. Subsequently, the process prompt PLM in Section 3.2 is applied to generate a corresponding subgoal set $\mathcal{D}_{\text{subgoal}}$ for derived states. For reward functions, the PLM is also utilized in designing the reward function to minimize human intervention within the SAMA framework and enhance generalizability. Prior work has validated the feasibility of this idea (Kwon et al., 2023; Yu et al., 2023). Initially, we de-duplicate subgoals in $\mathcal{D}_{\text{subgoal}}$ by prompting the PLM, similar to the *task manual generation* process from Section 3.1, to obtain a novel subgoal set $\mathcal{D}'_{\text{subgoal}}$. Next, inspired by prompt engineering in Yu et al. (2023), we adopt the technology from *state and action translation* in Section 3.1 for PLM querying to generate the corresponding `Python` code snippet capable of assessing subgoal completion. The methods within the code snippet set $\mathcal{D}_{\text{code}}$ accept the current state as input and return either $0$ or $1$. The returned values serve as binary rewards in training the language-grounded MARL.

**Remarks.** To monitor the training progress, we evaluate the agents at a very low frequency (Figure 5 is drawn according to the evaluation results). In light of the potential emergence of unaccounted environmental states during the evaluation process, periodic inclusion of the encountered states and corresponding PLM-generated subgoals to the dataset $\mathcal{D}_{\text{subgoal}}$ is undertaken. Moreover, an advantage of introducing PLM as a higher-level task planner is that we do not need to learn how to split subgoals, as in Wang* et al. (2020). This allows us to significantly reduce the training difficulty and sample complexity when training the language-grounded MARL algorithm. Additionally, it should be noted that SAMA's compatibility extends to any language grounding module. Given the absence of prior testing on the `Overcooked` task for extant language-grounded RL techniques, we opted for the

facile implementation of the `EMMA` model (Hanjie et al., 2021; Ding et al., 2023). As for `MiniRTS`, a corresponding language-grounded RL algorithm (Xu et al., 2022) exists; thus, we proceeded to retrain using its publicly available pretrained model. Refer to the Appendix E for more details.

### 3.4 SELF-REFLECTION

Prevalent PLMs frequently yield to typical errors, including hallucination or arbitrary decision-making, attributable to the incapacity to assimilate knowledge from long trajectories to refine subsequent planning. We adopt a self-reflection mechanism for discerning hallucination and suboptimal planning (Shinn et al., 2023). Employing this technique, the PLM planner will rectify (as necessitated) the semantically aligned goal generation, decomposition, or subgoal assignment contingent upon the outcomes derived from the language-grounded MARL agent's subgoal fulfillment.

As shown in blue line in Figure 2 and Figure 3 (Right), the self-reflection module contains two core steps, namely evaluation and reflection. Concretely, after the goal generation, decomposition and subgoal assignment, the language-grounded agent $i$ carries out a sequence of actions in response to the assigned subgoal $g_k^i$ at each round $k$. At each timestep $t_k$, the agent $i$ executes an action $a_{i,t_k}$ and the state transits to $o_{i,t_k+1}$. The evaluator, or the code snippet in $\mathcal{D}_{\text{code}}$ corresponding to the subgoal $g_k^i$, will determine whether $g_k^i$ has been completed. SAMA then calculates a heuristic $h$ predicated on the binary return value, potentially eliciting self-reflection. The application of heuristic $h$ in this paper is uncomplicated: if the return values *all* equal $1$ (indicating successful execution of the subgoal), self-reflection will not be prompted; conversely, it will be invoked.

If $h$ suggests self-reflection, SAMA prompts the PLM planner to deliberate on its extant semantically aligned goal generation, decomposition, or assignment following self-reflective memory and environmental state. Since PLM needs to regenerate goals or subgoals during the self-reflection, we need to recovery the environment to the previous state. For this, we record the subgoals generated sequentially during the task planning. In this way, the state can be recovered by resetting the environment and calling the lower-level language-grounded policy conditioned on the logged subgoal sequence. The above "reset-recovery" procedure is denominated as a *self-reflection trial*. Otherwise, SAMA prompts the PLM planner for subsequent semantically aligned planning. Since most MARL tasks are episodic and language-grounded MARL policy is fixed in evaluation, the above reset-recovery process is doable. In practice, we set a hyperparameter limit of a maximum of 3 reflections retained within the agent's history. The trial is discontinued if the agent surpasses the utmost quantity (we also set it to 3 in the following experiments). In quantifying the empirical performance, the instances encountered during self-reflection trials are not incorporated.

## 4 EXPERIMENTS

### 4.1 CASE STUDY: OVERCOOKED

In this section, we carry out an array of experiments in the `Overcooked` environment (Carroll et al., 2019; Charakorn et al., 2020; Knott et al., 2021; Li et al., 2023b), specifically devised to address sparse-reward, long-horizon, and coordination conundrums (Figure 4). In this dyadic common payoff game, each participant manipulates a chef within a culinary setting, collaboratively preparing and delivering soup, thereby accruing a shared reward of 20 for the team.

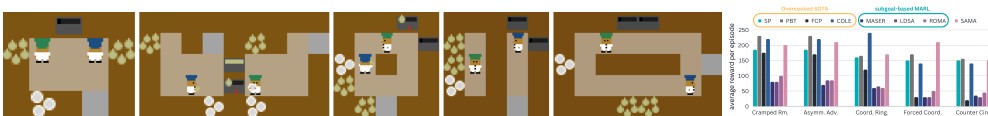

Figure 4: **Left:** Screenshots of the `Overcooked` layout (Carroll et al., 2019): Cramped Room, Asymmetric Advantages, Coordination Ring, Forced Coordination, and Counter Circuit. **Right:** The mean self-play rewards of episodes over 400 timesteps under 10 random seeds. SAMA can approach or even exceed the performance of SOTA algorithm. Refer to learning curves for variances.

We evaluate our approach in juxtaposition with alternative techniques, encompassing the SOTA method on the `Overcooked` task, i.e., selfplay (Tesauro, 1994; Carroll et al., 2019), PBT (Jaderberg

et al., 2017; Carroll et al., 2019), FCP (Strouse et al., 2021), and COLE (Li et al., 2023b), all of which employ PPO (Schulman et al., 2017) as the RL algorithm. In addition, we assess the ASG method, MASER (Jeon et al., 2022), LDSA (Yang et al., 2022a), and ROMA (Wang et al., 2020). Given that the prevailing SOTA methodologies on the `Overcooked` task primarily target zero-shot coordination dilemmas, we utilize self-play rewards to exhibit their performance.

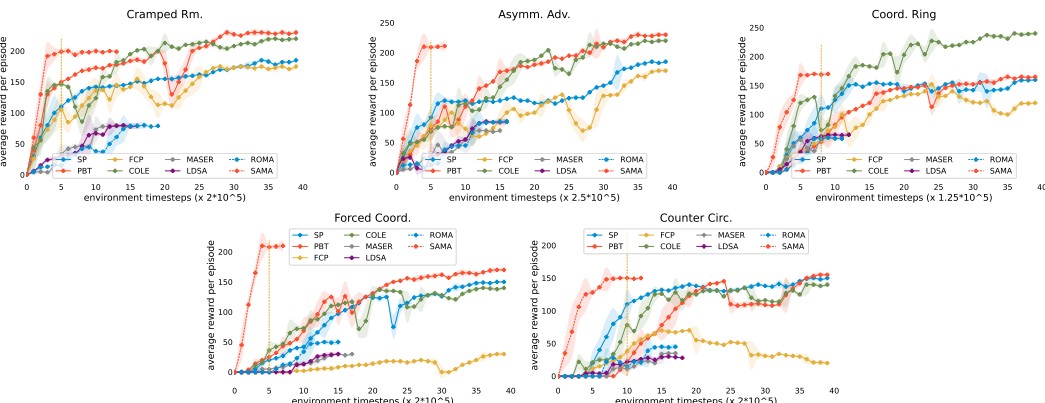

Figure 5: Pretrain learning curves of each methods in five `Overcooked`'s layouts.

Figure 4 (Right, see Figure 10 in Appendix E for more details) illustrates the mean rewards per episode over 400 timesteps of gameplay employing 10 random seeds. Figure 5 presents learning curves. Synthesizing these figures yields the following insights: (1) The prevailing SOTA in `Overcooked` necessitates a substantial quantity of training samples; (2) The extant ASG methodology exhibits suboptimal performance in long-horizon, sparse-reward, and cooperation tasks exemplified by `Overcooked`; (3) Capitalizing on human commonsense embedded within PLM, SAMA can approximate or even surpass the performance of the SOTA by utilizing merely five to one-tenth of the training samples due to semantically aligned task planning; (4) Given the inherent limitations of the current PLM in long-horizon reasoning, SAMA's performance cannot achieve optimal outcomes. It merits attention that complementing the existing ASG method with techniques such as self-play and population-based training, as demonstrated by algorithms like COLE, could engender a significant augmentation in performance. Nevertheless, this semantically non-aligned, end-to-end training devoid of prior knowledge would also entail considerable consumption of training data.

## 4.2 CASE STUDY: MINIRTS

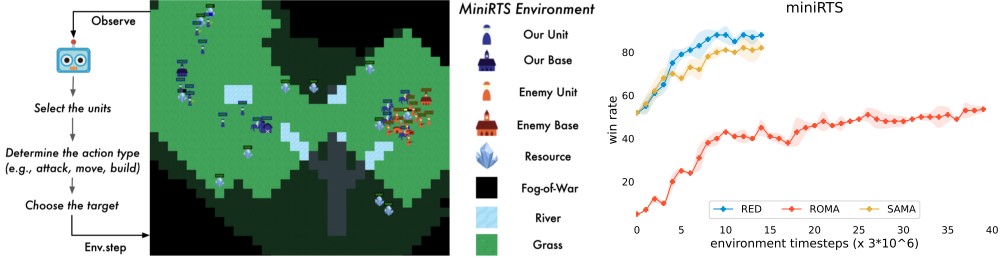

Figure 6: **Left:** Screenshots of the `MiniRTS` (Hu et al., 2019; Xu et al., 2022), which is a real-time strategy game where the player in blue needs to control its units to kill the enemy units in red. **Right:** Average win rates based on 100 test games and repeated over 3 random seeds. The pretrain learning curve of SAMA is that of the language-grounded MARL module.

This section elucidates the efficacy of SAMA concerning intricate tasks. `MiniRTS` (Hu et al., 2019), a grid-world environment, encapsulates salient features of elaborate real-time strategy games (see Figure 6). The setup consists of two entities: a player (blue) governed by human/policy intervention, juxtaposed against a built-in script AI (red). Key objectives for the player include resource acquisition, construction, and either obliterating enemy units or demolishing the enemy base to secure victory. 7

distinct unit types establish a rock-paper-scissors attack dynamic (Figure 9), with the environment facilitating a sparse reward mechanism. At a game's conclusion, the reward amounts to 1 if the agent wins and $-1$ in the event of a loss. During other timesteps, a reward of 0 is attainable.

MiniRTS furnishes an array of built-in script AIs, and we choose the medium-level AI as the opponent to facilitate expeditious prompt design for PLMs. Initially, this script dispatches all three available peasants to mine the nearest resource unit. It then randomly selects one of the seven army unit types, determining an army size $n$ ranging between 3 and 7. A building unit corresponding to the selected army is constructed, followed by training $n$ units of the elected army category for deployment in attacks. The script perpetuates army unit training, maintaining an army size of $n$.

In contrast to `StarCraft II` (Samvelyan et al., 2019; Ellis et al., 2022), where individual unit control strategies may be designated, `MiniRTS` employs one strategy (agent) to govern all units. To convert the latter into a multi-agent undertaking, we adopt a straightforward modification of its environment from RED (Xu et al., 2022). Two agents emerge in the modified environment, each governing half of the units. Analogously, we also modestly modified the built-in medium-level AI script, enabling the random selection of two types of army units in each iteration.

Given that the built-in script AI constructs only two army unit types per game, we establish an oracle prompt design strategy following the ground truth of enemy units and the attack graph. In the game's nascent stages, the PLM instructs the agent to construct suitable army units progressively. Subsequently, it transmits `NA` and encourages the policy to operate autonomously, analogous to RED. A competent language-grounded RL policy adheres to these directives, assembling the proper army units and executing actions independently, yielding a higher win rate.

Our language-grounded MARL agent commences with RED's pretrained policy, which exemplifies the SOTA in `MiniRTS`. In addition to RED, we examine ROMA as an alternative baseline. Simulation win rates in Figure 6 (right) derive from 100 test games, repetitive across 3 random seeds. ROMA does not constitute a language-grounded MARL algorithm, so RED's pretrained model is inapplicable and must be learned from scratch. Observations analogous to Overcooked emerge. In addition, while RED, leveraging Oracle commands, boasts unsurpassed performance, SAMA approximates RED's effectiveness due to the utilization of human commonsense embedded in the PLM.

## 5 Closing Remarks and Limitations

We introduce an innovative approach, SAMA, to tackle the "sample scarcity" and "over-representation on goals" challenges prevailing in cutting-edge MARL techniques when addressing credit assignment concerns. SAMA prompts pre-trained language models with chain-of-thought, enabling the proposal of semantically aligned goals, facilitating appropriate goal decomposition and subgoal allocation, and endorsing self-reflection-driven replanning. Moreover, SAMA integrates language-grounded MARL to train each agent's subgoal-conditioned policy. Our findings demonstrate that such innate predispositions in PLMs are advantageous for agents engaged in long-horizon, sparse-reward, and highly collaborative tasks, such as `Overcooked` and `MiniRTS`, necessitating common-sense behaviors that alternative methods cannot distill through end-to-end learning. This proves beneficial in circumstances featuring an extensive spectrum of potential behaviors, of which only a few could be deemed feasibly utilitarian. Conversely, it may be less advantageous in settings with restricted scope for goal-conditioned learning—where human rationality is immaterial or inexpressible in language or when state information is not inherently encoded as a natural language sequence.

In the tasks examined, the efficacy of PLM remains contingent upon prompt selection. Despite well-curated prompts and self-reflective mechanisms, PLMs occasionally err due to omitted domain-specific knowledge. Various avenues exist to surmount this constraint, including incorporating domain expertise within PLM prompts or fine-tuning PLMs on task-specific data. Furthermore, the quality of suggestions markedly augments with model sizes. The recurring prompting of massive PLMs might be temporally and financially unwieldy in specific MARL settings. As general-purpose generative models emerge across arenas beyond text, SAMA-inspired algorithms could be employed to generate reasonable visual goals or goals in alternate state representations. Consequently, SAMA may function as a foundation for subsequent research, facilitating the development of increasingly adaptable and versatile methodologies to incorporate human contextual understanding into MARL.

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

# Supplementary Material

## Table of Contents

## A  RELATED WORK

### A.1  TACKLING CREDIT ASSIGNMENT IN MULTI-AGENT REINFORCEMENT LEARNING

The challenge of credit assignment arises not only on temporal scales but also on structural scales. *Value decomposition*[1] serves as the primary paradigm for addressing structural credit assignment (Devlin et al., 2014; Nguyen et al., 2018; Foerster et al., 2018; Sunehag et al., 2018; Rashid et al., 2020b;

---

[1]In the study by Wang et al. (2021a), a theoretical demonstration is provided regarding the implicit nature of counterfactual difference rewards in individual value decomposition. Owing to this, we propose to include such classic mechanisms for managing structural credit assignment challenges under the broad classification of value decomposition, albeit with some degree of flexibility.

Son et al., 2019; Rashid et al., 2020a; Wang et al., 2021a; Böhmer et al., 2020; Kang et al., 2022). Consequently, most studies tackling temporal and structural credit assignment problems employ the "value decomposition+x" framework.

One prevalent example of "x" involves learning with efficient exploration, which shows efficacy in task selection (Mahajan et al., 2019; Yang et al., 2020; Zheng et al., 2021; Li et al., 2021a; Zhang et al., 2021; Gupta et al., 2021). Subgoal-based methodologies have recently emerged as a viable alternative inspired by approaches utilizing subgoals in single-agent RL (Kulkarni et al., 2016; Bellemare et al., 2016; Pathak et al., 2017; Burda et al., 2019; Ecoffet et al., 2021; Guo et al., 2022) (also referred to as intrinsically motivated RL (Colas et al., 2022)). Intuitively, these methods decompose a task into a series of goals while concurrently breaking down each goal into subgoals that require completion by agents. In doing so, both time and structural scales receive dense, goal-directed rewards, mitigating the credit assignment dilemma.

## A.2 SUBGOAL-BASED MULTI-AGENT REINFORCEMENT LEARNING

Numerous antecedent studies have delved into subgoal-based MARL characterized by sparse rewards. Early solutions primarily rely on artificially predefined rules (Becht et al., 1999; Lhaksmana et al., 2018; Min et al., 2018; Grote et al., 2020), which limits their dynamism and adaptability in new environments. Consequently, *automatic subgoal generation* (ASG) serves as a more practical implementation of "x," permitting application to various tasks without relying on domain knowledge.

The early endeavors of ASG primarily revolved around a semi-end-to-end framework, wherein the generation of subgoals occurred automatically, but the lower-level policies fulfilling these subgoals were pre-trained. HDMARL (Tang et al., 2018) introduces a hierarchical deep MARL framework devised to tackle sparse-reward challenges via temporal abstraction. HDMARL selects a singular subgoal from a predefined assortment contingent upon domain-specific knowledge; consequently, such subgoal construction proves inapplicable across disparate tasks. Works by Xiao et al. (2020; 2022) impose further constraints by necessitating the pre-definition of goal-conditioned policies corresponding with each respective goal.

Predominant ASG methodologies predominantly integrate resolutions for the four identified problems (**(a-d)** in Section 1) within a bifurcated, end-to-end learning process. Initially, agent-specific subgoals are conceived, succeeded by learning policies conducive to realizing these subgoals. These stratagems can be broadly compartmentalized into dual classifications contingent upon subgoal representation. Specific methods employ embeddings to encode subgoals, facilitating their generation via local observations (Yang et al., 2020a; Wang et al., 2020; 2021b), local message-passing (Li et al., 2021b; Shao et al., 2022), or selecting from a pre-formulated subgoal compendium utilizing global information (Yang et al., 2022a). After this, goal-conditioned policies are learned through standard reward maximization. Conversely, alternative approaches implicitly convert subgoals into intrinsic rewards, refining policies endorsing subgoal fruition by maximizing these shaped rewards (Phan et al., 2021; Jeon et al., 2022; Nguyen et al., 2022; Yang et al., 2022b; Li et al., 2023a).

## A.3 LANGUAGE GROUNDED REINFORCEMENT LEARNING

Language grounding encompasses the acquisition of natural languages unit comprehension, such as utterances, phrases, or words, by capitalizing on non-linguistic contexts. Numerous antecedent studies have addressed the conveyance of goals or instructions to agents through text, eliciting agent behavior in response to language grounding (Branavan et al., 2012; Janner et al., 2018; Wang et al., 2019; Blukis et al., 2020; Küttler et al., 2020; Tellex et al., 2020; Xu et al., 2022), thus establishing a robust correlation between the provided instructions and the executed policy. Recently, a plethora of research has delved into generalizations from multifaceted vantage points. Hill et al. (2020a;b; 2021) scrutinized the generalization concerning novel entity combinations, extending from synthetic template commands to human-generated natural instructions and object quantities. Choi et al. (2021) introduced a language-guided policy learning algorithm, facilitating expeditious task learning through linguistic adjustments. Moreover, Co-Reyes et al. (2019) advocated using language guidance for policies to achieve generalization in novel tasks by employing meta-learning techniques. An alternative line of inquiry has emphasized the incorporation of task manuals as supplementary information to bolster generalization (Narasimhan et al., 2018; Zhong et al., 2020; Hanjie et al., 2021;

Zhong et al., 2021). In contrast to the research mentioned above, EnDi (Ding et al., 2023) advances the discourse by examining language grounding at the entity level within multi-agent environments.

### A.4 FOUNDATION MODELS FOR DECISION MAKING

Foundation models, trained through extensive datasets, have demonstrated remarkable aptitudes in conjunction with fast adaptability for a plethora of downstream tasks in diverse fields such as vision (Yuan et al., 2021), language (Kenton & Toutanova, 2019; Brown et al., 2020), and cross-modalities (Ramesh et al., 2021; Jiang et al., 2022; Alayrac et al., 2022). Capitalizing on such capabilities, these models have been employed to furnish RL agents with rewards (Gupta et al., 2022; Fan et al., 2022; Kwon et al., 2023); a burgeoning line of research is now investigating using foundation models (particularly PLMs) for refining agent policies. In instances where agent actions are delineated through natural language, PLMs may be employed to devise more sophisticated strategies for long-horizon tasks, as linguistic descriptions of actions are anticipated to exhibit enhanced generalization than low-level motor controls (Huang et al., 2022b;a; Brohan et al., 2023; Wang et al., 2023; Dasgupta et al., 2023; Carta et al., 2023). Furthermore, when agent observations encompass imagery and textual descriptions, vision-language captioning models can augment agent observations with linguistic delineations (Tam et al., 2022; Du et al., 2023; Driess et al., 2023). Remarkably, even in situations where agent states, actions, and rewards do not encompass visual or textual elements, PLMs have been identified as advantageous policy initializers for both offline (Reid et al., 2022) and online RL (Li et al., 2022).

## B INTRINSICALLY MOTIVATED GOAL-CONDITIONED RL

Owing to the intimate correlation between the ASG framework and intrinsically motivated goal-conditioned RL (IMGC-RL; Colas et al., 2022b), a succinct exposition of the IMGC-RL is presented herein. Furthermore, this section delves into the distinctions and connections between the SAMA framework and the IMGC-RL paradigm. The proposal of the IMGC-RL framework primarily endeavors to address tasks characterized by temporal-extended, sparse or delayed rewards. By incessantly generating transitional phase subgoals and devising the corresponding intrinsic reward functions, the framework dissects the original task into a succession of subordinate tasks with shorter-horizons and dense rewards, thereby simplifying problem resolution. However, it is imperative to underscore that the IMGC-RL pertains to a single-agent setting, thereby exhibiting a considerable discrepancy when compared to MARL.

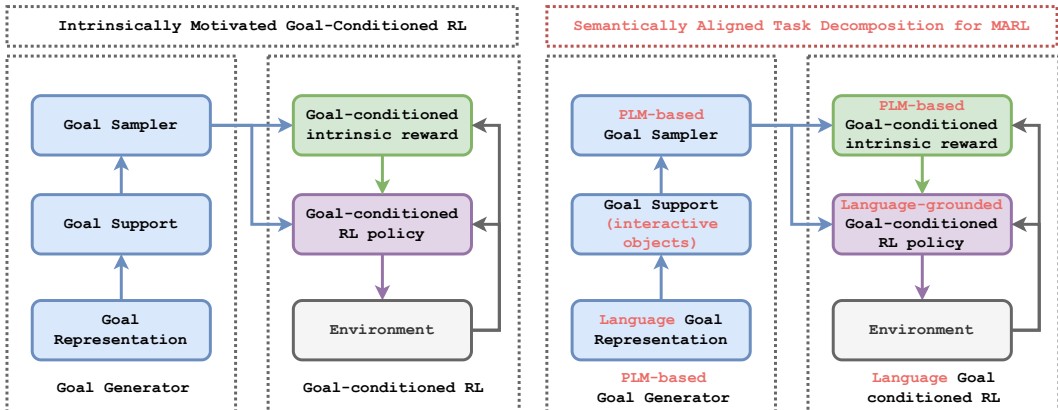

Figure 7: **Left:** The illustration diagram of intrinsically motivated goal-condtioned reinforcement learning (IMGC-RL); **Right:** The proposed SAMA framework under the IMGC-RL context.

From Figure 7, it is discernible that the IMGC-RL framework primarily consists of two modules: the goal generator and the goal-conditioned reinforcement learning. The former may be further subdivided into three submodules, namely goal representation learning, goal support set learning, and goal sampling; the latter encompasses goal-conditioned reward design and goal-conditioned RL policy learning. For an in-depth examination of each module and its submodules, along with

related works, kindly refer to this comprehensive review paper (Colas et al., 2022b). Furthermore, in categorizing existing MARL methodologies based on the ASG framework from the perspective of IMGC-RL, one may broadly distinguish two types according to the presence of goal-conditioned RL: one in which a pre-trained policy stemming from imitation learning replaces the goal-conditioned RL module (Tang et al., 2018; Xiao et al., 2020; 2022), and the other prevalent methodologies incorporates both, employing either a two-phase or end-to-end learning paradigm (Yang et al., 2020a; Wang et al., 2020; 2021b; Yang et al., 2022a; Phan et al., 2021; Jeon et al., 2022; Nguyen et al., 2022; Yang et al., 2022b; Li et al., 2023a).

**Connection with the SAMA.** The SAMA framework bears a remarkably close connection to IMGC-RL. Figure 7 delineates their intricate correspondence in detail. Initially, SAMA renders subgoals as natural language text whilst constraining the subgoal space or support by extracting interactive entities from the environment. Subsequently, SAMA implements goal sampling and the design of goal-conditioned reward functions by prompting PLMs. Ultimately, SAMA accomplishes the training of the goal-conditioned RL policy by incorporating language grounding techniques.

## C  MISSING DISCUSSIONS

There are some PLM-based decision-making methods bear the closest resemblance to our work: DEPS (Wang et al., 2023), ELLM (Du et al., 2023), Plan4MC (Yuan et al., 2023) and GITM (Zhu et al., 2023). DEPS (Wang et al., 2023) engenders the whole goal sequence via prompt PLM, and the goal-conditioned policy is ascertained through imitation learning in advance. In contrast, ELLM (Du et al., 2023), Plan4MC (Yuan et al., 2023) and GITM (Zhu et al., 2023), akin to SAMA, prompts the PLM to generate the subsequent goal contingent on the current state. ELLM (Du et al., 2023) trains goal-conditioned RL policy through synthetically devised goal-oriented, dense intrinsic rewards, Plan4MC (Yuan et al., 2023) and GITM (Zhu et al., 2023), similar with DEPS, adopt the pre-trained or pre-defined goal-condtioned policy to achieve goals.

SAMA transcends a mere augmentation of these works within multi-agent contexts, showcasing marked innovation in following three aspects:

**Refined and Automated Preprocessing.** In contrast to prevailing approaches reliant on manual translations or those confined to specific translated environments, such as Minedojo (Fan et al., 2022), SAMA presents a comprehensive procedural framework for environmental translation, incorporating the generation of task manuals, state, and action translations. Furthermore, by incorporating code understanding technology proposed by LangChain (Chase, 2022), SAMA alleviates the training complexities of language-grounded MARL and enhances self-reflection quality. This refined and automated preprocessing enables SAMA to transition to novel tasks at a less expense.

**Introspective Goal Decomposition.** Devising explicit subgoals for singular agents in sparse-reward multi-agent tasks optimally remains an untenable endeavor for humans. Generally, human aptitude is demonstrated in devising strategies for the aggregate system. Consequently, it can be logically deduced that PLMs, endowed with human knowledge, exhibit similar tendencies. Thus, SAMA deliberately orchestrates a *goal decomposition* (over multiple agents) phase. Moreover, SAMA incorporates a self-reflection mechanism to ameliorate PLM limitations in long-horizon reasoning.

**Automated Goal-conditioned Policy Optimization.** DEPS, Plan4MC and GITM, reliant on imitation-learning-based or predefiend skills, or goal-conditioned policy, necessitates extensive expert data encompassing an array of achieved goals. However, the unbounded nature of the goal space emitted by PLM renders it challenging for these method to be directly applied to alternative tasks; ELLM, contingent on the synthetic crafting of intrinsic rewards, engenders its complexities when extending to a multi-agent context. Reward design constitutes a longstanding and contentious issue in MARL, as both global rewards and local rewards have been shown to be inherently flawed: the former may foster indolence in agents, while the latter might yield egoistic agents (Du et al., 2019; Mao et al., 2020; Wang et al., 2022a; Hua et al., 2023). In contrast, SAMA leverages language-grounded MARL for the *automatic* optimization of a (language-based) goal-conditioned policy.

# D  ENVIRONMENT DETAILS

## D.1  OVERCOOKED

In this section, we carry out an array of experiments in the `Overcooked` environment (Carroll et al., 2019; Charakorn et al., 2020; Knott et al., 2021; Li et al., 2023b), specifically devised to address sparse-reward, long-horizon, and coordination conundrums. In this dyadic common payoff game, each participant manipulates a chef within a culinary setting, collaboratively preparing and delivering soup, thereby accruing a shared reward of 20 for the team. The `Overcooked` domain we employed encompasses five distinct configurations (refer to Figure 8), namely **Cramped Room**, **Asymmetric Advantages**, **Coordination Ring**, **Forced Coordination**, and **Counter Circuit**.

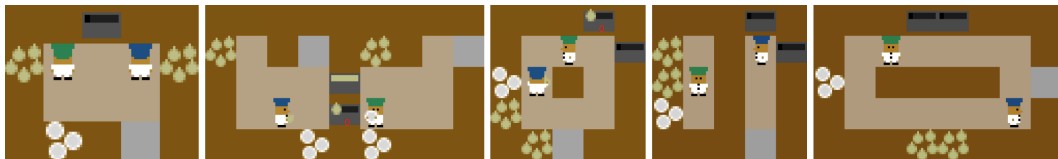

Figure 8: Screenshots of the `Overcooked` layout. From left to right: Cramped Room, Asymmetric Advantages, Coordination Ring, Forced Coordination, and Counter Circuit.

The comprehensive elucidation of the five configurations is presented below (Li et al., 2023b):

- **Cramped Room**. The cramped room constitutes a rudimentary setting wherein two participants are confined to a diminutive space possessing a singular pot (black box with gray base) and one serving point (light gray square). Consequently, individuals are anticipated to exploit the pot to its fullest and effectually dispense soup, even with elementary collaboration.

- **Asymmetric Advantages**. Within this arrangement, two players are situated in separate, isolated kitchens. As the appellation implies, the locations of onions, pots, and serving points exhibit asymmetry. In the left kitchen, onions are remote from the pots, while serving points are proximal to the central region of the configuration. In contrast, within the right kitchen, onions are positioned near the central area, and the serving points are distant from the pots.

- **Coordination Ring**. This annular configuration necessitates both participants to maintain constant motion to avoid impeding one another, particularly in the top-right and bottom-left vertices where the onions and pots are situated. To achieve optimal synergy, both pots should be employed.

- **Forced Coordination**. The Forced Coordination layout represents another separation of the two agents. The left side lacks pots or serving points, while the right side is devoid of onions or pots. As a result, the pair must synchronously orchestrate their actions to accomplish the task. The left player is expected to prepare onions and plates, while the right player oversees cooking and serving responsibilities.

- **Counter Circuit**. The Counter Circuit is an additional annular configuration but with an expanded cartographical scope. Within this layout, pots, onions, plates, and serving points occupy four distinct orientations. Constrained by the constricted passageways, players are prone to obstruction. Therefore, coordination and task execution prove challenging in this environment. Individuals must acquire the advanced technique of positioning onions in the central zone to facilitate rapid exchange, thereby augmenting performance.

We selected `Overcooked` as a focal case within this paper due to the facile transformation of its state space, action space, and reward function into comprehensible natural language text via pre-established scripts. Furthermore, the environment state information readily permits the discernment of successful subgoal completion. Consequently, this facilitates the expeditious training of language-grounded MARL policies and the integration of self-reflection mechanisms.

## D.2  MINIRTS

The ensuing content is gleaned from Hu et al. (2019) and Xu et al. (2022); for a comprehensive elucidation, one is encouraged to consult the source manuscript.

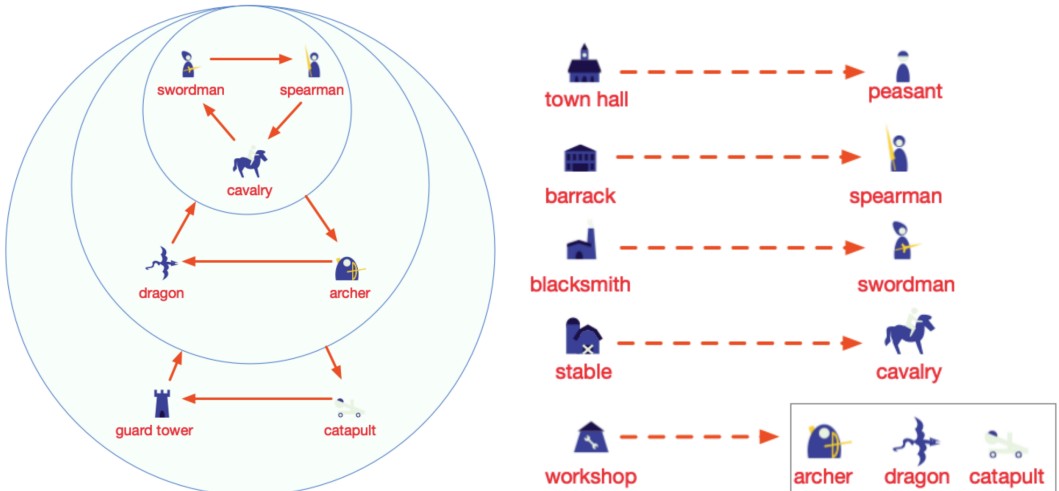

Figure 9: **Left:** Depicted in the screenshots (of RED) are the intricate attack dynamics within `MiniRTS`. Utilizing a rock-paper-scissors attack graph schema, each army type exhibits distinct strengths and vulnerabilities against opposing units. For instance, the "swordsman" efficaciously subdues the "spearman"; conversely, the former is susceptible to the prowess of the "cavalry." The triumvirate of "swordsman," "spearman," and "cavalry" maintains a superior offensive stance against the "archer." **Right:** Various building units generate diverse army units employing resource utilization (in RED). The "workshop,", for example, engenders the creation of "archer," "dragon," and "catapult" units, while alternative buildings yield a singular unit type. Exclusive to the "peasant" is the capability to extract resources from resource units and erect additional building units.

**Units and Map.** `MiniRTS` encompasses three unit classifications: resource, building, and army units. *Resource* units are stationary, invariably present at the game's commencement, and not producible by any entity. Solely "peasants" can extract resources from these resource units for building specific buildings or army units. Moreover, six stationary *Building* unit categories exist. Five building types can generate specific army units (Figure 9). Conversely, the "guard tower" neither spawns army units nor can launch assaults upon adversaries. "Peasants" erect building units on any vacant map location. Lastly, seven army unit variants can traverse and assail opponents. Except for the "peasant", the remaining six army units and "guard tower" employ a rock-paper-scissors paradigm.

The game map comprises a discrete $32 \times 32$ grid partitioned into grass and river cells. Grass cells permit building unit construction and passage for army units, whereas exclusively "dragon" may cross river cells. The initialized map contains one "town hall," three "peasants," and randomly positioned river cells and resource units.

**Observation Space.** An agent's observation encompasses a $32 \times 32$ map, supplemented by additional game states (e.g., observable units' health points and resource unit quantities). Regions not traversed are masked, and unobserved enemy units are eliminated.

**Action Space.** We follow the variant action space designed by (Xu et al., 2022). Two agents govern half of the total units. A $0/1$ action is appended to each unit, signifying whether it will act or remain *IDLE*. Specifically, agents generate a collective action and a $0/1$ flag for every units. For a unit designated with a 1, it executes the collective action; for a unit labeled 0, the action `CONTINUE` is implemented. After discerning which units will react, the agent then predicts an action type (e.g., `MOVE`, `ATTACK`) and subsequently anticipate action outputs based on the said action type. We collate all accessible action types and their outputs as follows.

- `IDLE`: *None* (the elected units remain stagnant).
- `CONTINUE`: *None* (Units persist with their prior actions).
- `GATHER`: *The target resource unit's ID*.
- `ATTACK`: *The enemy unit's ID*.
- `TRAIN`: *The army type to be trained*.

- `BUILD`:*The building type and the designated $(x, y)$ construction position.*
- `MOVE`: *The designated $(x, y)$ movement position.*

**Reward Function.** Sparse rewards define this environment. The agent garners a reward of $1$ if victorious and $-1$ if defeated. For all other timesteps, the environment yields a reward of $0$.

## E  IMPLEMENTATION AND TRAINING DETAILS

All code will be released soon, licensed under the MIT license (with `Overcooked`, `MiniRTS`, and RED licensed under their respective licenses). Concerning the PLM specifically `gpt-3.5-turbo`, we employ OpenAI's APIs for facilitation. The PLM is exclusively utilized during the pretraining phase of the language-grounded MARL policy (limited to `Overcooked`) and the evaluation processes, thus preventing any undue financial obligations to SAMA's implementation. We incorporate the technique of batch prompting[2] (Cheng et al., 2023) to simultaneously minimize token and temporal expenditures when engendering pretraining subgoals and appraising SAMA's efficacy under varied random seeds. The computing hardware comprises two servers, each with 256GB of memory, and a pair of NVIDIA GeForce RTX 3090s, outfitted with 24GB of video memory. The training time (plus PLM query time) for `Overcooked-AI` is around $43 + 3$ hours and $22 + 3$ hours for `MiniRTS`.

**Runtime**    During the testing process of SAMA, a query to the PLM is only made to generate the next goal and assign subgoals after the language-grounded MARL agents have completed their assigned tasks. While the self-reflection mechanism may require some queries to be repeated multiple times, the number of times (averaging around 20 times) the PLM is queried is relatively tiny compared to the entire episode horizon. According to our rough statistics, the wall time has increased by approximately 1-2 times compared to the baselines.

**Economic Cost for $1$ Episode**    The number of tokens required for each query to the PLM is similar for both `Overcooked-AI` and `MiniRTS`, at around $3,000$ tokens. Therefore, for `Overcooked-AI`, the total number of tokens needed per episode is approximately $20 \times 3,000 = 60,000$. The cost for GPT-3.5-turbo is \$0.06, while the cost for GPT-4 is \$1.8. The output of the PLM is around $1,500$ tokens. Thus, the total output tokens amount is approximately $30,000$. Since the cost of output tokens is twice that of input tokens, the cost is consistent. For `MiniRTS`, the total number of tokens needed per episode is approximately $1 \times 3,000 = 3,000$. The cost for GPT-3.5-turbo is \$0.003, while the cost for GPT-4 is \$0.09. The output of the PLM is also around $1,500$ tokens. Thus, the cost is also consistent with the input.

**Economic Cost for Training**    We conducted an average of ten evaluations in the online training process of different tasks to produce the learning curve represented in Figure 5 and 6. Considering the randomness of PLMs, each evaluation result is the average calculated outcome under 10 random seeds. For the `Overcooked-AI` task, an episode requires querying the PLM approximately 10 times on average, while for `MiniRTS`, a single query is needed at the beginning of the episode. Consequently, for the `Overcooked-AI` task, the total number of queries is 10 (evaluation times) times 10 (number of random seeds) times 10, which equals 1000 queries, and for MiniRTS, it is 10 times 10 times 1, which equals 100 queries. The number of tokens required for each query to the PLM is similar for both `Overcooked-AI` and `MiniRTS`, at around $3,000$ tokens. Therefore, for `Overcooked-AI`, the total number of tokens needed for training is approximately $1000 \times 3,000 = 3,000,000$. The cost for GPT-3.5-turbo is \$3, while the cost for GPT-4 is \$90. The output of the PLM is around $1,500$ tokens. Thus, the total output tokens amount is approximately $1,500,000$. Since the cost of output tokens is twice that of input tokens, the cost is consistent. For `MiniRTS`, the total number of tokens needed for training is approximately $100 \times 3,000 = 300,000$. The cost for GPT-3.5-turbo is \$0.3, while the cost for GPT-4 is \$9. The output of the PLM is also around $1,500$ tokens. Thus, the cost is also consistent with the input.

**Required Resources at Different Procedures**    Table 1 shows the resources required by SAMA different procedures.

---

[2]`https://github.com/HKUNLP/batch-prompting`.

Table 1: The resources needed at different procedures.

| Procedures | Required resources | MARL or not |
|---|---|---|
| Preprocessing | Papers and codes | no MARL |
| Semantically-aligned task decomposition | Rollout samples | no MARL |
| Language-grounded MARL | Papers, codes and rollout samples | MARL |
| Self-reflection | Rollout samples | no MARL |

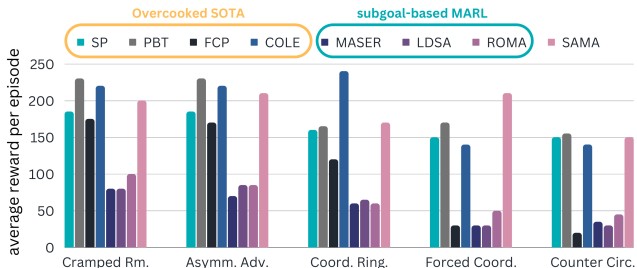

Figure 10: The mean self-play rewards of episodes over 400 timesteps under 10 random seeds. SAMA can approach or even exceed the performance of SOTA algorithm.

### E.1 OVERCOOKED

#### E.1.1 THE PROPOSED SAMA

At each timestep $t$, every agent $i$ acquires an $h \times w$ grid observation $o_t^i$ and yields a distribution over the action $\pi\left(a_t^i \mid o_t^i, z, g_t^i\right)$. The action is characterized by a one-hot matrix mirroring the observation's dimensions, signifying the agent's targeted relocation. It is imperative to note that, besides relocation, Overcooked agents' actions also enable manipulation of environmental objects, such as onions and pots. We augment the original Overcooked map to reduce the action space, thus facilitating the agent's co-location with the object. Consequently, the action of operating is seamlessly converted into a relocation action. We shall excise the subscript $t$ for expediency.

Possessing an encompassing linguistic knowledge, the agent initially aligns this language with environmental observations, employing a pre-existing language grounding module to engender a language-based representation $X = \text{Ground}(o, z, g) \in \mathbb{R}^{h \times w \times d}$. This conduit captures the intricate connections among the subgoal, the task manual, and the observation. Such a representation is harnessed to generate the policy. It is essential to highlight that SAMA remains congruent with any language grounding module. Pragmatically, we construct our framework upon the foundation of EMMA[3] (Hanjie et al., 2021), embracing their grounding modules - the multi-modal attention.

The architecture of EMMA is illustrated in Figure 11. The EMMA model encompasses three components: Text Encoder, Entity Representation Generator, and Action Module (Hanjie et al., 2021; Ding et al., 2023). The input for the Text Encoder constitutes a $h \times w$ grid observation imbued with entity descriptions. EMMA encodes each description utilizing a pretrained and fixed BERT-base model. Subsequently, the key and value embeddings are extracted from the encoder. Within the Entity Representation Generator, EMMA embeds each entity's symbol into a query embedding, attending to the descriptions accompanied by their respective key and value embeddings. For each entity $e$ in the observation, EMMA situates its representation $x_e$ within a tensor $X \in \mathbb{R}^{h \times w \times d}$, precisely aligning with the entity's position in the observation to preserve comprehensive spatial information. The agent's representation is merely a learned embedding of dimension $d$. In the Action Module, to furnish temporal information that facilitates grounding movement dynamics, EMMA concatenates the outputs of the representation generator from the most recent observations, procuring a tensor $X' \in \mathbb{R}^{h \times w \times d}$. EMMA conducts a 2D convolution on $X'$ over the $h, w$ dimensions, deriving a distribution over the actions with an identical shape to the grid observation. For further information regarding EMMA, the reader is directed to the original paper (Hanjie et al., 2021).

---

[3]https://github.com/ahjwang/messenger-emma, MIT License.

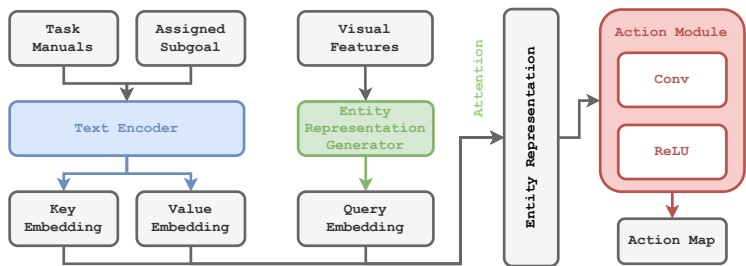

Figure 11: The three components of the EMMA model: Text Encoder, Entity Representation Generator, and Action Module. The Text Encoder uses a pretrained BERT-base model to encode entity descriptions within a grid observation, which are then processed by the Entity Representation Generator to position entity representations in a tensor. Finally, the Action Module derives action distributions based on the processed entity representation tensor.

The EMMA model exhibits end-to-end differentiability and is trained using the MAPPO[4] (Yu et al., 2022) algorithm with $\gamma = 0.99, \epsilon = 0.01$, along with the Adam optimizer, featuring a learning rate of $\alpha = 5 \times 10^5$. Each episode is delimited to encompass 400 timesteps. The validation games are employed to preserve the model parameters with the highest reward per episode throughout the training, subsequently utilizing these parameters to evaluate the models on the test games under 10 random seeds.

### E.1.2 BASELINES

We train and evaluate the self-play and PBT approaches utilizing the Human-Aware Reinforcement Learning repository[5] (Carroll et al., 2019), implementing PPO (Schulman et al., 2017) as the RL algorithm. We train FCP in accordance with the FCP publication (Strouse et al., 2021) and employ COLE's implementation (referenced below). The COLE agent, aligned with the manuscript (Li et al., 2023b), adapts a population size of 5 and incorporates the original implementation[6]. Furthermore, the MASER[7], LDSA[8], and ROMA[9] agents maintain the utilization of their original implementations.

### E.2 MINIRTS

Except for ROMA, all policies commence from a warm-start employing the RED pretrained model[10].

### E.2.1 SAMA AND RED IMPLEMENTATIONS

The training approach for SAMA and RED remains identical, with SAMA's required natural language instructions (i.e., subgoals) adhering to the oracle script furnished by (Xu et al., 2022). The sole distinction arises during testing, wherein SAMA receives instructions via PLM prompting, whereas RED acquires instructions through Oracle scripts. Given our adaptation of MiniRTS to accommodate multi-agent algorithms, we conformed to the RED implementation and training methodology to refine the pretrained language-grounded MARL policy.

The policy architecture parallels that of Hu et al. (2019), albeit with the incorporation of an ancillary value head as the critic, as proposed by (Xu et al., 2022). We refine the policy training employing a concurrent MAPPO process. Synchronous parallel workers gather transitions $(s, a, r, s')$, with the amassed data subdivided into four distinct batches to execute MAPPO training, utilizing a discount

---

[4] https://github.com/marlbenchmark/on-policy, MIT License.

[5] https://github.com/HumanCompatibleAI/human_aware_rl/tree/neurips2019.

[6] https://sites.google.com/view/cole-2023

[7] https://github.com/Jiwonjeon9603/MASER

[8] https://openreview.net/attachment?id=J5e13zmpj-Z&name=supplementary_material, Apache-2.0 License.

[9] https://github.com/TonghanWang/ROMA, Apache-2.0 License.

[10] https://drive.google.com/file/d/1dEjG9IOCwunUjVUqOOdDxaYr0lmg9Li1/view?usp=sharing, MIT License.

factor of $0.999$. Generalized Advantage Estimation (GAE) advantage is applied to each data point, culminating in $100$ training epochs per iteration. The Behaviour Cloning (BC) training procedure bears resemblance to Hu et al. (2019), introducing an additional $0/1$ action to govern the controllable unit's behavior. These supplementary actions derive from prevalent action type commonalities.

We employ the `Adam` optimizer, assigning discrete optimizers for MARL and BC training endeavors. A total of $15$ training iterations, equivalent to $1,500$ MAPPO epochs, is employed. We establish $\beta = (0.9, 0.999)$ for both phases, accompanied by a constant learning rate of $2 \times 10^{-4}$ for BC training. The MARL training's learning rate is adapted across $1,500$ MAPPO epochs, commencing linearly from $0$ to $5 \times 10^{-5}$ within the initial $300$ epochs and progressively diminishing from $5 \times 10^{-5}$ to $0$. For further elucidation, please consult the original paper (Xu et al., 2022).

### E.2.2 ROMA IMPLEMENTATION

We develop ROMA agents employing the original implementation[11]. To accommodate the original implementation within the state and action space of the `MiniRTS`, we supplanted its network structure with the policy architecture employed by SAMA and RED.

## F PROMPT ENGINEERING

### F.1 TASK MANUAL GENERATION

Upon paragraphing the LaTeX document, we employ question sets $Q_{rel}$ and $Q_{game}$, as utilized in (Wu et al., 2023), to filter for relevance and extract pivotal information respectively, as shown in Listing 1 and Listing 2.

Listing 1: $Q_{rel}$

```
1. Would this paragarph help me succeed in this game?
2. Does this paragraph contain information on the game mechanics,
   ↪ or game strategies?
```

Listing 2: $Q_{game}$

```
1. Write all information helpful for the game in a numbered list.
2. In plain text. List all objects I need to interact/avoid to
   ↪ survive in the game. Use "I would like to X object Y" in
   ↪ each step. Replace Y by the actual object, X by the actual
   ↪ interaction.
3. Write all game objectives numbered list. For each objective,
   ↪ list its requirements.
4. Write all actions as a numbered list. For each action, list its
   ↪  requirements.
```

The generated task manuals with GPT-4 are shown in Listing 3 for `Overcooked` and Listing 4 for `MiniRTS`. We use four papers (Carroll et al., 2019; Charakorn et al., 2020; Knott et al., 2021; Li et al., 2023b) for `Overcooked`, and another two papers (Hu et al., 2019; Xu et al., 2022) for `MiniRTS`.

Listing 3: Task Manual for `Overcooked`

```
You are a player playing a game.
In this game, two chefs in a restaurant serve onion soup.
This game aims to serve as many orders as possible before the end
   ↪ of time.
To complete an order, the following steps need to be completed in
   ↪ order:
1. Take three onions from the onion storage room and put them on
   ↪ the crafting table;
```

---

[11]https://github.com/TonghanWang/ROMA, Apache-2.0 License.

```
2. It takes 20 seconds for the crafting table to automatically
   ↪ make an onion soup;
3. Take a plate from the sideboard to the crafting table to serve
   ↪ the onion soup;
4. Take the onion soup from the crafting table, put it on the
   ↪ serving counter, and an order is completed now.

The above steps require two chefs to cooperate to complete.
In addition to the above steps, there are the following rules in
   ↪ the game:
1. Each chef can only transport one onion at a time;
2. The crafting table starts working if and only if there are
   ↪ three onions on the crafting table;
3. The finished onion soup must be served on a plate taken from
   ↪ the sideboard before being brought to the counter.

A brief description of the game map is as follows:
two chefs are in separate rooms.
The room on the left contains only the onion storage room and the
   ↪ sideboard, while the room on the right contains two crafting
   ↪  tables and a serving counter.
There are two crafting tables, one above the right room and one to
   ↪  the right of the right room.
There is a shared bar between the two rooms, which can be used to
   ↪ pass items.
```

Listing 4: Task Manual for `MiniRTS`

```
MiniRTS is a game environment designed to encompass the
   ↪ fundamental complexities of real-time strategy games.
Consisting of two playing sides- a user-run unit governed by a pre
   ↪ -set policy and an in-built AI enemy, players control their
   ↪ units to gather resources, manage constructions, and defeat
   ↪ enemy units by destroying their base or eliminating all
   ↪ opposition on the battlefield.

MiniRTS contains 6 unique building types, each fulfilling a
   ↪ specific role in-game. Using allocated resources, the
   ↪ PEASANT unit type can construct any building type in any
   ↪ available area on the map.
The constructed buildings can then be utilized to construct
   ↪ various unit types. While most building types generate up to
   ↪  one distinct unit type, the WORKSHOP can produce 3
   ↪ different unit types.
For further specifics, please refer to the following table:

Building name | Description
TOWN HALL | The main building of the game, it allows a player to
   ↪ train PEASANTs and serves as a storage for mined resources.
BARRACK | Produces SPEARMEN.
BLACKSMITH | Produces SWORDMEN.
STABLE | Produces CAVALRY.
WORKSHOP | Produces CATAPULT, DRAGON and ARCHER. The only building
   ↪  that can produce multiple unit types.
GUARD TOWER | A building that can attack enemies, but cannot move.

A total of 7 unit types form a rock-paper-scissors attacking
   ↪ dynamics.
For further specifics, please refer to the following table:
```

```
Unit name | Description
PEASANT | Gathers minerals and constructs buildings, not good at
    ↪ fighting.
SPEARMAN | Effective against cavalry.
SWORDMAN | Effective against spearmen.
CAVALRY | Effective against swordmen.
DRAGON | Can fly over obstacles, can only be attacked by archers
    ↪ and towers.
ARCHER | Great counter unit against dragons.
CATAPULT | Easily demolishes buildings.

Kindly note that the maximum number of each building cannot exceed
    ↪ 1 and likewise the maximum number of each type of unit
    ↪ cannot exceed 6.
```

## F.2 STATE AND ACTION TRANSLATION

The prompt for state and action translation is shown in Listing 5.

Listing 5: Prompt for State and Action Translation

```
At present, you serve as a translator of environmental state,
    ↪ agent states, and actions. It is incumbent upon you to
    ↪ extract the explicit semantics represented by the structured
    ↪  information I provide, guided by the task manual and
    ↪ drawing from the code repository. Synthesize the acquired
    ↪ information into one sentence, articulating it in refined
    ↪ natural language.
```

## F.3 INTERACTIVE OBJECTS EXTRACTION

The prompt for state and action translation is shown in Listing 6.

Listing 6: Prompt for Interactive Objects Extraction

```
Extract all pertinent, interactive objects from the task manual,
    ↪ which are instrumental in accomplishing the task. Refrain
    ↪ from including extraneous items, and present the extracted
    ↪ objects in a comma-separated list.
```

## F.4 OFFLINE SUBGOAL GENERATION

The prompt for diverse goal generation is shown in Listing 7.

Listing 7: Prompt for Diverse Goal Generation

```
At present, you serve as a generator of environmental state and
    ↪ agent states. Kindly generate additional, diverse states in
    ↪ accordance with the structured status information I have
    ↪ inputted, following the task manual and code repository. The
    ↪  produced states must adhere to the subsequent constraints:
    ↪ 1. Conformity with the input format; 2. Ensuring state
    ↪ legality.
```

## F.5 REWARD DESIGN

The prompt for reward design is shown in Listing 8.

Listing 8: Prompt for Reward Design

```
At present, you function as a Python method generator, formulating
 ↪  custom Python methods grounded in given objectives, the
 ↪  task manual, and a code repository. These functions
 ↪  necessitate structured environmental states as input-such as
 ↪   [some examples]-and ascertaining whether the input has
 ↪  fulfilled the stated objectives; they must yield "1" if
 ↪  achieved and "0" if unmet. Crucially, no values other than
 ↪  "0" or "1" should be generated.
```

## F.6   GOAL GENERATION, DECOMPOSITION AND SUBGOAL ASSIGNMENT

### F.6.1   OVERCOOKED

Figure 12 and Figure 13 are illstrate the colored prompts for better understanding.

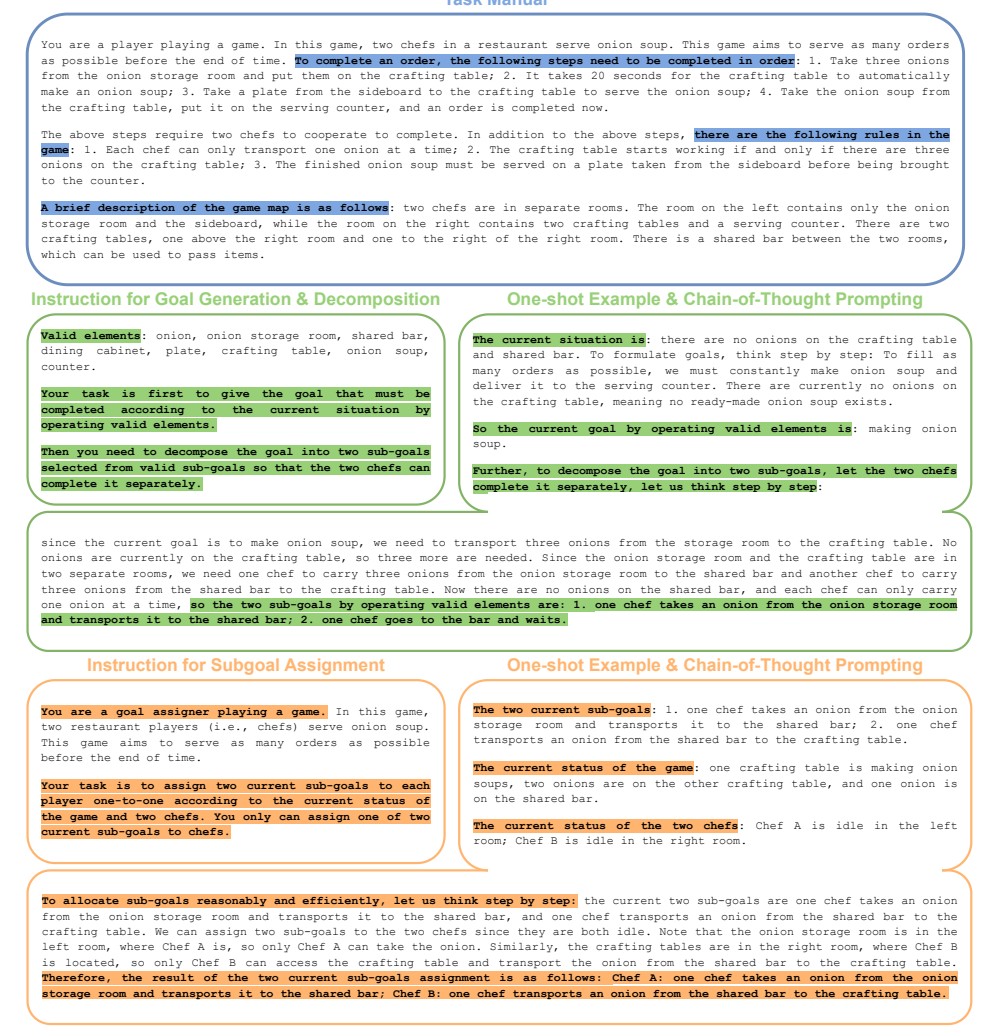

Figure 12: The prompt designed for the **Forced Coordination** layout in Overcooked.

**Overview**   Contemplating the lucidity of the material, we refrain from exhibiting the prompts germane to each layout within the Overcooked. The subsequent excerpt furnishes an illustration of a prompt pertaining to **Forced Coordination**. Divergent designs adhere to an analogous blueprint,

**Instruction for Self-reflection**

**As a decision inspector**, you are playing a game where two restaurant players (chefs) are serving onion soup, with the intention of fulfilling as many orders as possible before the time runs out.

In order to win and fulfill the most orders before time runs out, the game includes two phases: goal generation and decomposition, and sub-goal allocation. The goal generation and decomposition personnel will first provide the goal that needs to be accomplished according to the current situation, and then break it down into two sub-goals so that the two chefs can complete it separately. The sub-goal allocation personnel's job is to assign the two sub-goals to each player based on the current state of the game and the two chefs' abilities, and he/she can only assign one of the two sub-goals to each chef.

**Valid phase:** goal generation and decomposition, subgoal assignment.

**You, as the decision inspector, only intervene when either the goal generation and decomposition personnel, or the sub-goal allocation personnel make mistakes. Your job consists of two parts.** Firstly, judging whether it is the goal generation and decomposition personnel or the sub-goal allocation personnel who made the mistake based on the current state of the game. You can only choose from the valid phases.

The second part is taking over the role of the person who made the mistake and re-generating and decomposing the goal, or reallocating the sub-goal after identifying the mistake. **Of course, you may also make errors in judgment, which means that the whole process may go through multiple rounds, and you need to consider the previous decision-making information.**

**One-shot Example and Chain-of-Thought Prompting for Step 1**

**The current situation is:** there are no onions on the crafting table and shared bar. **The generated goal from the goal generation and decomposition personnel is:** making onion soup. **The decomposed two sub-goals from the goal generation and decomposition personnel are:** one chef takes an onion from the onion storage room and transports it to the shared bar; one chef transports an onion from the shared bar to the crafting table. **The current status of the game:** there are no onions on the crafting table and shared bar. **The current status of the two chefs:** Chef A is idle in the left room; Chef B is idle in the right room. **The result of the two current sub-goals assignment decided by the sub-goal allocation personnel is as follows:** Chef A: one chef takes an onion from the onion storage room and transports it to the shared bar; Chef B: one chef transports an onion from the shared bar to the crafting table. **The failed decision inspection history:** None.

**To determine whether any errors occurred during the phase of goal generation and decomposition, or during the subgoal allocation stage, let us think step by step.** Firstly, we will concentrate on the goal generation and decomposition phase. Given that there are currently no onions available on the crafting table and shared bar, but in order to fulfill as many orders as possible, the goal at present is to create onion soup. Hence, there seems to be no issue with goal generation. In the next step, we must subdivide the above goal into two subgoals. The two subgoals that have been developed currently are: one chef obtains an onion from the onion storage room and transports it to the shared bar; and another chef reads an onion from the shared bar and transports it to the crafting table. Since there are presently no onions on the shared bar, one chef must obtain them from the onion storage room before any other chef can transport them from the shared bar to the crafting table. Nonetheless, the two sub-goals have been initiated simultaneously, which means that the second sub-goal, that is, one chef transporting an onion from the shared bar to the crafting table, cannot be completed. Based on the above analysis, it appears that the problematic stage is: **goal generation and decomposition.**

**One-shot Example and Chain-of-Thought Prompting for Step 2**

**Previous failed goals:** None.

**Previous failed subgoals:** one chef takes an onion from the onion storage room and transports it to the shared bar; one chef transports an onion from the shared bar to the crafting table.

**For more reasonable goal generation and decomposition, the goals you generate or the subgoals you decompose should no longer be consistent with previously failed goals or subgoals.**

[Similar with instructions for goal generation and decomposition]

**Previous failed subgoal assignments:** Chef A: one chef takes an onion from the onion storage room and transports it to the shared bar; Chef B: one chef transports an onion from the shared bar to the crafting table.

**For more reasonable subgoal assignment, the assignment you make should no longer be consistent with previously failed subgoal assignments.**

[Similar with instructions for subgoal assignment]

Figure 13: The prompt of self-reflection designed for the **Forced Coordination** layout in `Overcooked`. The prompts are composed of task manual in Figure 12, instruction and one-shot CoT example.

conceding the substitution of content correlated to the layouts, encompassing layout delineation, and the few-shot CoT examples, among others.

Listing 9: Instruction for Goal Generation and Decomposition

```
Valid elements: onion, onion storage room, shared bar, dining
    ↪ cabinet, plate, crafting table, onion soup, counter.

Your task is first to give the goal that must be completed
    ↪ according to the current situation by operating valid
    ↪ elements.

Then you need to decompose the goal into two sub-goals selected
    ↪ from valid sub-goals so that the two chefs can complete it
    ↪ separately.
```

Listing 10: One-shot Example and Chain-of-Thought Prompting for Goal Generation and Decomposition

```
The current situation is: there are no onions on the crafting
    ↪ table and shared bar.
To formulate goals, think step by step:
```

```
To fill as many orders as possible, we must constantly make onion
    ↪ soup and deliver it to the serving counter.
There are currently no onions on the crafting table, meaning no
    ↪ ready-made onion soup exists.

So the current goal by operating valid elements is: making onion
    ↪ soup.

Further, to decompose the goal into two sub-goals, let the two
    ↪ chefs complete it separately, let us think step by step:
since the current goal is to make onion soup, we need to transport
    ↪  three onions from the storage room to the crafting table.
No onions are currently on the crafting table, so three more are
    ↪ needed.
Since the onion storage room and the crafting table are in two
    ↪ separate rooms, we need one chef to carry three onions from
    ↪ the onion storage room to the shared bar and another chef to
    ↪  carry three onions from the shared bar to the crafting
    ↪ table.
Now there are no onions on the shared bar, and each chef can only
    ↪ carry one onion at a time, so the two sub-goals by operating
    ↪  valid elements are:
1. one chef takes an onion from the onion storage room and
    ↪ transports it to the shared bar;
2. one chef goes to the bar and waits.
```

Listing 11: Instruction for Subgoal Allocation

```
You are a goal assigner playing a game.
In this game, two restaurant players (i.e., chefs) serve onion
    ↪ soup.
This game aims to serve as many orders as possible before the end
    ↪ of time.

Your task is to assign two current sub-goals to each player one-to
    ↪ -one according to the current status of the game and two
    ↪ chefs.
You only can assign one of two current sub-goals to chefs.
```

Listing 12: One-shot Example and Chain-of-Thought Prompting for Subgoal Allocation

```
The two current sub-goals:
1. one chef takes an onion from the onion storage room and
    ↪ transports it to the shared bar;
2. one chef transports an onion from the shared bar to the
    ↪ crafting table.

The current status of the game:
one crafting table is making onion soups, two onions are on the
    ↪ other crafting table, and one onion is on the shared bar.

The current status of the two chefs: Chef A is idle in the left
    ↪ room; Chef B is idle in the right room.

To allocate sub-goals reasonably and efficiently, let us think
    ↪ step by step:
the current two sub-goals are one chef takes an onion from the
    ↪ onion storage room and transports it to the shared bar, and
```

```
      ↪ one chef transports an onion from the shared bar to the
      ↪ crafting table.
We can assign two sub-goals to the two chefs since they are both
    ↪ idle.
Note that the onion storage room is in the left room, where Chef A
    ↪  is, so only Chef A can take the onion.
Similarly, the crafting tables are in the right room, where Chef B
    ↪  is located, so only Chef B can access the crafting table
    ↪ and transport the onion from the shared bar to the crafting
    ↪ table.
Therefore, the result of the two current sub-goals assignment is
    ↪ as follows:
Chef A: one chef takes an onion from the onion storage room and
    ↪ transports it to the shared bar;
Chef B: one chef transports an onion from the shared bar to the
    ↪ crafting table.
```

Listing 13: Instruction for Self-reflection

```
As a decision inspector, you are playing a game where two
    ↪ restaurant players (chefs) are serving onion soup, with the
    ↪ intention of fulfilling as many orders as possible before
    ↪ the time runs out.

In order to win and fulfill the most orders before time runs out,
    ↪ the game includes two phases: goal generation and
    ↪ decomposition, and sub-goal allocation.
The goal generation and decomposition personnel will first provide
    ↪  the goal that needs to be accomplished according to the
    ↪ current situation, and then break it down into two sub-goals
    ↪  so that the two chefs can complete it separately.
The sub-goal allocation personnel's job is to assign the two sub-
    ↪ goals to each player based on the current state of the game
    ↪ and the two chefs' abilities, and he/she can only assign one
    ↪  of the two sub-goals to each chef.

Valid phase: goal generation and decomposition, subgoal assignment
    ↪ .

You, as the decision inspector, only intervene when either the
    ↪ goal generation and decomposition personnel, or the sub-goal
    ↪  allocation personnel make mistakes.
Your job consists of two parts.
Firstly, judging whether it is the goal generation and
    ↪ decomposition personnel or the sub-goal allocation personnel
    ↪  who made the mistake based on the current state of the game.
    ↪
You can only choose from the valid phases.

The second part is taking over the role of the person who made the
    ↪  mistake and re-generating and decomposing the goal, or
    ↪ reallocating the sub-goal after identifying the mistake.
Of course, you may also make errors in judgment, which means that
    ↪ the whole process may go through multiple rounds, and you
    ↪ need to consider the previous decision-making information.
```

Listing 14: One-shot Example and Chain-of-Thought Prompting for Step 1

```
The current situation is: there are no onions on the crafting
    ↪ table and shared bar.
The generated goal from the goal generation and decomposition
    ↪ personnel is: making onion soup.
The decomposed two sub-goals from the goal generation and
    ↪ decomposition personnel are: one chef takes an onion from
    ↪ the onion storage room and transports it to the shared bar;
    ↪ one chef transports an onion from the shared bar to the
    ↪ crafting table.
The current status of the game: there are no onions on the
    ↪ crafting table and shared bar.
The current status of the two chefs: Chef A is idle in the left
    ↪ room; Chef B is idle in the right room.
The result of the two current sub-goals assignment decided by the
    ↪ sub-goal allocation personnel is as follows: Chef A: one
    ↪ chef takes an onion from the onion storage room and
    ↪ transports it to the shared bar; Chef B: one chef transports
    ↪  an onion from the shared bar to the crafting table.
The failed decision inspection history: None.

To determine whether any errors occurred during the phase of goal
    ↪ generation and decomposition, or during the subgoal
    ↪ allocation stage, let us think step by step.
Firstly, we will concentrate on the goal generation and
    ↪ decomposition phase.
Given that there are currently no onions available on the crafting
    ↪  table and shared bar, but in order to fulfill as many
    ↪ orders as possible, the goal at present is to create onion
    ↪ soup.
Hence, there seems to be no issue with goal generation.
In the next step, we must subdivide the above goal into two sub-
    ↪ goals.
The two sub-goals that have been developed currently are: one chef
    ↪  obtains an onion from the onion storage room and transports
    ↪  it to the shared bar; and another chef reads an onion from
    ↪ the shared bar and transports it to the crafting table.
Since there are presently no onions on the shared bar, one chef
    ↪ must obtain them from the onion storage room before any
    ↪ other chef can transport them from the shared bar to the
    ↪ crafting table.
Nonetheless, the two sub-goals have been initiated simultaneously,
    ↪  which means that the second sub-goal, that is, one chef
    ↪ transporting an onion from the shared bar to the crafting
    ↪ table, cannot be completed.
Based on the above analysis, it appears that the problematic stage
    ↪  is: goal generation and decomposition.
```

Listing 15: One-shot Example and Chain-of-Thought Prompting for Step 2

```
[For Goal Generation and Decomposition Phase]

Previous failed goals: None.

Previous failed sub-goals: one chef takes an onion from the onion
    ↪ storage room and transports it to the shared bar; one chef
    ↪ transports an onion from the shared bar to the crafting
    ↪ table.
```

```
For more reasonable goal generation and decomposition, the goals I
    ↪ generate or the sub-goals I decompose should no longer be
    ↪ consistent with previously failed goals or sub-goals.

[Similar with instructions for goal generation and decomposition]

---

[For Subgoal Allocation Phase]

Previous failed subgoal assignments: Chef A: one chef takes an
    ↪ onion from the onion storage room and transports it to the
    ↪ shared bar; Chef B: one chef transports an onion from the
    ↪ shared bar to the crafting table.

For more reasonable subgoal assignment, the assignment I make
    ↪ should no longer be consistent with previously failed
    ↪ subgoal assignments.

[Similar with instructions for subgoal allocation]
```

### F.6.2 MINIRTS

MiniRTS furnishes an array of built-in script AIs, and we choose the medium-level AI as the opponent to facilitate expeditious prompt design for PLMs. Initially, this script dispatches all three available peasants to mine the nearest resource unit. It then randomly selects one of the seven army unit types, determining an army size $n$ ranging between 3 and 7. A building unit corresponding to the selected army is constructed, followed by training $n$ units of the elected army category for deployment in attacks. The script perpetuates army unit training, maintaining an army size of $n$.

Given that the built-in script AI constructs only two army unit types per game, we establish an oracle prompt design strategy following the ground truth of enemy units and the attack graph (Figure 9), similar with (Xu et al., 2022). In the game's nascent stages, the PLM instructs the agent to construct suitable army units progressively according to following prompts.

Listing 16: Instructions for Goal Decomposition and Allocation

```
As a player of a MiniRTS game, your objective is to strategically
    ↪ construct units that can effectively attack and defeat the
    ↪ AI.
To build a unit, you must follow specific procedures:

1. Secure adequate resources.
  If the current resources are insufficient, then resources must
      ↪ be collected.
2. Generate the unit if sufficient resources are available and
    ↪ there is a building that can produce the unit.
  If there isn't such a building, construct it first, then
      ↪ proceed to generate the unit.

Your responsibility is to focus on constructing the appropriate
    ↪ units, without concerning yourself with how to operate them
    ↪ for an attack on the AI. To defeat the AI as swiftly as
    ↪ possible.

Valid sub-goals: mine with all idle peasant, build one [building
    ↪ name], build [number] [unit name]
```

```
Your task is to select the most viable two sub-goals, relative to
    ↪ the current situation, from valid sub-goals so that the two
    ↪ agents can complete it separately.
You may substitute the [building name] field in valid sub-goals
    ↪ with the corresponding building name as it appears in the
    ↪ game, replace [unit name] with the name of the unit in the
    ↪ game, and [number] with a numeric value representing the
    ↪ desired quantity.
Kindly note that the maximum number of each building cannot exceed
    ↪ 1 and likewise the maximum number of each type of unit
    ↪ cannot exceed 6.
```

Listing 17: Few-shot Chain-of-Thought Examples for Goal Decomposition and Allocation

```
The current situation is: The enemy has 5 swordman and 7 catapult;
    ↪ our side has 3 spearman, 1 barrack building, 2 peasant,
    ↪ sufficient resources. . .

To formulate sub-goals, let us think step by step:
To overcome the enemy's weaponry, our forces must produce superior
    ↪ arms that are based on attacking dynamics.
Firstly, the opposition possesses 5 swordsmen, which can be bested
    ↪ by cavalry units.
Unfortunately, we do not currently have this unit available, so it
    ↪ is imperative that we construct the stable building in
    ↪ order to produce cavalry.
Fortunately, we presently have ample resources to build the stable
    ↪ , thus it is imperative that we prioritize this subgoal.
Additionally, the enemy's artillery includes 7 catapults, which
    ↪ can be countered by all other types of units.
With 3 spearman units available to us, we are equipped to restrain
    ↪ enemy catapults.
However, we require additional forces to match the enemy's 7
    ↪ artillery units.
To achieve combat parity, we must procure 7-3=4 additional
    ↪ spearman units.
Luckily, we already have a barracks building to generate spearman
    ↪ and with our current resources, we can readily construct the
    ↪ required 4 units.

So the current two sub-goals select from valid sub-goals are:
    ↪ build one stable, and build 4 spearman.

---

The current situation is: The enemy has 7 catapult; our side has 3
    ↪ spearman, 1 barrack building, 2 peasant, sufficient
    ↪ resources. . .

To formulate sub-goals, let us think step by step:
To overcome the enemy's weaponry, our forces must produce superior
    ↪ arms that are based on attacking dynamics.
Firstly, the enemy's artillery includes 7 catapults, which can be
    ↪ countered by all other types of units.
With 3 spearman units available to us, we are equipped to restrain
    ↪ enemy catapults.
However, we require additional forces to match the enemy's 7
    ↪ artillery units.
```

```
To achieve combat parity, we must procure 7-3=4 additional
   ↪ spearman units.
Luckily, we already have a barracks building to generate spearman
   ↪ and with our current resources, we can readily construct the
   ↪  required 4 units.
The aforementioned subgoal can be achieved by a single agent, and
   ↪ there are no other enemy units to be vanquished.
To avoid leaving another agent idle, our second subgoal could be
   ↪ deemed more extensive in scope.
To swiftly counter the enemy's strategy, ample resources must be
   ↪ at our disposal.
As it stands, our force consists of two peasant, hence the second
   ↪ sub-goal entails having these peasant gather resources.

So the current two sub-goals select from valid sub-goals are:
   ↪ build 4 spearman, and mine with all idle peasant.
```

Listing 18: Instructions for Self-Reflection

```
As an arbiter of decisions, you partake in a competition that
   ↪ comprises of two opposing factions – a user-directed
   ↪ regiment adherent to predetermined principles and an AI
   ↪ adversary.
Through this competition, participants oversee the resource
   ↪ gathering, facility maintenance, and obliteration of
   ↪ adversary forces to either demolish their stronghold or
   ↪ vanquish all opposition on the battleground.

To emerge victorious and conquer all opposition, the game must
   ↪ incorporate personnel responsible for goal decomposition.
These individuals will select the two most viable sub-goals, based
   ↪  on the current situation, from a pool of valid sub-goals
   ↪ for the two agents to accomplish separately.

As the decision inspector, you only intervene when the goal
   ↪ decomposition personnel make mistakes.
Your role is to assume responsibility for the goal decomposition.
However, even you may err in judgement, necessitating multiple
   ↪ rounds of refinement while considering previous goal
   ↪ decomposition information.
```

Listing 19: Few-shot Examples and Chain-of-Thought Prompting for Self-Reflection

```
The current situation is: The enemy has 5 swordman and 7 catapult;
   ↪  our side has 3 spearman, 1 barrack building, 2 peasant,
   ↪ sufficient resources. . .
The generated sub-goals goal from the goal decomposition personnel
   ↪  is: build one barrack, and build 4 spearman.
The failed decision inspection history: None.

To determine what errors occurred during the phase of goal
   ↪ decomposition, let us think step by step:
According to the task manual, only one building of each type is
   ↪ allowed in the game.
As we already have a barracks on our side, it is not possible to
   ↪ construct another one.
Therefore, the current subgoal instruction of ``build one barrack
   ↪ '' is incorrect and must be regenerated.
```

```
[Continue appending the following]

Previous failed sub-goals: build one barrack

For more reasonable goal decomposition, the sub-goals I generate
    ↪ should no longer be consistent with previously failed sub-
    ↪ goals.

[The following is similar with instructions for goal decomposition
    ↪  and allocation]
```

## G  MISSING PLM OUTPUTS

This section presents selected intermediate results of randomly chosen SAMA instances in two benchmark tasks.

Listing 20: Some Translated (Environment and Agent) States in `Overcooked`

```
1. Both crafting tables are making onion soups, and there is one
    ↪ plate on the shared bar.
2. There are three onions on one crafting table, one onion on the
    ↪ shared bar, and one plate on the sideboard.
3. One onion is on the shared bar, and there are no onions or
    ↪ plates on either of the crafting tables or the serving
    ↪ counter.
4. Two completed onion soups are on one crafting table, and three
    ↪ onions are on the other crafting table.
5. One chef is holding an onion, and the other chef is holding a
    ↪ plate, with one onion on one crafting table and two onions
    ↪ on the other crafting table.
6. Both crafting tables have three onions each, and both chefs are
    ↪  holding a plate.
7. One crafted onion soup is on the shared bar, waiting for a chef
    ↪  to serve it on a plate, and the other crafting table has
    ↪ two onions.
8. Two onions are on one crafting table, one onion is on the
    ↪ shared bar, and a completed onion soup is on a plate on the
    ↪ serving counter.
9. One chef is holding an onion, one onion is on the shared bar,
    ↪ and both crafting tables have two onions each.
10. A crafted onion soup is on a plate on one crafting table,
    ↪ while the other crafting table is making onion soup with
    ↪ three onions on it.
```

Listing 21: Some Translated (Environment and Agent) States in `MiniRTS`

```
1. The enemy has 4 swordman and 3 cavalry; our side has 2 archer,
    ↪ 1 workshop building, 1 peasant, limited resources.
2. The enemy has 2 dragon and 5 spearman; our side has 3 archer, 1
    ↪  stable building, 2 peasant, sufficient resources.
3. The enemy has 6 spearmen and 1 dragon; our side has 4 cavalry,
    ↪ 1 blacksmith building, 2 peasant, limited resources.
4. The enemy has 3 dragon and 4 archer; our side has 5 peasant, 1
    ↪ town hall building, limited resources.
5. The enemy has 2 cavalry and 3 catapult; our side has 1 guard
    ↪ tower, 1 stable building, 3 peasant, sufficient resources.
6. The enemy has 3 archer and 4 swordman; our side has 4 spearman,
    ↪  1 blacksmith building, 1 peasant, limited resources.
```

```
7. The enemy has 1 dragon, 4 spearman, and 2 catapult; our side
   ↪ has 2 archer, 1 workshop building, 2 peasant, sufficient
   ↪ resources.
8. The enemy has 4 cavalry and 3 swordman; our side has 4 spearmen
   ↪ , 1 barrack building, 1 peasant, limited resources.
9. The enemy has 2 archer, 3 spearman, and 1 dragon; our side has
   ↪ 1 guard tower, 1 stable building, 1 peasant, sufficient
   ↪ resources.
10. The enemy has 5 spearmen and 2 swordman; our side has 3
    ↪ peasant, 1 town hall building, limited resources.
```

To monitor the training progress, we evaluate the agents at a very low frequency (Figure 5 is drawn according to the evaluation results). In light of the potential emergence of unaccounted environmental states during the evaluation process, periodic inclusion of the encountered states and corresponding PLM-generated subgoals to the dataset $\mathcal{D}_{subgoal}$ is undertaken. However, determining whether a goal or subgoal generated by a PLM exists in the offline dataset is not a simple task. One approach is to obtain the embeddings of the two goals to be compared using an open-source PLM and then assess their similarity (e.g., cosine similarity) to see if it exceeds a predefined threshold. Nevertheless, this method can be influenced by the performance of the open-source PLM and the chosen threshold. In our paper, we adopted an alternative method, using a query to GPT-4 to discern if a goal is classified as a new one. According to our experimental statistics, over 97% of the goals were covered during the offline generation phase.

Listing 22: Some Generated Goals in `Overcooked`

```
1. Create onion soup.
2. Deliver onion soup to counter.
3. Cook onion soup.
4. Complete onion soup making process.
5. Prepare onion soup.
6. Serve onion soup.
7. Make onion soup.
8. Transfer onion soup to counter.
9. Craft onion soup.
10. Serve cooked soup to customers.
```

Listing 23: Some Generated SubGoals in `Overcooked`

```
1. one chef takes an onion from the onion storage room and
   ↪ transports it to the shared bar;
2. one chef goes to the bar and waits.
3. Another chef picks up the onion from the shared bar and places
   ↪ it on the crafting table;
4. The second chef returns to the onion storage room to get
   ↪ another onion;
5. One chef takes a plate from the sideboard and transports it to
   ↪ the shared bar;
6. The other chef picks up the plate from the shared bar and sets
   ↪ it down next to the crafting table;
7. A chef takes the second onion from the onion storage room and
   ↪ brings it to the shared bar;
8. The other chef collects the second onion from the shared bar
   ↪ and places it on the crafting table;
9. The first chef retrieves the third onion from the onion storage
   ↪  room and delivers it to the shared bar;
10. The other chef obtains the third onion from the shared bar and
    ↪  places it on the crafting table, triggering the crafting
    ↪ process;
11. One chef goes to the shared bar and waits for the onion soup
    ↪ to finish;
```

12. The second chef picks up the plate from the sideboard and
    ↪ brings it to the shared bar;
13. The other chef collects the plate from the shared bar and
    ↪ places it near the crafting table;
14. One chef moves to the crafting table and waits for the onion
    ↪ soup to finish;
15. The first chef takes the completed onion soup from the
    ↪ crafting table and sets it on the plate;
16. A chef transports the plated onion soup from the crafting
    ↪ table to the serving counter;
17. The other chef retrieves another onion from the onion storage
    ↪ room and brings it to the shared bar;
18. One chef picks up the onion from the shared bar and places it
    ↪ on the second crafting table;
19. A chef takes an onion from the onion storage room and places
    ↪ it on the shared bar;
20. The second chef picks up the onion from the shared bar and
    ↪ places it on the second crafting table.

Listing 24: Some Generated Goals in `MiniRTS`

```
1. mine with all idle peasant
2. build one guard tower
3. build one blacksmith
4. build one barrack
5. build one stable
6. build one workshop
7. build 3 cavalry
8. build 2 catapult
9. build 5 archer
10. build 4 swordman
```

Listing 25: Reward Function Snippet for "*make onion soup*"

```python
if num_onions < 0:
    return 0
if num_onions > Recipe.MAX_NUM_INGREDIENTS:
    return 0
if cooking_tick >= 0 and num_onions == 0:
    return 0
if finished and num_onions == 0:
    return 0
if finished:
    return 1
```

Listing 26: Reward Function Snippet for "*place one onion on the crafting table*"

```python
if not ingredient.name in Recipe.ALL_INGREDIENTS:
    return 0
if self.is_full:
    return 0
else:
    return 1
```

Listing 27: Reward Function Snippet for "*build [#number] [unit_name]*"

```python
build_type = build_cmd['target_type']

for my_unit in current['my_units']:
```

```
    unit_type = my_unit['unit_type']
    if unit_type != build_type:
        continue

    new_unit = True
    unit_id = my_unit['unit_id']
    for prev_unit in previous['my_units']:
        if unit_id == prev_unit['unit_id']:
            new_unit = False
    if new_unit:
        return 1

return 0
```

Listing 28: Self-Reflection Process in `Overcooked-AI`

```
The current situation is: there are no onions on the crafting
    ↪ table and shared bar.
The generated goal from the goal generation and decomposition
    ↪ personnel is: making onion soup.
The decomposed two sub-goals from the goal generation and
    decomposition personnel are:  one chef takes an onion from the
    onion storage room and transports it to the shared bar; one
    chef transports an onion from the shared bar to the crafting
    table.
The current status of the game: there are no onions on the
    ↪ crafting table and shared bar.
The current status of the two chefs: Chef A is idle in the left
    ↪ room; Chef B is idle in the right room.
The result of the two current sub-goals assignment decided by the
    ↪ sub-goal allocation personnel is as follows: Chef A: one
    ↪ chef takes an onion from the onion storage room and
    ↪ transports it to the shared bar; Chef B: one chef transports
    ↪  an onion from the shared bar to the crafting table.
The failed decision inspection history: None.

To determine whether any errors occurred during the phase of goal
    ↪ generation and decomposition, or during the subgoal
    ↪ allocation stage, let us think step by step.
Firstly, we will concentrate on the goal generation and
    ↪ decomposition phase.
Given that there are currently no onions available on the crafting
    ↪  table and shared bar, but in order to fulfill as many
    ↪ orders as possible, the goal at present is to create onion
    ↪ soup.
Hence, there seems to be no issue with goal generation.
In the next step, we must subdivide the above goal into two sub-
    ↪ goals.
The two sub-goals that have been developed currently are: one chef
    ↪  obtains an onion from the onion storage room and transports
    ↪  it to the shared bar; and another chef reads an onion from
    ↪ the shared bar and transports it to the crafting table.
Since there are presently no onions on the shared bar, one chef
    ↪ must obtain them from the onion storage room before any
    ↪ other chef can transport them from the shared bar to the
    ↪ crafting table.
Nonetheless, the two sub-goals have been initiated simultaneously,
    ↪  which means that the second sub-goal, that is, one chef
    ↪ transporting an onion from the shared bar to the crafting
    ↪ table, cannot be completed.
```

```
Based on the above analysis, it appears that the problematic stage
    ↪ is:  goal generation and decomposition.

Previous failed sub-goals: one chef takes an onion from the onion
    ↪ storage room and transports it to the shared bar; one chef
    ↪ transports an onion from the shared bar to the crafting
    ↪ table.
For more reasonable goal generation and decomposition, the goals I
    ↪  generate or the sub-goals I decompose should no longer be
    ↪ consistent with previously failed goals or sub-goals.

The current situation is: there are no onions on the crafting
    ↪ table and shared bar.
To formulate goals, think step by step:
To fill as many orders as possible, we must constantly make onion
    ↪ soup and deliver it to the serving counter.
There are currently no onions on the crafting table, meaning no
    ↪ ready-made onion soup exists.

So the current goal by operating valid elements is: making onion
    ↪ soup.

Further, to decompose the goal into two sub-goals, let the two
    ↪ chefs complete it separately, let us think step by step:
since the current goal is to make onion soup, we need to transport
    ↪  three onions from the storage room to the crafting table.
No onions are currently on the crafting table, so three more are
    ↪ needed.
Since the onion storage room and the crafting table are in two
    ↪ separate rooms, we need one chef to carry three onions from
    ↪ the onion storage room to the shared bar and another chef to
    ↪  carry three onions from the shared bar to the crafting
    ↪ table.
Now there are no onions on the shared bar, and each chef can only
    ↪ carry one onion at a time, so the two sub-goals by operating
    valid elements are:1.  one chef takes an onion from the onion
    storage room and transports it to the shared bar;2.  one chef
    goes to the bar and waits.
```

## H  MISSING ABLATION STUDIES

**Is the learned policy grounded?** Observe that Figure 4 (Right) and Figure 6 (Right) have already furnished ample evidence supporting the command-following performance of the SAMA policy, as the attained rewards and win rates proximate to SOTA methods can solely be accomplished by adhering to the Oracle commands. Consequently, we introduce an ancillary metric, a modification of the criterion postulated by RED: intrinsically, as a command-following policy, its reward or win rate ought to diminish if it receives "inferior" commands; conversely, if the reward or win rate remains stable, the policy must inadequately adhere to the commands. Thus, we interpolate between the PLM-generated subgoal and the wholly random subgoal, subsequently evaluating disparate policies' reward and win rates with varying proportions of random commands. The outcomes are delineated in Figure 14. The findings corroborate our preliminary analysis.

**Does self-reflection prove advantageous?** To ascertain whether the self-reflection stimulates PLMs to rectify logical or factual inaccuracies they commit during semantically aligned goal generation, decomposition, and allocation, we examined the efficacy of SAMA incorporating diverse quantities of self-reflection trials. Gleaning from Figure 14 (Right), we garner two notable observations: (1) The self-reflection mechanism engenders a marked enhancement in performance; (2) In most instances, exceeding a single self-reflection trial elicits a conspicuous elevation in performance.

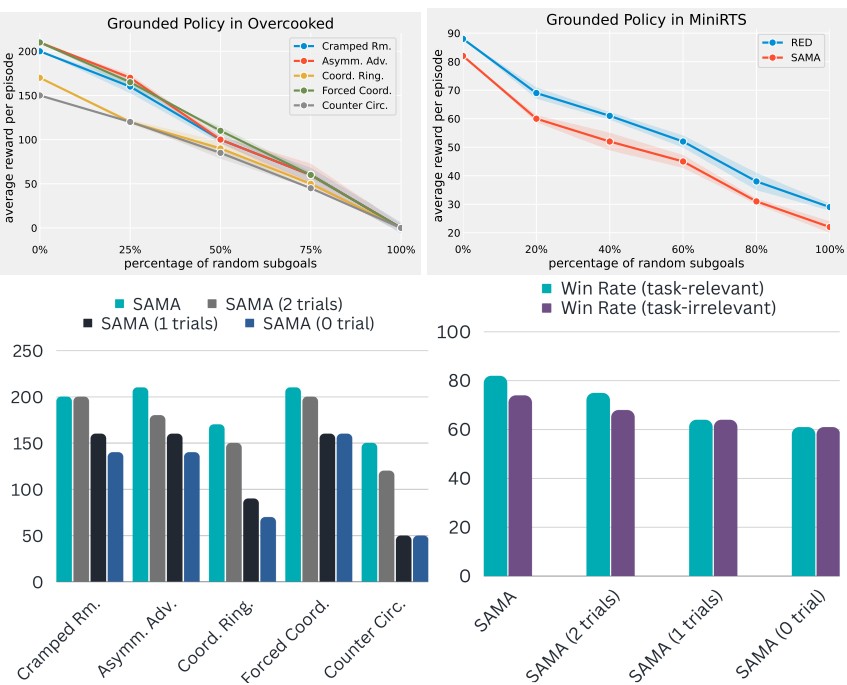

Figure 14: **Top:** Rewards per episode and win rates with an increasing amount of random commands. **Bottom:** Effect of different number of self-reflection trials on performance. "0 trial" means no self-reflection. Generally, only more than 1 trial can bring about significant performance improvement. The in-context examples that are not task-relevant also demonstrate decent performance, which suggests that the self-reflection mechanism has a certain level of generalizability. This can help to some extent in reducing the amount of prompt engineering required.

**Is self-reflection generalizable?** To further test the generality of the self-reflection component, we conducted an additional quick experiment. Specifically, inspired by Min et al. (2022), we tried applying the `Overcooked-AI`-designed self-reflection in-context example to the `MiniRTS` task. A plausible explanation for promising results lies in the evidence presented by Min et al. (2022), suggesting that in-context examples can improve PLM's classification ability even without providing correct labels for classification tasks. One potential reason is that these examples constrain the input and output spaces, guiding the PLM's behavior. Similarly, in our case, while the `Overcooked-AI`-designed self-reflection chain of thought cannot directly guide `MiniRTS` task decomposition, it does specify the "chain-of-thought space" from input (current state, failure flag, etc.) to output (regenerated goals, subgoals, etc.). As shown in Figure 14, even non-task-relevant in-context examples exhibit decent performance, indicating a certain level of generalizability in the self-reflection mechanism, which could help reduce the extent of prompt engineering required.

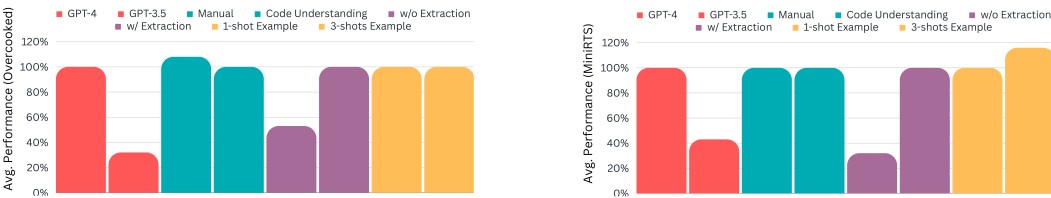

Figure 15: **Left:** the average performance of different ablations in all tasks in `Overcooked`. **Right:** the average win rate of different ablations in `MiniRTS`.

**Do different versions of GPT have a big impact on performance?** We tested the average performance of GPT-3.5 and GPT-4 on all tasks in two environments, as shown in the Figure 15. The vertical axis shows the percentage of different algorithms compared to SAMA. As can be seen from

the figure, there is a huge gap in performance between GPT-3.5 and GPT-4 (default in SAMA). The poorer performance of GPT-3.5 is also consistent with some existing findings (Chen et al., 2023a).

**Can PLM design high-quality reward functions?** When pre-training language-grounded MARL agents, intrinsic rewards need to be designed to indicate whether the subgoal is completed. Since the state fed back by `Overcooked` and the `MiniRTS` environment contains high-level semantic information, it is easy for humans to write a judgment function for whether the corresponding subgoal is completed. For example, in `Overcooked`, you can directly read from the state whether there are onions on the cooking table, how many onions there are, etc.; in `MiniRTS`, you can also read from the state the type and quantity of the currently manufactured building or army. Then we can use a manually designed reward function to evaluate the performance of the reward function automatically generated by PLM. As can be seen from Figure 15, in the `Overcooked` environment, the reward function automatically generated by PLM is slightly weaker than the manual design; but it shows comparable performance in `MiniRTS`. A simple analysis shows that the latter subgoals are easier to understand by PLM, such as confirming the type of building, type of army and number of troops built. The former subgoal requires a certain level of understanding. For example, for the subgoal of "putting onions from the storage room to the cooking table", the storage room is actually meaningless. The key is to detect whether there is an extra onion on the cooking table. And here it's no longer a simple counting task, but a comparison with the number of onions currently on the cooking table.

**Does the extraction of interactive objects effectively limit the goal space generated by PLM and further improve performance?** We tested not performing the preprocessing step of interactive objects extraction during the goal generation, decomposition and subgoal assignment phase, but directly prompting PLM to generate goals and subgoals according to the task manual. As can be seen from the Figure 15, this brings a significant performance degradation. The fundamental reason is that unbounded goal space causes language input to contain a large amount of redundant information, which greatly affects the pre-training and performance of language-grounded policy. Poor language-grounded policy ultimately leads to reduced task completion.

**Can more hand-designed few-shot examples improve performance?** Few-shot, or in-context examples, are considered critical to improving PLM performance. But more examples mean more labor costs and worse generalization capabilities. We evaluate the performance of the algorithm with 1 and 3 few-shot examples in all sessions that require prompt PLM. As can be seen from Figure 15, more few-shot examples will not bring performance improvements in all tasks. Considering the balance between performance and cost, 1-shot example is used as the default setting in SAMA.

# I  MORE ENVIRONMENTS AND BASELINES

## I.1  ENVIRONMENTS

This section mainly wants to verify two points: first, whether SAMA can handle tasks with more agents; second, whether SAMA can handle tasks where human common sense plays little role.

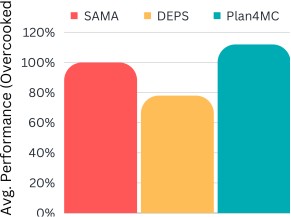 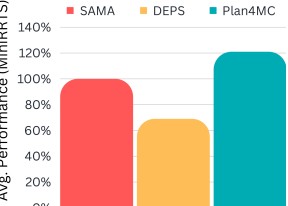 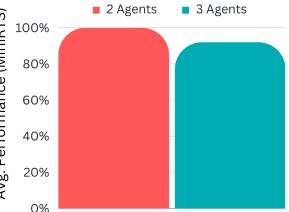

Figure 16: **Left:** the average performance of different PLM-based task planner in all tasks in `Overcooked`. **Middle:** the average win rate of different PLM-based task planner in `MiniRTS`. **Right:** the average win rate of SAMA in `MiniRTS` with different number of agents.

For the first point, since there are currently few environments in MARL that can better adapt to language models and be solved by human commonsense like `Overcooked` or `MiniRTS`, we made simple modifications to the latter to enable it to accommodate more agents at the same time. Similar to modifying the environment from a single-agent to a 2-agents environment, we can also use a

similar method to extend it to $N$ agents. Here we set $N$ to 3. Specifically, 3 agents emerge in the modified environment, each governing $1/3$ of the units. Analogously, we also modestly modified the built-in medium-level AI script, enabling the random selection of 3 types of army units in each iteration. Given that the built-in script AI constructs only 3 army unit types per game, we establish an oracle prompt design strategy following the ground truth of enemy units and the attack graph.

When the number of agents is larger, feasible plans will grow exponentially as the task horizon increases, which will be more challenging for current PLM. As can be seen from Figure 16, the performance of SAMA does not drop significantly in scenarios with more agents. This shows that the SAMA framework itself has certain scalability.

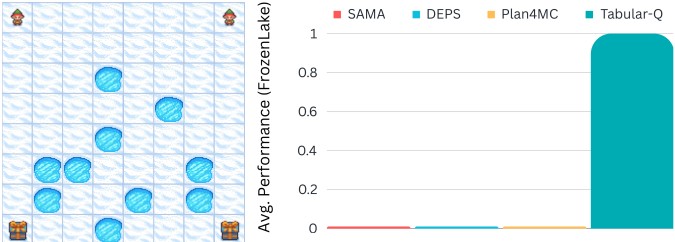

Figure 17: **Left:** The map of multi-agent version `FrozenLake` task.. **Right:** the average success rate of different PLM-based task planner in multi-agent version `FrozenLake`.

For the second point, we have modified the classic `FrozenLake` environment[12] in OpenAI Gym and expanded it into a task for two agents, which can be regarded as a discrete version of the Spread task in MPE[13]. In this task, two agents need to avoid holes in the ice and each reach a target location (Figure 17 Left). The difficulty with this task is that each agent's observation space contains only integers representing its own coordinates and those of another agent. Other than that, it contains no information and can only be remembered through continuous exploration to remember all possible situations. In this case, human commonsense cannot give the optimal planning in advance.

We verify the performance of SAMA and other PLM-based task planners (see details below) on the multi-agent version `FrozenLake`. Figure 17 (Right) shows the average rewards of different algorithms over 20 episodes of the game. Note that `FrozenLake` is a sparse reward task, and there is a $+1$ reward only when the agent reaches the target point. As can be seen from the figure, the PLM-based task planner we selected, including SAMA, is unable to solve this type of problem. How we empower PLM to solve such tasks will be left to further exploration later.

There could be two potential approaches to enable language agents, such as SAMA, to handle this type of problem. The first approach is lightweight, in which the language agents can learn through trial and error, similar to how MARL agents explore their environment, gather experience, and assist with subsequent decision-making. The feasibility of this approach has been preliminarily verified in a recent work (Anonymous, 2023). The second approach is a heavier one involving knowledge editing (De Cao et al., 2021), where task-specific knowledge is injected or modified in a task-oriented manner without altering the PLM's general capabilities. This second method is gradually becoming popular in the NLP field, with many papers published (Cao et al., 2023).

## I.2 BASELINES

In this section, we additionally compare the performance of other PLM-based task planners on `Overcooked` and `MiniRTS`. Specifically, we selected two algorithms, DEPS (Wang et al., 2023) and Plan4MC (Yuan et al., 2023), to compare with SAMA. The reason is that both algorithms require planning based on existing goal or skill sets. SAMA needs to generate a diverse set of goals in the language-grounded MARL training stage, so it is very suitable to replace the SAMA process with DEPS or Plan4MC.

---

[12]https://gymnasium.farama.org/environments/toy_text/frozen_lake/.
[13]https://github.com/openai/multiagent-particle-envs.

Specifically, DEPS generates an entire goal sequence at the beginning of the task. Then the reflection mechanism is used to adjust the goal sequence based on the policy execution results. Plan4MC will determine the goal to be generated by searching on the pre-constructed skill graph at each round of goal generation, and SAMA also adopt this paradigm. It is worth noting that both baselines are designed for single-agent tasks (i.e. Minecraft). Therefore, for a fair comparison, we use the goal decomposition in SAMA and subsequent processes after goal generation in DEPS or Plan4MC. In other words, we only replace the goal generation procedure in SAMA with DEPS or Plan4MC.

As can be seen from Figure 16 (Left and Middle), DEPS needs to generate an entire goal sequence at one time, which greatly increases the chance of PLM making mistakes, resulting in a performance degradation compared to SAMA. Plan4MC generates goal round-by-round and introduces graph search technology, so it has a slight performance improvement compared to SAMA. But it is worth noting that in SAMA's framework, goal generation is only one of many modules.

## J    OTHER LIMITAIONS

**The Correctness of the Generated Task Manual and Reward Function**    In the current version of SAMA, we have not designed additional mechanisms to ensure the correctness of the generated task manual and the generated reward function. From the experimental results, this does not affect SAMA's ability to approach SOTA baselines and achieve a leading sample efficiency. In Figure 15, we have also compared the performance impact of the human-designed reward function with that of the PLM-generated reward function. It can be seen from the figure that both methods show comparable performance levels. We plan to introduce a meta-prompt mechanism (Yang et al., 2023; Ma et al., 2023) to adjust the output of the PLM during the task manual and reward function generation stages based on the performance of the final algorithm, thus further improving the performance of the SAMA algorithm. However, as this mechanism may bring significant framework changes to the SAMA method, and the current version of SAMA is mainly designed for proof-of-concept purposes, we plan to explore the meta-prompt mechanism in more depth in our subsequent work.

**Document-Free Preprocessing**    Although the vast majority of tasks are usually accompanied by academic papers, technical reports, or documentation, there may still be a small number of boundary tasks that do not meet these requirements. There are several potential technical approaches to generate task manuals and carry out state/action translations in different ways. These approaches have their respective advantages and disadvantages when compared to the method employed by SAMA.

Firstly, we can have language agents create task manuals incrementally by continually interacting with the environment, receiving feedback, and summarizing their experiences without relying on assumed resources like research papers or technical reports. The feasibility of this approach has already been preliminarily validated in some studies (Chen et al., 2023b). However, a limitation of this approach is that the generation of task manuals deeply intertwines with the task-solving process, and the language agents' capabilities influence the quality of these manuals in solving specific tasks, such as exploration, planning, reasoning, and induction. This deep coupling may lead to negative feedback loops, causing the quality of the manuals and the solving strategies to converge prematurely to suboptimal, or even trivial, solutions.

Secondly, for state/action translation, we can use pre-trained multimodal large models to process image frames (Wang et al., 2023a) or directly compare differences between adjacent frames (Du et al., 2023b;a). This approach has also received preliminary validation through previous work. Nevertheless, it performs poorly on non-image tasks, making it less versatile than SAMA. Additionally, most pre-trained multimodal models use predominantly natural images for training, which limits their generalizability to tasks involving non-natural images, such as those found in gaming or simulation environments.

## K    SOCIETAL IMPACT

Though PLMs priors have demonstrated remarkable common-sense aptitudes, it is widely recognized that these models exhibit significant susceptibility to detrimental social prejudices and stereotypes (Bender et al., 2021; Abid et al., 2021; Nadeem et al., 2021). When employing such models as goal generators, decomposers, and allocators within the MARL context, as SAMA exemplifies, it is

imperative to thoroughly comprehend and counteract any potential adverse manifestations engendered by these biases. While our investigation is centered upon simulated environments and tasks, we assert the necessity for more rigorous examination if such systems were to be implemented in pursuit of open-ended learning within real-world settings, such as autonomous driving, multi-agent pathfinding, cloud computing, etc. Potential ameliorative approaches specific to SAMA could encompass (Du et al., 2023): the proactive screening of PLM generations to eradicate defamatory content prior to their utilization as generated goals and subgoals, prompting the PLM with directives concerning the nature of goals or subgoals to generate and/or the exclusive employment of the closed-form SAMA variant accompanied by meticulously confined goal and subgoal spaces.

## REFERENCES FOR SUPPLEMENTARY MATERIAL

Abubakar Abid, Maheen Farooqi, and James Zou. Persistent anti-muslim bias in large language models. In *AIES*, 2021.

Jean-Baptiste Alayrac, Jeff Donahue, Pauline Luc, Antoine Miech, Iain Barr, Yana Hasson, Karel Lenc, Arthur Mensch, Katherine Millican, Malcolm Reynolds, et al. Flamingo: a visual language model for few-shot learning. In *NeurIPS*, 2022.

Anonymous. Can language agents approach the performance of RL? an empirical study on openAI gym. In *Submitted to The Twelfth International Conference on Learning Representations*, 2023. URL https://openreview.net/forum?id=F0q880yOgY. under review.

Michael Becht, Thorsten Gurzki, Jürgen Klarmann, and Matthias Muscholl. Rope: Role oriented programming environment for multiagent systems. In *IFCIS*, 1999.

Marc Bellemare, Sriram Srinivasan, Georg Ostrovski, Tom Schaul, David Saxton, and Remi Munos. Unifying count-based exploration and intrinsic motivation. In *NeurIPS*, 2016.

Emily M Bender, Timnit Gebru, Angelina McMillan-Major, and Shmargaret Shmitchell. On the dangers of stochastic parrots: Can language models be too big? In *ACM FAccT*, 2021.

Valts Blukis, Yannick Terme, Eyvind Niklasson, Ross A Knepper, and Yoav Artzi. Learning to map natural language instructions to physical quadcopter control using simulated flight. In *CoRL*, 2020.

Wendelin Böhmer, Vitaly Kurin, and Shimon Whiteson. Deep coordination graphs. In *ICML*. PMLR, 2020.

SRK Branavan, David Silver, and Regina Barzilay. Learning to win by reading manuals in a monte-carlo framework. *Journal of Artificial Intelligence Research*, 43:661–704, 2012.

Anthony Brohan, Yevgen Chebotar, Chelsea Finn, Karol Hausman, Alexander Herzog, Daniel Ho, Julian Ibarz, Alex Irpan, Eric Jang, Ryan Julian, et al. Do as i can, not as i say: Grounding language in robotic affordances. In *CoRL*, 2023.

Tom Brown, Benjamin Mann, Nick Ryder, Melanie Subbiah, Jared D Kaplan, Prafulla Dhariwal, Arvind Neelakantan, Pranav Shyam, Girish Sastry, Amanda Askell, et al. Language models are few-shot learners. In *NeurIPS*, 2020.

Yuri Burda, Harrison Edwards, Amos Storkey, and Oleg Klimov. Exploration by random network distillation. In *ICLR*, 2019.

Boxi Cao, Hongyu Lin, Xianpei Han, and Le Sun. The life cycle of knowledge in big language models: A survey. *arXiv preprint arXiv:2303.07616*, 2023.

Micah Carroll, Rohin Shah, Mark K Ho, Tom Griffiths, Sanjit Seshia, Pieter Abbeel, and Anca Dragan. On the utility of learning about humans for human-ai coordination. In *NeurIPS*, 2019.

Thomas Carta, Clément Romac, Thomas Wolf, Sylvain Lamprier, Olivier Sigaud, and Pierre-Yves Oudeyer. Grounding large language models in interactive environments with online reinforcement learning. *arXiv preprint arXiv:2302.02662*, 2023.

Rujikorn Charakorn, Poramate Manoonpong, and Nat Dilokthanakul. Investigating partner diversification methods in cooperative multi-agent deep reinforcement learning. In *ICONIP*, 2020.

Lingjiao Chen, Matei Zaharia, and James Zou. How is chatgpt's behavior changing over time? *arXiv preprint arXiv:2307.09009*, 2023a.

Liting Chen, Lu Wang, Hang Dong, Yali Du, Jie Yan, Fangkai Yang, Shuang Li, Pu Zhao, Si Qin, Saravan Rajmohan, et al. Introspective tips: Large language model for in-context decision making. *arXiv preprint arXiv:2305.11598*, 2023b.

Zhoujun Cheng, Jungo Kasai, and Tao Yu. Batch prompting: Efficient inference with large language model apis. *arXiv preprint arXiv:2301.08721*, 2023.

Jongwook Choi, Archit Sharma, Honglak Lee, Sergey Levine, and Shixiang Shane Gu. Variational empowerment as representation learning for goal-conditioned reinforcement learning. In *ICML*, 2021.

John D Co-Reyes, Abhishek Gupta, Suvansh Sanjeev, Nick Altieri, John DeNero, Pieter Abbeel, and Sergey Levine. Meta-learning language-guided policy learning. In *ICLR*, 2019.

Cédric Colas, Tristan Karch, Olivier Sigaud, and Pierre-Yves Oudeyer. Autotelic agents with intrinsically motivated goal-conditioned reinforcement learning: a short survey. *Journal of Artificial Intelligence Research*, 74:1159–1199, 2022a.

Cédric Colas, Tristan Karch, Olivier Sigaud, and Pierre-Yves Oudeyer. Autotelic agents with intrinsically motivated goal-conditioned reinforcement learning: a short survey. *Journal of Artificial Intelligence Research*, 74:1159–1199, 2022b.

Ishita Dasgupta, Christine Kaeser-Chen, Kenneth Marino, Arun Ahuja, Sheila Babayan, Felix Hill, and Rob Fergus. Collaborating with language models for embodied reasoning. *arXiv preprint arXiv:2302.00763*, 2023.

Nicola De Cao, Wilker Aziz, and Ivan Titov. Editing factual knowledge in language models. In *EMNLP*, 2021.

Sam Devlin, Logan Yliniemi, Daniel Kudenko, and Kagan Tumer. Potential-based difference rewards for multiagent reinforcement learning. In *AAMAS*, 2014.

Ziluo Ding, Wanpeng Zhang, Junpeng Yue, Xiangjun Wang, Tiejun Huang, and Zongqing Lu. Entity divider with language grounding in multi-agent reinforcement learning. In *ICML*, 2023.

Danny Driess, Fei Xia, Mehdi SM Sajjadi, Corey Lynch, Aakanksha Chowdhery, Brian Ichter, Ayzaan Wahid, Jonathan Tompson, Quan Vuong, Tianhe Yu, et al. Palm-e: An embodied multimodal language model. *arXiv preprint arXiv:2303.03378*, 2023.

Yilun Du, Mengjiao Yang, Pete Florence, Fei Xia, Ayzaan Wahid, Brian Ichter, Pierre Sermanet, Tianhe Yu, Pieter Abbeel, Joshua B Tenenbaum, et al. Video language planning. *arXiv preprint arXiv:2310.10625*, 2023a.

Yilun Du, Sherry Yang, Bo Dai, Hanjun Dai, Ofir Nachum, Joshua B Tenenbaum, Dale Schuurmans, and Pieter Abbeel. Learning universal policies via text-guided video generation. In *NeurIPS*, 2023b.

Yuqing Du, Olivia Watkins, Zihan Wang, Cédric Colas, Trevor Darrell, Pieter Abbeel, Abhishek Gupta, and Jacob Andreas. Guiding pretraining in reinforcement learning with large language models. *arXiv preprint arXiv:2302.06692*, 2023c.

Adrien Ecoffet, Joost Huizinga, Joel Lehman, Kenneth O Stanley, and Jeff Clune. First return, then explore. *Nature*, 590(7847):580–586, 2021.

Linxi Fan, Guanzhi Wang, Yunfan Jiang, Ajay Mandlekar, Yuncong Yang, Haoyi Zhu, Andrew Tang, De-An Huang, Yuke Zhu, and Anima Anandkumar. Minedojo: Building open-ended embodied agents with internet-scale knowledge. In *NeurIPS Datasets and Benchmarks Track*, 2022.

Jakob Foerster, Gregory Farquhar, Triantafyllos Afouras, Nantas Nardelli, and Shimon Whiteson. Counterfactual multi-agent policy gradients. In *AAAI*, 2018.

Eva-Maria Grote, Stefan Achilles Pfeifer, Daniel Röltgen, Arno Kühn, and Roman Dumitrescu. Towards defining role models in advanced systems engineering. In *ISSE*, 2020.

Zhaohan Guo, Shantanu Thakoor, Miruna Pîslar, Bernardo Avila Pires, Florent Altché, Corentin Tallec, Alaa Saade, Daniele Calandriello, Jean-Bastien Grill, Yunhao Tang, et al. Byol-explore: Exploration by bootstrapped prediction. In *NeurIPS*, 2022.

Tarun Gupta, Anuj Mahajan, Bei Peng, Wendelin Böhmer, and Shimon Whiteson. Uneven: Universal value exploration for multi-agent reinforcement learning. In *ICML*, 2021.

Tarun Gupta, Peter Karkus, Tong Che, Danfei Xu, and Marco Pavone. Foundation models for semantic novelty in reinforcement learning. In *NeurIPS Foundation Models for Decision Making Workshop*, 2022.

Austin W Hanjie, Victor Y Zhong, and Karthik Narasimhan. Grounding language to entities and dynamics for generalization in reinforcement learning. In *ICML*, 2021.

Felix Hill, Andrew Lampinen, Rosalia Schneider, Stephen Clark, Matthew Botvinick, James L. McClelland, and Adam Santoro. Environmental drivers of systematicity and generalization in a situated agent. In *ICLR*, 2020a.

Felix Hill, Sona Mokra, Nathaniel Wong, and Tim Harley. Human instruction-following with deep reinforcement learning via transfer-learning from text. *arXiv preprint arXiv:2005.09382*, 2020b.

Felix Hill, Olivier Tieleman, Tamara von Glehn, Nathaniel Wong, Hamza Merzic, and Stephen Clark. Grounded language learning fast and slow. In *ICLR*, 2021.

Hengyuan Hu, Denis Yarats, Qucheng Gong, Yuandong Tian, and Mike Lewis. Hierarchical decision making by generating and following natural language instructions. In *NeurIPS*, 2019.

Wenlong Huang, Pieter Abbeel, Deepak Pathak, and Igor Mordatch. Language models as zero-shot planners: Extracting actionable knowledge for embodied agents. In *ICML*, 2022a.

Wenlong Huang, Fei Xia, Ted Xiao, Harris Chan, Jacky Liang, Pete Florence, Andy Zeng, Jonathan Tompson, Igor Mordatch, Yevgen Chebotar, Pierre Sermanet, Tomas Jackson, Noah Brown, Linda Luu, Sergey Levine, Karol Hausman, and brian ichter. Inner monologue: Embodied reasoning through planning with language models. In *CoRL*, 2022b.

Michael Janner, Karthik Narasimhan, and Regina Barzilay. Representation learning for grounded spatial reasoning. In *ACL*, 2018.

Jeewon Jeon, Woojun Kim, Whiyoung Jung, and Youngchul Sung. Maser: Multi-agent reinforcement learning with subgoals generated from experience replay buffer. In *ICML*, 2022.

Yunfan Jiang, Agrim Gupta, Zichen Zhang, Guanzhi Wang, Yongqiang Dou, Yanjun Chen, Li Fei-Fei, Anima Anandkumar, Yuke Zhu, and Linxi Fan. Vima: General robot manipulation with multimodal prompts. In *NeurIPS 2022 Foundation Models for Decision Making Workshop*, 2022.

Yipeng Kang, Tonghan Wang, Qianlan Yang, Xiaoran Wu, and Chongjie Zhang. Non-linear coordination graphs. In *NeurIPS*, 2022.

Jacob Devlin Ming-Wei Chang Kenton and Lee Kristina Toutanova. Bert: Pre-training of deep bidirectional transformers for language understanding. In *NAACL*, 2019.

Paul Knott, Micah Carroll, Sam Devlin, Kamil Ciosek, Katja Hofmann, Anca Dragan, and Rohin Shah. Evaluating the robustness of collaborative agents. In *AAMAS*, 2021.

Tejas D Kulkarni, Karthik Narasimhan, Ardavan Saeedi, and Josh Tenenbaum. Hierarchical deep reinforcement learning: Integrating temporal abstraction and intrinsic motivation. In *NeurIPS*, 2016.

Heinrich Küttler, Nantas Nardelli, Alexander Miller, Roberta Raileanu, Marco Selvatici, Edward Grefenstette, and Tim Rocktäschel. The nethack learning environment. In *NeurIPS*, 2020.

Minae Kwon, Sang Michael Xie, Kalesha Bullard, and Dorsa Sadigh. Reward design with language models. In *ICLR*, 2023.

Kemas M Lhaksmana, Yohei Murakami, and Toru Ishida. Role-based modeling for designing agent behavior in self-organizing multi-agent systems. *International Journal of Software Engineering and Knowledge Engineering*, 28(01):79–96, 2018.

Chenghao Li, Tonghan Wang, Chengjie Wu, Qianchuan Zhao, Jun Yang, and Chongjie Zhang. Celebrating diversity in shared multi-agent reinforcement learning. In *NeurIPS*, 2021a.

Shuang Li, Xavier Puig, Chris Paxton, Yilun Du, Clinton Wang, Linxi Fan, Tao Chen, De-An Huang, Ekin Akyürek, Anima Anandkumar, et al. Pre-trained language models for interactive decision-making. In *NeurIPS*, 2022.

Wenhao Li, Xiangfeng Wang, Bo Jin, Junjie Sheng, Yun Hua, and Hongyuan Zha. Structured diversification emergence via reinforced organization control and hierachical consensus learning. In *AAMAS*, 2021b.

Wenhao Li, Xiangfeng Wang, Bo Jin, Jingyi Lu, and Hongyuan Zha. Learning roles with emergent social value orientations. *arXiv preprint arXiv:2301.13812*, 2023a.

Yang Li, Shao Zhang, Jichen Sun, Yali Du, Ying Wen, Xinbing Wang, and Wei Pan. Cooperative open-ended learning framework for zero-shot coordination. *arXiv preprint arXiv:2302.04831*, 2023b.

Yecheng Jason Ma, William Liang, Guanzhi Wang, De-An Huang, Osbert Bastani, Dinesh Jayaraman, Yuke Zhu, Linxi Fan, and Anima Anandkumar. Eureka: Human-level reward design via coding large language models. *arXiv preprint arXiv:2310.12931*, 2023.

Anuj Mahajan, Tabish Rashid, Mikayel Samvelyan, and Shimon Whiteson. Maven: Multi-agent variational exploration. In *NeurIPS*, 2019.

Hong Min, Jinman Jung, Seoyeon Kim, Bongjae Kim, and Junyoung Heo. Role-based automatic programming framework for interworking a drone and wireless sensor networks. In *SAC*, 2018.

Sewon Min, Xinxi Lyu, Ari Holtzman, Mikel Artetxe, Mike Lewis, Hannaneh Hajishirzi, and Luke Zettlemoyer. Rethinking the role of demonstrations: What makes in-context learning work? In *EMNLP*, 2022.

Moin Nadeem, Anna Bethke, and Siva Reddy. Stereoset: Measuring stereotypical bias in pretrained language models. In *ACL*, 2021.

Karthik Narasimhan, Regina Barzilay, and Tommi Jaakkola. Grounding language for transfer in deep reinforcement learning. *Journal of Artificial Intelligence Research*, 63:849–874, 2018.

Duc Thien Nguyen, Akshat Kumar, and Hoong Chuin Lau. Credit assignment for collective multiagent rl with global rewards. In *NeurIPS*, 2018.

Dung Nguyen, Phuoc Nguyen, Svetha Venkatesh, and Truyen Tran. Learning to transfer role assignment across team sizes. In *AAMAS*, 2022.

Deepak Pathak, Pulkit Agrawal, Alexei A Efros, and Trevor Darrell. Curiosity-driven exploration by self-supervised prediction. In *ICML*, 2017.

Thomy Phan, Fabian Ritz, Lenz Belzner, Philipp Altmann, Thomas Gabor, and Claudia Linnhoff-Popien. Vast: Value function factorization with variable agent sub-teams. In *NeurIPS*, 2021.

Aditya Ramesh, Mikhail Pavlov, Gabriel Goh, Scott Gray, Chelsea Voss, Alec Radford, Mark Chen, and Ilya Sutskever. Zero-shot text-to-image generation. In *ICML*, 2021.

Tabish Rashid, Gregory Farquhar, Bei Peng, and Shimon Whiteson. Weighted qmix: Expanding monotonic value function factorisation for deep multi-agent reinforcement learning. In *NeurIPS*, 2020a.

Tabish Rashid, Mikayel Samvelyan, Christian Schroeder De Witt, Gregory Farquhar, Jakob Foerster, and Shimon Whiteson. Monotonic value function factorisation for deep multi-agent reinforcement learning. *The Journal of Machine Learning Research*, 21(1):7234–7284, 2020b.

Machel Reid, Yutaro Yamada, and Shixiang Shane Gu. Can wikipedia help offline reinforcement learning? *arXiv preprint arXiv:2201.12122*, 2022.

John Schulman, Filip Wolski, Prafulla Dhariwal, Alec Radford, and Oleg Klimov. Proximal policy optimization algorithms. *arXiv preprint arXiv:1707.06347*, 2017.

Jianzhun Shao, Zhiqiang Lou, Hongchang Zhang, Yuhang Jiang, Shuncheng He, and Xiangyang Ji. Self-organized group for cooperative multi-agent reinforcement learning. In *NeurIPS*, 2022.

Kyunghwan Son, Daewoo Kim, Wan Ju Kang, David Earl Hostallero, and Yung Yi. Qtran: Learning to factorize with transformation for cooperative multi-agent reinforcement learning. In *ICML*, 2019.

DJ Strouse, Kevin McKee, Matt Botvinick, Edward Hughes, and Richard Everett. Collaborating with humans without human data. In *NeurIPS*, 2021.

Peter Sunehag, Guy Lever, Audrunas Gruslys, Wojciech Marian Czarnecki, Vinicius Zambaldi, Max Jaderberg, Marc Lanctot, Nicolas Sonnerat, Joel Z Leibo, Karl Tuyls, et al. Value-decomposition networks for cooperative multi-agent learning based on team reward. In *AAMAS*, 2018.

Allison Tam, Neil Rabinowitz, Andrew Lampinen, Nicholas A Roy, Stephanie Chan, DJ Strouse, Jane Wang, Andrea Banino, and Felix Hill. Semantic exploration from language abstractions and pretrained representations. In *NeurIPS*, 2022.

Hongyao Tang, Jianye Hao, Tangjie Lv, Yingfeng Chen, Zongzhang Zhang, Hangtian Jia, Chunxu Ren, Yan Zheng, Zhaopeng Meng, Changjie Fan, et al. Hierarchical deep multiagent reinforcement learning with temporal abstraction. *arXiv preprint arXiv:1809.09332*, 2018.

Stefanie Tellex, Nakul Gopalan, Hadas Kress-Gazit, and Cynthia Matuszek. Robots that use language. *Annual Review of Control, Robotics, and Autonomous Systems*, 3:25–55, 2020.

Jianhao Wang, Zhizhou Ren, Beining Han, Jianing Ye, and Chongjie Zhang. Towards understanding cooperative multi-agent q-learning with value factorization. In *NeurIPS*, 2021a.

Tonghan Wang, Heng Dong, Victor Lesser, and Chongjie Zhang. Roma: multi-agent reinforcement learning with emergent roles. In *ICML*, 2020.

Tonghan Wang, Tarun Gupta, Anuj Mahajan, Bei Peng, Shimon Whiteson, and Chongjie Zhang. {RODE}: Learning roles to decompose multi-agent tasks. In *ICLR*, 2021b.

Xin Wang, Qiuyuan Huang, Asli Celikyilmaz, Jianfeng Gao, Dinghan Shen, Yuan-Fang Wang, William Yang Wang, and Lei Zhang. Reinforced cross-modal matching and self-supervised imitation learning for vision-language navigation. In *CVPR*, 2019.

Zihao Wang, Shaofei Cai, Anji Liu, Yonggang Jin, Jinbing Hou, Bowei Zhang, Haowei Lin, Zhaofeng He, Zilong Zheng, Yaodong Yang, et al. Jarvis-1: Open-world multi-task agents with memory-augmented multimodal language models. *arXiv preprint arXiv:2311.05997*, 2023a.

Zihao Wang, Shaofei Cai, Anji Liu, Xiaojian Ma, and Yitao Liang. Describe, explain, plan and select: Interactive planning with large language models enables open-world multi-task agents. *arXiv preprint arXiv:2302.01560*, 2023b.

Yuchen Xiao, Joshua Hoffman, and Christopher Amato. Macro-action-based deep multi-agent reinforcement learning. In *CoRL*, 2020.

Yuchen Xiao, Weihao Tan, and Christopher Amato. Asynchronous actor-critic for multi-agent reinforcement learning. In *NeurIPS*, 2022.

Shusheng Xu, Huaijie Wang, and Yi Wu. Grounded reinforcement learning: Learning to win the game under human commands. In *NeurIPS*, 2022.

Chengrun Yang, Xuezhi Wang, Yifeng Lu, Hanxiao Liu, Quoc V Le, Denny Zhou, and Xinyun Chen. Large language models as optimizers. *arXiv preprint arXiv:2309.03409*, 2023.

Jiachen Yang, Igor Borovikov, and Hongyuan Zha. Hierarchical cooperative multi-agent reinforcement learning with skill discovery. In *AAMAS*, 2020a.

Mingyu Yang, Jian Zhao, Xunhan Hu, Wengang Zhou, Jiangcheng Zhu, and Houqiang Li. Ldsa: Learning dynamic subtask assignment in cooperative multi-agent reinforcement learning. In *NeurIPS*, 2022a.

Qianlan Yang, Weijun Dong, Zhizhou Ren, Jianhao Wang, Tonghan Wang, and Chongjie Zhang. Self-organized polynomial-time coordination graphs. In *ICML*, 2022b.

Yaodong Yang, Ying Wen, Jun Wang, Liheng Chen, Kun Shao, David Mguni, and Weinan Zhang. Multi-agent determinantal q-learning. In *ICML*, 2020b.

Chao Yu, Akash Velu, Eugene Vinitsky, Jiaxuan Gao, Yu Wang, Alexandre Bayen, and Yi Wu. The surprising effectiveness of ppo in cooperative multi-agent games. In *NeurIPS*, 2022.

Haoqi Yuan, Chi Zhang, Hongcheng Wang, Feiyang Xie, Penglin Cai, Hao Dong, and Zongqing Lu. Plan4mc: Skill reinforcement learning and planning for open-world minecraft tasks. *arXiv preprint arXiv:2303.16563*, 2023.

Lu Yuan, Dongdong Chen, Yi-Ling Chen, Noel Codella, Xiyang Dai, Jianfeng Gao, Houdong Hu, Xuedong Huang, Boxin Li, Chunyuan Li, et al. Florence: A new foundation model for computer vision. *arXiv preprint arXiv:2111.11432*, 2021.

Tianhao Zhang, Yueheng Li, Chen Wang, Guangming Xie, and Zongqing Lu. Fop: Factorizing optimal joint policy of maximum-entropy multi-agent reinforcement learning. In *ICML*, 2021.

Lulu Zheng, Jiarui Chen, Jianhao Wang, Jiamin He, Yujing Hu, Yingfeng Chen, Changjie Fan, Yang Gao, and Chongjie Zhang. Episodic multi-agent reinforcement learning with curiosity-driven exploration. In *NeurIPS*, 2021.

Victor Zhong, Tim Rocktäschel, and Edward Grefenstette. Rtfm: Generalising to new environment dynamics via reading. In *ICLR*, 2020.

Victor Zhong, Austin W Hanjie, Sida Wang, Karthik Narasimhan, and Luke Zettlemoyer. Silg: The multi-domain symbolic interactive language grounding benchmark. In *NeurIPS*, 2021.

Xizhou Zhu, Yuntao Chen, Hao Tian, Chenxin Tao, Weijie Su, Chenyu Yang, Gao Huang, Bin Li, Lewei Lu, Xiaogang Wang, et al. Ghost in the minecraft: Generally capable agents for open-world enviroments via large language models with text-based knowledge and memory. *arXiv preprint arXiv:2305.17144*, 2023.

