# OpenReview forum: "Semantically Aligned Task Decomposition in Multi-Agent Reinforcement Learning"
_ICLR.cc/2024/Conference — Submitted to ICLR 2024_

### Official Review · Reviewer_DBEC · 2023-10-23

**Soundness:** 3 good
**Presentation:** 2 fair
**Contribution:** 3 good
**Rating:** 6
**Confidence:** 4

**Summary:**

Based on automatic subgoal generation, the authors design a sophisticated framework to perform goal generation, sub-goal assignment, and language-grounded goal-based MARL, along with effective techniques like in-context learning and self-reflection in the prompting engineering domain. The authors evaluate performance of derived policies on two benchmarks, Overcooked and MiniRTS.

**Strengths:**

1. The authors compose several techniques of prompting engineering to realize subgoal generation and train goal-conditioned agents, which seems significantly improve sample efficiency. The authors well introduce multiple advancements from current LLM-agent research and provide thorough discussions about related methods.
2. The framework the authors propose is comprehensive, though maybe too complicated, it provide a guidance of design an LLM-assisted agent.
3. The framework, SAMA, realizes semantically useful task decomposition by generating and assign explainable subgoals, which is advantageous compared to previous goal-based methods.

**Weaknesses:**

1. The major concern is that the sophisticated design of SAMA may hinder its general use on other benchmarks. It seems that the authors create exhaustive prompts for running SAMA on these two benchmarks (Page 23-36).
2. The accompanied concern is that the deployment of SAMA may induce high costs of calling LLMs (like OpenAI API) and training language-grounded agents.
3. In the experiments part, the authors do not make some illustrative examples of goal generation but only provide performance curves.

Some minor mistakes:
1. In the caption of Figure 2, "subgaol" -> "subgoal". The authors should also consider unifying the term to "subgoal" or "sub-goal" as they both appear in this paper.
2. Figure 4 (right) is too small to be read.

**Questions:**

1. Can you make a list of how many tasks are accomplished with the help of PLMs? Among them, how many can be done in offline setting and how many cannot? I think it will be better to evaluate the contributions.
2. To hack SAMA for another environment, can you summarize how many prompts/components the practitioner should modify?
3. How much percentage of goals and subgoals is generated in the offline manner? How many queries are needed during online training?
4. Can you provide the financial costs and time costs of training SAMA agents?
5. As the evaluation of Overcooked is based on self-play performance, what is the difference among those ad hoc teamwork methods (SP, FCP, and COLE)? In my opinion, the SP may be reduced to a general MAPPO algorithm. Why do ASG methods perform much worse than SP?
6. Why do you select the MiniRTS benchmark with splitting units rather than directly evaluating on other MARL benchmarks? It seems a little wired. Meanwhile, why cannot SAMA surpass RED?

---

> ### Author Response · Authors · 2023-11-18
>
> We greatly appreciate your taking the time to carefully review our paper and provide valuable suggestions and constructive feedback. This has significantly helped us improve the rigor and completeness of our paper. Additionally, we are grateful for your recognition of our work! Below, we have organized your questions and will address them individually.
>
> `Q1: The primary concern is that the sophisticated design of SAMA may hinder its general use on other benchmarks. The authors seem to create exhaustive prompts for running SAMA on these two benchmarks (Pages 23-36).`
>
> Thank you very much for your input. We apologize for any confusion arising from placing some experimental details in the appendix section of our work. The overall process of SAMA is not overly complex. Although it includes several stages, such as preprocessing, task decomposition, language-grounded MARL, and self-reflection, the first, second, and fourth stages primarily involve using different prompt engineering techniques to query PLMs. We've primarily relied on existing GPT plugins or third-party frameworks (e.g., LangChain [1]) when designing these prompt engineering techniques.
>
> In light of the preprocessing stage involving a higher density of queries for PLM, we would like to elaborate on this topic in more detail for a better understanding. Concretely, the preprocessing stage in generating the task manual is completed through prompt PLM, with key prompts listed in Appendix F.1. As you can see, only a few prompts are necessary. We utilized the open-source LangChain framework throughout the process, minimizing the coding burden. Furthermore, the preprocessing stage relies on retrieval-augmented generation, precisely document question answering, which is already supported by GPT-4 plugins. We believe that as large open-source models continue to develop, similar technologies will become increasingly mature while the barriers to entry and costs significantly decrease.
>
> Most of the prompt engineering in this paper comprises task decomposition, self-reflection stages, and in-context examples. Although pages $23$-$36$ occupy a significant portion of the text, part of this section includes the task manual generated by the PLM and other outputs. Figures $12$ and $13$ contain redundant presentations of the in-context examples. For task decomposition and self-reflection stages, it is only necessary to create a single in-context example with a total token count not exceeding $1000$, which is approximately equivalent to $750$ words. We believe this level of prompt engineering workload results in improved generalization performance compared to few-shot learning algorithms when encountering new tasks while maintaining a polite and gentle tone.
>
> [1] https://langchain-langchain.vercel.app/docs/get_started/introduction
>
> `Q2: The concern is that deploying SAMA may induce high costs of calling LLMs (like OpenAI API) and training language-grounded agents.`
>
> Thank you very much for your question! In subsequent discussions, we have thoroughly analyzed the time and economic costs associated with SAMA training. In summary, the training time for SAMA does not significantly exceed that of complex MARL methods, and its economic costs are within an acceptable range. We believe that as the open-source large model community grows, these economic costs will be significantly reduced shortly, making it more accessible for everyone.
>
> `Q3: In the experiments part, the authors do not make some illustrative examples of goal generation but only provide performance curves.`
>
> Thank you for your insightful questions! They are incredibly beneficial to the completeness of our paper. Due to space constraints, we apologize for not providing more experimental details in the main text. To offer you a clearer understanding of the SAMA algorithm's operation, we have randomly selected some goals generated by PLMs and their corresponding subgoals in Appendix G. We hope these intermediate results will help clarify the process involved!
>
> `Q4: In the caption of Figure 2, "subgaol" -> "subgoal". The authors should also consider unifying the term "subgoal" or "sub-goal" as they appear in this paper.`
>
> Thank you very much for your suggestions! We apologize for any oversight on our part. In the revised version, we have refined the text and corrected typos. Additionally, we have consistently used the term "subgoal" throughout the manuscript, except in the prompt.
>
> `Q5: Figure 4 (right) is too small to be read.`
>
> Thank you very much for your suggestion! In the revised version, we have enlarged the display of this figure in Appendix E, making it easier for readers to see the details.

---

> ### Author Response · Authors · 2023-11-18
>
> `Q6: Can you list how many tasks are accomplished with the help of PLMs? How many can be done offline, and how many cannot? I think it will be better to evaluate the contributions.`
>
> We apologize if our main text might not have been clear enough, leading to potential misconceptions about the offline or pre-training process. Rest assured that all tasks are carried out with the assistance of PLM. Allow us to provide you with a more detailed explanation below.
>
> Figure 5 shows the training progress of the language-grounded MARL agent. In other words, Figure 5 demonstrates the "pre-training" progress of the MARL agent. We refer to the language-grounded MARL agent as a pre-trained model to emphasize that the PLM does not participate in the agent's training process, which would result in significant time and financial costs. The language-grounded MARL agent learns the correspondence between subgoals and actions based on a pre-generated set of subgoals from the PLM, continually interacting with its environment. We regularly test the currently trained MARL agent using the PLM, record its performance, and plot the training curve in Figure 5. Additionally, we update the subgoals with unseen ones during testing to promote the MARL agent's training.
>
> To make our paper more rigorous, we have revised some of the descriptions in Section 3.3 of the revised version to minimize the potential for confusion among readers. Thank you again for your question!
>
> `Q7: To hack SAMA for another environment, can you summarize how many prompts/components the practitioner should modify?`
>
> Thank you very much for your question! We apologize if our paper did not sufficiently explain the generalization issue, which may have led to some confusion. In the four stages of SAMA, which include preprocessing, task decomposition, language grounding, and self-reflection, both the preprocessing and language grounding phases are task-agnostic, covering the aspect of prompt engineering. For the task decomposition and self-reflection stages, it is only necessary to design a 1-shot in-context example with a total token count not exceeding 1,000 tokens, which is roughly equivalent to 750 words. Specifically, you would need to modify the prompts in Appendix F - Listing 9-15 (designed for Overcooked-AI) or Listing 16-18 (designed for MiniRTS).
>
> `Q8: How much percentage of goals and subgoals is generated offline? How many queries are needed during online training?`
>
> Thank you for your insightful questions! They are incredibly beneficial to the completeness of our paper. Firstly, concerning the first question, determining whether a goal or subgoal generated by a PLM exists in the offline dataset is not a simple task. One approach is to obtain the embeddings of the two goals to be compared using an open-source PLM and then assess their similarity (e.g., cosine similarity) to see if it exceeds a predefined threshold. However, this method can be influenced by the performance of the open-source PLM and the chosen threshold. In our paper, we adopted an alternative method, using a query to GPT-4 to discern if a goal is classified as a new one. According to our experimental statistics, over 97% of the goals were covered during the offline generation phase.
>
> Secondly, regarding the second question, we conducted an average of ten evaluations in the online training process of different tasks to produce the learning curve represented in Figure 5 and 6. Considering the randomness of PLMs, each evaluation result is the average calculated outcome under $10$ random seeds. For the Overcooked-AI task, an episode requires querying the PLM approximately $10$ times on average, while for MiniRTS, a single query is needed at the beginning of the episode. Consequently, for the Overcooked-AI task, the total number of queries is $10$ (evaluation times) times $10$ (number of random seeds) times $10$, which equals $1000$ queries, and for MiniRTS, it is $10$ times $10$ times $1$, which equals $100$ queries.
>
> We have added the above discussion to the Appendix E and G in the revised version.

---

> ### Author Response · Authors · 2023-11-18
>
> `Q9: Can you provide the financial and time costs of training SAMA agents?`
>
> Thank you for your insightful questions! They are incredibly beneficial to the completeness of our paper. The number of tokens required for each query to the PLM is similar for both Overcooked-AI and MiniRTS, at around 3,000 tokens. Therefore, for Overcooked-AI, the total number of tokens needed for training is approximately 1000 * 3,000 = 3,000,000. The cost for GPT-3.5-turbo is 3 dollars, while the cost for GPT-4 is 90 dollars. The output of the PLM is around 1,500 tokens. Thus, the total output tokens amount is approximately 1,500,000. Since the cost of output tokens is twice that of input tokens, the cost is consistent. For MiniRTS, the total number of tokens needed for training is approximately 100 * 3,000 = 300,000. The cost for GPT-3.5-turbo is 0.3 dollars, while the cost for GPT-4 is 9 dollars. The output of the PLM is also around 1,500 tokens. Thus, the cost is also consistent with the input.
>
> The computing hardware comprises $2$ servers, each with 256GB of memory, and a pair of NVIDIA GeForce RTX 3090s, outfitted with 24GB of video memory. The training time (plus PLM query time) for Overcooked-AI is around $43+3$ hours and $22+3$ hours for MiniRTS.
>
> We have added the above discussion to the Appendix E in the revised version.
>
> `Q10: As the evaluation of Overcooked is based on self-play performance, what is the difference among those ad hoc teamwork methods (SP, FCP, and COLE)? I think the SP may be reduced to a general MAPPO algorithm. Why do ASG methods perform much worse than SP?`
>
> This is an excellent question. SP, FCP, COLE, and other representative works all fall under population-based training (PBT) methods. Although they ultimately rely on self-play for policy evaluation, there are distinctions among these methods within the context of population training. For example, differences can arise in how individuals within the population are matched and how new individuals are generated. Some methods also incorporate mechanisms like theory-of-mind and recursive reasoning to assist in training.
>
> From the current line of research on Overcooked-AI, it appears that PBT is a more efficient approach to solving this problem. While the basic framework of SP can be implemented using MAPPO, its PBT training process differs significantly from that of MAPPO. We believe that the main reason ASG methods have been unable to outperform PBT approaches is that existing ASG methods have not taken Overcooked-AI's strong cooperation aspect as a standard test environment and have not optimized specifically for it. This has led to suboptimal performance for ASG methods in the Overcooked-AI task even after considerable parameter tuning.
>
> However, we still regard ASG as a highly versatile framework. It may be worth exploring the incorporation of specialized mechanisms for tasks involving ad hoc teamwork and strong cooperation within the ASG framework. We believe this is an intriguing and promising research direction.

---

> ### Author Response · Authors · 2023-11-18
>
> `Q11: Why do you select the MiniRTS benchmark with splitting units rather than directly evaluating other MARL benchmarks? It seems a little weird. Meanwhile, why cannot SAMA surpass RED?`
>
> Thank you very much for your questions. For the first one, we chose Overcooked-AI and MiniRTS over the more complex task of SMAC for the following principal reasons: Firstly, the tasks in these two environments can benefit more significantly from human common sense and have more straightforward manuals and state/action translations. Secondly, their task decomposition and sub-goal allocation processes can be analyzed more intuitively. For example, cooking involves chopping and plating, while MiniRTS has transparent unit counter relationships. Finally, both environments allow for more accessible training of language-grounded MARL agents, with MiniRTS even having pre-trained agents available. These features make these two environments highly suitable for our goal of designing a proof-of-concept algorithm.
>
> The experimental results indicate that SAMA cannot fully solve the tasks in Overcooked-AI and MiniRTS. Hence, using environments like SMAC directly would not be very reasonable. Moreover, employing more complex environments like SMAC would require more meticulous prompt engineering and higher-cost language-grounded MARL agent training and make the experimental results more complicated to analyze. These points do not align with our intention of quickly designing a proof-of-concept algorithm.
>
> Additionally, the supplementary experimental results in Appendix I.1 show a certain degree of scalability for the SAMA pipeline. We believe that by employing more advanced and carefully designed tools or technologies to improve each module, SAMA could attempt to tackle complex tasks like SMAC. However, in this paper, we believe that the most significant contribution of SAMA lies in connecting the use of PLM and MARL methods for solving multi-agent tasks and validating its effectiveness and substantial optimization for sample complexity through a proof-of-concept experiment.
>
> For the second one, the inability of SAMA to surpass RED is primarily due to the inherent limitations of the PLM itself, which prevent it from consistently generating accurate goals, decomposing them, and outputting the correct commands—even with the assistance of the self-reflection mechanism. On the contrary, RED achieves better results by utilizing an RL training paradigm that optimizes commands while ensuring language grounding, ultimately maximizing the win rate.

---

> ### Author Response · Authors · 2023-11-21
> **Have we addressed your concerns?**
>
> Thanks again for your time and effort in reviewing our paper! As the discussion period is coming to a close, we would like to know if we have resolved your concerns expressed in the original reviews. We remain open to any further feedback and are committed to making additional improvements if needed. If you find that these concerns have been resolved, we would be grateful if you would consider reflecting this in your rating of our paper :)

---

> > ### Comment · Reviewer_DBEC · 2023-11-22
> > **Some followup comments on your rebuttal**
> >
> > I thank the authors for their detailed rebuttal, which addresses some of my concerns. I appreciate the clarification about financial cost and difficulties of domain adapataions. I'd like to check some of my understandings about SAMA.
> >
> > ---
> >
> > SAMA firstly generates subgoals and their assignments in an offline manner. Then it write code snippets as the reward functions. Does this process require pre-collected data or is purely executed from papers/docs?
> >
> > ---
> >
> > I see the authors illustrate Figure 5 and Figure 6 as the pretrained phases. The authors claim that
> >
> > > We refer to the language-grounded MARL agent as a pre-trained model to emphasize that the PLM does not participate in the agent's training process, which would result in significant time and financial costs.
> >
> > I wonder what the pretraining process and training process are exactly. Is the training process about freezing the grounding representation and then fine-tuning? How can PLMs not participate in the agent's training process? In my opinion, the process of self-reflection needs interactive feedback.
> >
> > ---
> >
> > In general, I hope the authors should carefully consider the presentation. Though some fancy figures have been made, I think more illustrations on the overall process will be beneficial. For example, the authors may consider distinguishing procedures that don't need additional data, require interaction data, and require RL training. The current version still confuses me a lot about what kinds of training/LLM calls are indeed operated.

---

> ### Author Response · Authors · 2023-11-22
>
> We greatly appreciate your prompt response and the opportunity you've given us to continue our discussion. We'll now address each of your additional questions one by one.
>
> `Q1: SAMA first generates subgoals and their assignments in an offline manner. Then, it writes code snippets as the reward functions. Does this process require pre-collected data, or is it purely executed from papers/docs?`
>
> Thank you very much for your question! We apologize for not emphasizing this issue's explanation in the main text. To address it, **the process relies solely on papers and codes**. Specifically, we first utilize the technology implemented for **state and action translation** to transform PLM prompt content from "translation" tasks to the generation of a diverse set of states, $\mathcal{D}_{\text{state}}$. Subsequently, we can generate subgoals and allocate them by employing the methods used in the **task decomposition** stage. Simultaneously, we produce code snippets corresponding to the subgoals' reward functions.
>
> `Q2: I wonder what the pretraining process and training process are precisely. Is the training process about freezing the grounding representation and then fine-tuning it? How can PLMs not participate in the agent's training process? In my opinion, the process of self-reflection needs interactive feedback.`
>
> `Q3: I hope the authors should consider the presentation carefully. Though some fancy figures have been made, I think more illustrations of the overall process will be beneficial. For example, the authors may consider distinguishing procedures that don't need additional data, require interaction data, and require RL training. The current version still confuses me a lot about what kinds of training/LLM calls are operated.`
>
> We apologize for any confusion caused by our inappropriate use of the term "pretrain." In the revised version, we have abandoned the concept of pretraining and removed related descriptions, except in the MiniRTS task where we indeed employed an open-source pre-trained model. The training process for language grounding agents is as follows (Section 3.3): First, we generate a training set for language grounding agents using the procedure described in Q1, which consists of subgoals and reward functions. Then, we train the language grounding agents by allowing them to interact with the environment according to the training set and periodically evaluating the agents. However, the evaluations are quite sparse (averaging about 10 times), so they do not incur substantial overhead. Figure 5 is drawn based on these evaluation results. In summary, **while PLMs are involved in the training of language grounding agents, the purpose is to monitor training progress**.
>
> Additionally, since the evaluation process may generate subgoals not seen in the training set, we incorporate newly generated subgoals into the training set after each evaluation. Once the language grounding agents have been trained, the PLM can utilize the trained agents when executing tasks.
>
> Lastly, we truly appreciate your suggestion to include illustrative diagrams and tables, as they will greatly improve the readability of our paper. In the revised version, we have added Table 1 to **explicitly state the resources required at different procedures**, such as the paper PDF files, code, environment interaction data, etc., as well as whether neural network parameter updates (MARL training) are needed. The table is provided below:
>
> | Procedures | Required resources | MARL or not |
> | --- | --- | --- |
> | Preprocessing | Papers and codes | no MARL |
> | Semantically-aligned task decomposition | Rollout samples | no MARL |
> | Language-grounded MARL | Papers, codes and rollout samples | MARL |
> | Self-reflection | Rollout samples | no MARL |

---

> > ### Comment · Reviewer_DBEC · 2023-11-23
> > **Score update and further comments**
> >
> > I thank the authors for their explanations, which address most of my concerns. I'm willing to update my score to 6.
> >
> > I believe that the contribution of this paper is generally good and can be helpful as an example of introducing complex LLM agent framework to MARL. However, current presentation may hinder my understanding about the benefits of each component. For further revision, I suggest the authors should reorganize the paper in the below ways:
> >
> > ---
> >
> > > The training process for language grounding agents is as follows (Section 3.3): First, we generate a training set for language grounding agents using the procedure described in Q1, which consists of subgoals and reward functions. Then, we train the language grounding agents by allowing them to interact with the environment according to the training set and periodically evaluating the agents. However, the evaluations are quite sparse (averaging about 10 times), so they do not incur substantial overhead.
> >
> > 1. I hope the authors can make this explanation in the paper.
> > 2. I think the authors can provide an illustration in the algorithmic form in the main paper to clarify the process.
> >
> > ---
> >
> > I still have a question about your comment:
> >
> > > we first utilize the technology implemented for state and action translation to transform PLM prompt content from "translation" tasks to the generation of a diverse set of states, $\mathcal{D}_{states}$.
> >
> > Can you please elaborate more about this sentence? I don't see how a diverse set of states can be acquired solely from papers and codes? I either cannot get my answer from the State and Action Translation section. Is it about writing codes to generate states?

---

> > > ### Author Response · Authors · 2023-11-23
> > > **Following discussions**
> > >
> > > Thank you very much for recognizing our work! We're pleased to hear that our clarifications have positively influenced your evaluation, and we also believe that the hard work we have done is worthwhile.
> > >
> > > ---
> > >
> > > We will now address each of the comments or questions you added in turn.
> > >
> > > `Q1: I believe that the contribution of this paper is generally good and can be helpful as an example of introducing complex LLM agent framework to MARL. However, current presentation may hinder my understanding about the benefits of each component. For further revision, I suggest the authors should reorganize the paper in the below ways:`
> > > `1. I hope the authors can make this explanation in the paper.`
> > > `2. I think the authors can provide an illustration in the algorithmic form in the main paper to clarify the process.`
> > >
> > > Thank you very much for your detailed suggestions! We have incorporated the mentioned content into Section 3.3 of the revised version and provided additional explanations in Appendix G. We will follow your advice and include pseudocode for the language grounded MARL algorithm training in the main text. As space in the revised version's main text is currently limited, we will redraw some of the diagrams to make room for the algorithm pseudocode. However, this may take some time, and considering the rebuttal period is coming to an end, we might not be able to update the paper in time. We appreciate your understanding.
> > >
> > > `Q2: Can you please elaborate more about this sentence? I don't see how a diverse set of states can be acquired solely from papers and codes? I either cannot get my answer from the State and Action Translation section. Is it about writing codes to generate states?`
> > >
> > > We apologize for not explaining the process in detail earlier. Specifically, since states are characterized by code rather than paper, we utilize environment code files for diverse state generation. Similar to the state and action translation section, we also employ LangChain's code understanding module.
> > >
> > > To elaborate, LangChain first divides the entire code project into multiple documents based on code functions and classes, using context-aware splitting. Each document is then encoded and stored in a vector database. Ultimately, the PLM is enhanced with the ability to answer relevant questions by retrieving information from the database.
> > >
> > > During the diverse state generation process, the relevant questions or prompts are used as shown below (similarly, refer to Appendix F.4):
> > >
> > > ```
> > > Currently, you function as a generator of environmental states and agent states. Please generate additional, diverse states based on the structured status information I have provided, following the task manual and code repository. The resulting states must adhere to the following constraints: 1. Compliance with the input format; 2. Ensuring state legality.
> > >
> > > ```
> > >
> > > In addition to the prompt, we provide a small number of environment-provided state representations (non-textual and structured). As the states of both Overcooked and MiniRTS are **represented by structures with a limited number of discrete variables**, we considered implementing diverse state generation through code understanding. Experiments have shown that this approach is indeed feasible and effective.
> > >
> > > According to our experimental statistics, **over 97% of the goals** (generated from states) were covered during this diverse state generation phase. We have added the above discussion to Appendix G.
> > >
> > > ---
> > >
> > > The above inspiring discussion has greatly improved the quality of our paper. Thank you very much. We also enjoy the inspiring and insightful discussions with you!
> > >
> > > -- Respect and Best Wishes from the authors ^_^

---

> ### Comment · Reviewer_DBEC · 2023-11-23
>
> > Specifically, since states are characterized by code rather than paper, we utilize environment code files for diverse state generation. Similar to the state and action translation section, we also employ LangChain's code understanding module.
>
> Thanks for the authors' elaboration. Using task manual and code repository to generate states as an initialization is a really interesting idea. Though this approach may somewhat restrict domains it can be used, I think this offline manner is novel and encouraged.
>
> In general, I appreciate the authors' efforts during the rebuttal process. Please forgive me since I do not spend much time in checking written prompts and other details in the appendix. The authors did supplement related information well, which helps address my concerns. Hopefully the authors can release the code for so many ad hoc things they have implemented. Good luck :)

---

> > ### Author Response · Authors · 2023-11-23
> >
> > In response to your suggestions, we are planning to make the SAMA pipeline and prompts, which are mainly based on LangChain, available in an open-source format ASAP. We are currently in the process of cleaning and refactoring our code due to significant changes in the LangChain API and the introduction of many new convenient features during the review period. We believe that sharing this code will contribute positively to the research community.
> >
> > Once again, we express our sincere gratitude for your valuable input and recognition of our work. We are committed to implementing your feedback and enhancing our research further.

---

### Official Review · Reviewer_T35w · 2023-10-26

**Soundness:** 2 fair
**Presentation:** 2 fair
**Contribution:** 2 fair
**Rating:** 5
**Confidence:** 2

**Summary:**

This paper investigates long-horizon, sparse reward tasks in cooperative multi-agent RL problems. The proposed method is to use the pre-trained large language model and pre-trained language-ground RL agent. The method prompts the pre-trained language model to generate potential goals, decompose the goal into sub-goals, assign sub-goals to each agent in the multi-agent setting, and replan when the pre-trained RL agents fail to achieve the goal.

The experiments are conducted on two challenging benchmarks for MARL, Overcooked and MiniRTS. The proposed method outperforms the SOTA baselines.

**Strengths:**

The problem of long-horizon, sparse reward tasks in multi-agent RL is super challenging and significant.

It is well-motivated to apply the large-language model, to use prior knowledge and common sense for goal generation and task planning.

Empirically, the proposed method outperforms the SOTA.

**Weaknesses:**

In general, the presentation can be improved. Since there are too many components in the pipeline, it will be better to emphasize and explain the most novel and important part in detail, rather than briefly mention each component within space limit.

The proposed method is not fully analyzed. For example, as for the reward design part, how is it accurate to determine task completion? About the pre-trained language-grounded RL agent, how does it perform the training and validation set of states and sub-goals? In the self-reflection phase, how does the task planning evolve?

**Questions:**

Could you please clarify in Figure 5, which component of the proposed method is updated as environment steps increase? If the language-grounded RL agent is trained here, why is it called pre-trained?

---

> ### Author Response · Authors · 2023-11-18
>
> We are highly grateful for your taking the time to thoroughly review our paper and provide numerous constructive comments and suggestions. This has been of great help in enhancing the rigor and completeness of our paper. Below, we have compiled and addressed each of the questions you raised.
>
> `Q1: In general, the presentation can be improved. Since there are too many components in the pipeline, it will be better to emphasize and explain the most novel and vital part in detail rather than briefly mention each component within the space limit.`
>
> Thank you very much for your suggestions! SAMA is a proof-of-concept algorithm incorporating human common sense from pre-trained language models into multi-agent reinforcement learning to achieve low-sample complexity algorithm training. It accomplishes this through four modules: preprocessing, task decomposition, language grounding, and self-reflection. We have demonstrated the feasibility and effectiveness of this pipeline using two typical long-horizon, sparse-reward tasks: Overcooked-AI and MiniRTS. This represents the major innovation and contribution of our paper.
>
> It's important to note that the techniques employed in each part of the pipeline, such as document question answering for task manual generation, code understanding for state/action translation, and mechanisms for language grounding and self-reflection, are not novel in this paper. As a result, we believe that providing a comprehensive description of the entire pipeline in the main text is extremely necessary. Furthermore, our current version of the main text already covers key implementation details for each module, with only the engineering aspects detailed in the appendix for reference.
>
> `Q2: The proposed method is not thoroughly analyzed. For example, how is it accurate to determine task completion for the reward design part? About the pre-trained language-grounded RL agent, how does it perform the training and validation set of states and sub-goals? In the self-reflection phase, how does the task planning evolve?`
>
> Thank you very much for your question! For the first question, in the current version of SAMA, we have not designed additional mechanisms to ensure the correctness of the generated reward functions. From the experimental results, this does not affect SAMA's ability to approach SOTA baselines and achieve a leading sample efficiency. In Appendix H's Figure 17, we have also compared the performance impact of the human-designed reward function with that of the PLM-generated reward function. It can be seen from the figure that both methods show comparable performance levels. We plan to introduce a meta-prompt mechanism [1, 2] to adjust the output of the PLM during the reward function generation stages based on the performance of the final algorithm, thus further improving the performance of the SAMA algorithm. However, as this mechanism may bring significant framework changes to the SAMA method, and the current version of SAMA is mainly designed for proof-of-concept purposes, we plan to explore the meta-prompt mechanism in more depth in our subsequent work.
>
> Regarding the second question, we have not yet considered the division of training and validation sets, meaning that the current version does not consider the generalization capability of language grounding. In light of the potential emergence of unaccounted environmental states during the evaluation process, periodic inclusion of the encountered states and corresponding PLM-generated subgoals to the dataset is undertaken, facilitating the training of the language-grounding policy.
>
> For the third question, we randomly select an output from the self-reflection process of a SAMA, demonstrating the adjusted goal generation and decomposition results. Here is the excerpt for your reference:

---

> ### Author Response · Authors · 2023-11-18
>
> `The current situation is:` there are no onions on the crafting table and shared bar.
>
> `The generated goal from the goal generation and decomposition personnel is:` making onion soup.
>
> `The decomposed two sub-goals from the goal generation and decomposition personnel are:`
> - **one chef takes an onion from the onion storage room and transports it to the shared bar**;
> - **one chef transports an onion from the shared bar to the crafting table**.
>
> `The current status of the game:` there are no onions on the crafting table and shared bar.
>
> `The current status of the two chefs:` Chef A is idle in the left room; Chef B is idle in the right room.
>
> `The result of the two current sub-goals assignment decided by the sub-goal allocation personnel is as follows:`
> - Chef A: one chef takes an onion from the onion storage room and transports it to the shared bar;
> - Chef B: one chef transports an onion from the shared bar to the crafting table.
>
> `The failed decision inspection history:` None.
>
> `To determine whether any errors occurred during the phase of goal generation and decomposition, or during the subgoal allocation stage, let us think step by step:`
>
> Firstly, we will concentrate on the goal generation and decomposition phase.
> Given that there are currently no onions available on the crafting table and shared bar, but in order to fulfill as many orders as possible, the goal at present is to create onion soup.
> Hence, there seems to be no issue with goal generation.
> In the next step, we must subdivide the above goal into two sub-goals.
> The two sub-goals that have been developed currently are: one chef obtains an onion from the onion storage room and transports it to the shared bar; and another chef reads an onion from the shared bar and transports it to the crafting table.
> Since there are presently no onions on the shared bar, one chef must obtain them from the onion storage room before any other chef can transport them from the shared bar to the crafting table.
> Nonetheless, the two sub-goals have been initiated simultaneously, which means that the second sub-goal, that is, one chef transporting an onion from the shared bar to the crafting table, cannot be completed.
> `Based on the above analysis, it appears that the problematic stage is:` **goal generation and decomposition**.
>
> `Previous failed sub-goals:`
> - one chef takes an onion from the onion storage room and transports it to the shared bar;
> - one chef transports an onion from the shared bar to the crafting table.
>
> `For more reasonable goal generation and decomposition, the goals I generate or the sub-goals I decompose should no longer be consistent with previously failed goals or sub-goals.`
>
> `The current situation is:` there are no onions on the crafting table and shared bar.
>
> `To formulate goals, think step by step:`
> To fill as many orders as possible, we must constantly make onion soup and deliver it to the serving counter.
> There are currently no onions on the crafting table, meaning no ready-made onion soup exists.
>
> `So the current goal by operating valid elements is:` making onion soup.
>
> `Further, to decompose the goal into two sub-goals, let the two chefs complete it separately, let us think step by step:`
> since the current goal is to make onion soup, we need to transport three onions from the storage room to the crafting table.
> No onions are currently on the crafting table, so three more are needed.
> Since the onion storage room and the crafting table are in two separate rooms, we need one chef to carry three onions from the onion storage room to the shared bar and another chef to carry three onions from the shared bar to the crafting table.
> Now there are no onions on the shared bar, and each chef can only carry one onion at a time, so the two sub-goals by operating valid elements are:
> - **one chef takes an onion from the onion storage room and transports it to the shared bar**;
> - **one chef goes to the bar and waits**.
>
> We have added the above self-reflection process to the Appendix G Listing 28 in the revised version.
>
> [1] Yang, Chengrun, et al. "Large language models as optimizers." *arXiv preprint arXiv:2309.03409* (2023).
>
> [2] Ma, Yecheng Jason, et al. "Eureka: Human-Level Reward Design via Coding Large Language Models." *arXiv preprint arXiv:2310.12931* (2023).

---

> ### Author Response · Authors · 2023-11-18
>
> `Q3: Could you please clarify in Figure 5 which component of the proposed method is updated as environment steps increase? If the language-grounded RL agent is trained here, why is it called pre-trained?`
>
> We apologize if our unclear explanations led to any misunderstandings. Figure 5 shows the training progress of the language-grounded MARL agent. In other words, Figure 5 demonstrates the "pre-training" progress of the MARL agent. We refer to the language-grounded MARL agent as a pre-trained model to emphasize that the PLM does not participate in the agent's training process, which would result in significant time and financial costs. The language-grounded MARL agent learns the correspondence between subgoals and actions based on a pre-generated set of subgoals from the PLM, continually interacting with its environment. We regularly test the currently trained MARL agent using the PLM, record its performance, and plot the training curve in Figure 5. Additionally, we update the subgoals with unseen ones during testing to promote the MARL agent's training.
>
> To make our paper more rigorous, we have revised some of the descriptions in Section 3.3 of the revised version to minimize the potential for confusion among readers. Thank you very much for your question!

---

> ### Comment · Reviewer_T35w · 2023-11-20
> **Thanks for the detailed response**
>
> Thank authors for the detailed response.  I appreciate the Figure 17 in Appendix H to compare with human-designed rewards. However, I'm still concerned about the excerpt in the self-reflection phase. It looks following a strict template, which requires human prior knowledge to interact with LLM. I'm concerned how could the proposed approach be generalized to other tasks/benchmarks without experts to communicate with LLM. Since it still requires lots of human engineering/prior knowledge, I'm concerned about its advantages over the baselines. I'd like to keep my score.

---

> > ### Author Response · Authors · 2023-11-23
> > **Have the additional experiments and intuitive explanation addressed your concerns?**
> >
> > Thanks again for your time and effort in reviewing our paper! As the discussion period is coming to a close, we would like to know if we have resolved your concerns with the **additional experiments and intuitive explanation**. We remain open to any further feedback and are committed to making additional improvements if needed. If you find that these concerns have been resolved, we would be grateful if you would consider reflecting this in your rating of our paper :)

---

> ### Author Response · Authors · 2023-11-21
> **Additional experiments + intuitive explanation**
>
> Thank you very much for your timely response! We apologize for any confusion caused by the lack of proper formatting in our presentation of the self-reflection process excerpt. We have **reformatted the content (in this [reply](https://openreview.net/forum?id=1OP4crhgkD&noteId=6A3KSPNEID))**, which may help clarify your estimates about the prompt engineering workload. It is worth noting that most of the content consists of **task-agnostic** templates or generated subgoals by the pre-trained language model (PLM), apart from the chain-of-thought (CoT) section, which requires more task-specific prompt engineering.
>
> Regarding the CoT in-context examples, they're essential to achieving good performance in representative CoT methods such as Self-Consistency [1], Tree-of-Thoughts [2], and Graph-of-Thought [3], as well as self-reflection methods like Reflexion [4], Self-Debugging [5], and Self-Refine [6]. While this aspect of prompt engineering may seem inevitable in the short term, we believe it won't obstruct the development of language agents in the long run, thanks to the improvement in large models' zero-shot reasoning abilities.
>
> To further test the generality of the self-reflection component, we conducted an additional quick experiment. Specifically, inspired by Sewon et al. [7], **we tried applying the Overcooked-AI-designed self-reflection in-context example to the MiniRTS task**. A plausible explanation for promising results lies in the evidence presented by Sewon et al., suggesting that in-context examples can improve PLM's classification ability even without providing correct labels for classification tasks. One potential reason is that these examples constrain the input and output spaces, guiding the PLM's behavior. Similarly, in our case, while the Overcooked-AI-designed self-reflection chain of thought cannot directly guide MiniRTS task decomposition, it does **specify the "chain of thought space"** from input (current state, failure flag, etc.) to output (regenerated goals, subgoals, etc.).
>
> As shown in Figure 14 ([screenshot](https://ibb.co/jv0hS4r)), even **non-task-relevant in-context examples exhibit decent performance**, indicating a certain level of generalizability in the self-reflection mechanism, which could help reduce the extent of prompt engineering required.
>
> We appreciate the new insights you've provided and your valuable contribution to the depth of our work! **We have included the above analysis in Appendix H**. Please let us know if you have any further questions.
>
> If you find that these concerns have been addressed, we would be grateful if you would consider reflecting this in your rating of our paper :)
>
> ---
>
> [1] Wang, Xuezhi, et al. "Self-consistency improves chain of thought reasoning in language models." *ICLR* 2023.
>
> [2] Yao, Shunyu, et al. "Tree of thoughts: Deliberate problem solving with large language models." *NeurIPS* 2023.
>
> [3] Besta, Maciej, et al. "Graph of thoughts: Solving elaborate problems with large language models." *arXiv preprint arXiv:2308.09687* (2023).
>
> [4] Shinn, Noah, Beck Labash, and Ashwin Gopinath. "Reflexion: an autonomous agent with dynamic memory and self-reflection." *NeurIPS* 2023.
>
> [5] Chen, Xinyun, et al. "Teaching large language models to self-debug." *arXiv preprint arXiv:2304.05128* (2023).
>
> [6] Madaan, Aman, et al. "Self-refine: Iterative refinement with self-feedback." *arXiv preprint arXiv:2303.17651* (2023).
>
> [7] Min, Sewon, et al. "Rethinking the Role of Demonstrations: What Makes In-Context Learning Work?." *EMNLP* 2022.

---

### Official Review · Reviewer_b4Ao · 2023-10-27

**Soundness:** 3 good
**Presentation:** 3 good
**Contribution:** 2 fair
**Rating:** 3
**Confidence:** 4

**Summary:**

The authors propose Semantically Aligned task decomposition in MARL (SAMA), a method that aims to generate subgoals for MARL tasks with sparse reward signals. By taking advantage of pretrained language models, the proposed method shows to be more sample efficient during MARL training.

**Strengths:**

Generally, the paper is well written and well organized.

This paper:
* proposes a complex method to integrate language models and MARL from a task decomposition perspective
* shows that using prior knowledge from language models improves sample efficiency in MARL
* provides detailed analysis regarding the language learned for the tasks

**Weaknesses:**

* Overall, there is a big limitation from the proposed method: the approach introduced requires the prior existence of the required resources to create an accurate task manual and state action translations. This is indeed pointed by the authors in section 3.1: "Nevertheless, linguistic task manuals still need to be made available for general multi-agent tasks. Furthermore, except for a handful of text-based tasks, typical multi-agent environment interfaces cannot provide text-based representations of environmental or agent state information and action information."; this makes this method very limiting
* It is stated that the language task manual is generated from the latex code of the paper of the multi-agent task. Once again, this sounds very limiting.
* While the proposed method shows to reduce the required samples in the tasks (Fig. 5), it requires very complex prior preprocessing to create the required text-based rules and manuals. This makes me think that it possible that the method becomes even more costly, in general, after all of this processing, despite the sample efficiency in the task.
* From my understanding, the goal decomposition is made prior to the task, meaning that a lot of prior knowledge is required (as mentioned before). Other methods such as MASER [1] or [3] decompose the tasks on a more flexible manner, which saves a lot of potential preprocessing.
* Throughout the paper, the authors claim several times that their method does not need to learn how to split subgoals or to generate them on the go as other methods do, reducing the sample complexity. However, the preprocessing carried needed to achieve this seems very complex and requires a lot of prior knowledge and carefuly engineered features. I wonder again whether this is really more advantageous than following the standard approaches.
* I have concerns regarding the claims that this method addresses the credit assignment problem. Also since the authors test is environments with only two agents, this can be difficult to analyse (overcooked with 2 agents and MINIRTS with 2 agents; in related works environments with many more agents such as SMAC [2] are used).
* In the conclusion it is stated as a limitation: "where human rationality is immaterial or inexpressible in language or when state information is not inherently encoded as a natural language sequence."; Yet i believe this is a very interesting remark and would be interesting to see how a method such SAMA can be used to tackle these problems, instead of having a preset of convenient "manuals".
* The authors mention throughout the paper (introduction and Fig. 2, for example) the potential for generalizability of the proposed method. However, this is not shown or further discussed, and due to the required preprocessing I fail to understand how generalization to different environments/tasks can be easily done.

Overall, I think that the proposed way of integrating language models with MARL from a task decomposition perspective is interesting and can contribute to the explainability of MARL systems. However, I feel that the proposed method in this paper has several limitations and requires a very complex preprocessing. If the manuals cannot be properly generated, then it is not possible to tackle the problems. I also wonder whether the shown sample efficiency is really worth it since it needs all this preprocessing.

[1] https://arxiv.org/abs/2206.10607

[2] https://arxiv.org/abs/1902.04043

[3] https://ieeexplore.ieee.org/document/9119863


Minor:
- in section 2:  "endeavoring to parse it into N distinct sub-goals g1k, · · · , gN"; missing brackets
- in section 3.1: "illustrate the process in the yellow line of Figure 2" the word figure shouldnt be in yellow; same here (purple line in Figure 2) and in the others that follow in the paper

**Questions:**

1. To extend the proposed method to other cases, would it be possible to create a task manual and state action translation for environments that do not follow the conventions presented in this paper? For instance, in more complex environments such as SMAC [2].
2. In section 3.1: "For each paragraph $S^i_{para}$, we filter paragraphs for relevance and retain only those deemed relevant by at least one prompt from $Q_{rel}$."; how is the filtering of the relevant paragraphs done? Are they manually filtered?
3. In overcooked, despite the method being more sample efficient during training (figure 5) we can see in figure 4-right that the testing performance of the proposed method stays below other sota methods. Is this because MARL might not be good enough for this environment?

---

> ### Author Response · Authors · 2023-11-18
>
> We are incredibly grateful for your willingness to take the time to conduct a thorough review of our paper and provide numerous constructive comments and suggestions. This has been immensely helpful in enhancing the rigor and completeness of our paper. Below, we have organized the questions you raised and responded to each of them one by one.
>
> `Q1: Overall, the proposed method has a big limitation: the approach introduced requires the prior existence of the required resources to create an accurate task manual and state action translations. The authors point out this in section 3.1: "Nevertheless, linguistic task manuals still need to be made available for general multi-agent tasks. Furthermore, except for a handful of text-based tasks, typical multi-agent environment interfaces cannot provide text-based representations of environmental or agent state information and action information."; this makes this method very limiting.`
>
> We appreciate the feedback you have provided. However, we disagree with the notion that it is a significant limitation for SAMA to require pre-existing resources, such as papers, code, and so on, for generating task manuals and translating states/actions.
>
> Firstly, for various simulation environments proposed by the MARL community, there are corresponding codes, explanatory papers, technical reports, and in most cases, pre-existing task manuals. The use of papers for generating task manuals in SAMA has already considered relatively boundary situations.
>
> Secondly, the relevant code and documentation are even more standardized for real-world tasks. In addition, although expert systems are no longer a very popular research field, previous academic research has contributed many expert systems to numerous real-world tasks. These expert systems also contain a wealth of task-related documents, knowledge bases, and the like.
>
> In summary, we believe SAMA's assumption of the pre-existence of relevant resources for solving tasks is quite reasonable.
>
> `Q2: It is stated that the language task manual is generated from the latex code of the paper of the multi-agent task. Once again, this sounds very limiting.`
>
> Thank you very much for your feedback. We believe that the most significant contribution of SAMA lies in establishing a technical path to solve multi-agent problems more efficiently using PLM and MARL methods. For the simulation environments currently proposed in the academic community, most of them opt to make their work open-source on the arXiv platform. As a result, acquiring the corresponding LaTeX code is not very difficult. Even if the LaTeX code is unavailable, we can still process these articles, technical reports, and other documents using PDF content extraction tools, OCR tools, or even multimodal pre-trained large models. Integrating support for these tools in SAMA's task manual generation process does not align with the proof-of-concept focus of our paper. We plan further to improve SAMA's capabilities in our subsequent work.
>
> `Q3: While the proposed method reduces the required samples in the tasks (Fig. 5), it requires very complex prior preprocessing to create the required text-based rules and manuals. This makes me think that the method may become even more costly, in general, after all of this processing, despite the sample efficiency in the task.`
>
> Thank you very much for your input. We apologize for any confusion arising from placing some experimental details in the appendix section of our work. The preprocessing stage in generating the task manual is completed through prompt PLM, with key prompts listed in Appendix F.1. As you can see, only a few prompts are necessary. We utilized the open-source LangChain framework [1] throughout the process, minimizing the coding burden. Furthermore, the preprocessing stage relies on retrieval-augmented generation, precisely document question answering, which is already supported by GPT-4 plugins. We believe that as large open-source models continue to develop, similar technologies will become increasingly mature while the barriers to entry and costs significantly decrease.
>
> In summary, we believe the preprocessing technology used in SAMA's task manual generation phase is not overly complex and will become even more convenient as relevant tools develop. As a proof-of-concept, the SAMA method has already demonstrated the feasibility of this preprocessing technology for solving MARL tasks.
>
> [1] https://langchain-langchain.vercel.app/docs/get_started/introduction

---

> ### Author Response · Authors · 2023-11-18
>
> `Q4: From my understanding, the goal decomposition is made before the task, meaning that much prior knowledge is required (as mentioned before). Other methods, such as MASER [1] or [3], decompose the tasks more flexibly, saving a lot of potential preprocessing.`
>
> Thank you very much for sharing your thoughts. While it's true that goal decomposition in SAMA requires substantial prior knowledge, this knowledge is obtained by querying the PLM and not through learning from scratch or requiring intensive prompt engineering. This is precisely what sets SAMA apart from the baselines compared in the paper.
>
> Methods such as MASER attempt to learn complex target generation, decomposition, and completion steps in an end-to-end reinforcement learning paradigm. This leads to high sample complexity and the potential for over-generalization issues. SAMA, conversely, introduces a natural language-based goal space and leverages human priors embedded in the PLM to accomplish goal generation, decomposition, and completion in a disentangled and semantically aligned manner.
>
> By doing so, SAMA significantly reduces the goal space that needs to be explored, thereby improving sample efficiency. Moreover, this approach offers greater transparency and interpretability to the task decomposition process.
>
> `Q5: Throughout the paper, the authors claim several times that their method does not need to learn how to split subgoals or generate them on the go as other methods do, reducing the sample complexity. However, the preprocessing needed to achieve this looks pretty complex and requires much prior knowledge and carefully engineered features. I wonder again whether this is more advantageous than following the standard approaches.`
>
> Thank you very much for your question! As discussed in the responses to previous questions, the overall process of SAMA is not overly complex. Although it includes several stages, such as preprocessing, task decomposition, language-grounded MARL, and self-reflection, the first, second, and fourth stages primarily involve using different prompt engineering techniques to query PLMs. We've primarily relied on existing GPT plugins or third-party frameworks (e.g., LangChain) when designing these prompt engineering techniques.
>
> Moreover, we believe there is a significant difference in the overall design style of language agents compared to current data-driven, RL, or MARL paradigms. That is, shifting the focus from tedious parameter tuning towards prompt engineering. We think the workload introduced by SAMA's prompt engineering isn't notably more significant than that of traditional end-to-end task decomposition in MARL algorithms; it substantially improves sample efficiency. This is a primary advantage of language agents over traditional MARL approaches in multi-agent tasks where human common sense can play a crucial supporting role.
>
> Furthermore, a few studies have attempted to leverage PLMs to automate prompt engineering further [1]. This is one of the reasons why we believe integrating PLMs into the MARL framework is a promising research direction with great potential.
>
> [1] https://github.com/OpenBMB/ProAgent

---

> ### Author Response · Authors · 2023-11-18
>
> `Q6: I have concerns regarding the claims that this method addresses the credit assignment problem. Also, since the authors' test is in environments with only two agents, this can be difficult to analyze (overcooked with two agents and MINIRTS with two agents; in related works environments with many more agents such as SMAC [2] are used).`
>
> Thank you very much for your valuable suggestions, which significantly contribute to the integrity of our work. We have verified a variant of MiniRTS involving three agents in Appendix I.1 of the original manuscript version.
>
> Specifically, $3$ agents emerge in the modified environment, each governing $1/3$ of the units.
> Analogously, we modestly modified the built-in medium-level AI script, enabling each iteration to select $3$ types of army units randomly. Given that the built-in script AI constructs only $3$ army unit types per game, we establish an oracle prompt design strategy following the ground truth of enemy units and the attack graph. When the number of agents is larger, feasible plans will grow exponentially as the task horizon increases, which will be more challenging for the current PLM. As seen from Figure 13, the performance of SAMA does not drop significantly in scenarios with more agents. This shows that the SAMA framework itself has sure scalability.
>
> Additionally, we chose Overcooked-AI and MiniRTS over the more complex task of SMAC for the following principal reasons: Firstly, the tasks in these two environments can benefit more significantly from human common sense and have more straightforward manuals and state/action translations. Secondly, their task decomposition and sub-goal allocation processes can be analyzed more intuitively. For example, cooking involves chopping and plating, while MiniRTS has transparent unit counter relationships. Finally, both environments allow for more accessible training of language-grounded MARL agents, with MiniRTS even having pre-trained agents available. These features make these two environments highly suitable for our goal of designing a proof-of-concept algorithm.
>
> The experimental results indicate that SAMA cannot fully solve the tasks in Overcooked-AI and MiniRTS. Hence, using environments like SMAC directly would not be very reasonable. Moreover, employing more complex environments like SMAC would require more meticulous prompt engineering and higher-cost language-grounded MARL agent training and make the experimental results more complicated to analyze. These points do not align with our intention of quickly designing a proof-of-concept algorithm.
>
> In summary, the supplementary experimental results in Appendix I.1 show a certain degree of scalability for the SAMA pipeline. We believe that by employing more advanced and carefully designed tools or technologies to improve each module, SAMA could attempt to tackle complex tasks like SMAC. However, in this paper, we believe that the most significant contribution of SAMA lies in connecting the use of PLM and MARL methods for solving multi-agent tasks and validating its effectiveness and substantial optimization for sample complexity through a proof-of-concept experiment.

---

> ### Author Response · Authors · 2023-11-18
>
> `Q7: The conclusion is stated as a limitation: "where human rationality is immaterial or inexpressible in language or when state information is not inherently encoded as a natural language sequence."; Yet I believe this is a very interesting remark, and would be interesting to see how a method such as SAMA can tackle these problems instead of having a preset of convenient "manuals.”`
>
> Thank you very much for your recognition of our work. About this remark, we have also conducted a preliminary experiment in Appendix I.1 to provide some initial validation of its reasonableness.
>
> Concretely, we have modified the classic FrozenLake environment in OpenAI Gym and expanded it into a task for two agents, which can be regarded as a discrete version of the Spread task in MPE. In this task, two agents must avoid holes in the ice and reach a target location (Figure 14 Left). The difficulty with this task is that each agent's observation space contains only integers representing its coordinates and those of another agent. Besides, it contains no information and can only be remembered through continuous exploration to remember all possible situations. In this case, human commonsense cannot give optimal planning in advance.
>
> We verify the performance of SAMA and other PLM-based task planners (DEPS [1] and Plan4MC [2]) on the multi-agent version FrozenLake. Figure 14 (Right) shows the average rewards of different algorithms over $20$ game episodes. FrozenLake is a sparse reward task with a $+1$ reward only when the agent reaches the target point. As can be seen from the figure, the PLM-based task planner we selected, including SAMA, cannot solve this type of problem.
>
> After a preliminary discussion and analysis, we believe that there could be two potential approaches to enable language agents, such as SAMA, to handle this type of problem. The first approach is lightweight, in which the language agents can learn through trial and error, similar to how RL agents explore their environment, gather experience, and assist with subsequent decision-making. The feasibility of this approach has been preliminarily verified in a recent work [3]. The second approach is a heavier one involving knowledge editing [4], where task-specific knowledge is injected or modified in a task-oriented manner without altering the PLM's general capabilities. This second method is gradually becoming popular in the NLP field, with many papers published [5].
>
> We have added the above analysis to the Appendix I.1 in the revised version, and we hope the above analysis helps answer your question.
>
> [1] Wang, Zihao, et al. "Describe, explain, plan and select: Interactive planning with large language models enables open-world multi-task agents." *NeurIPS* 2023.
>
> [2] Yuan, Haoqi, et al. "Plan4mc: Skill reinforcement learning and planning for open-world minecraft tasks." *arXiv preprint arXiv:2303.16563* (2023).
>
> [3] "Can Language Agents Approach the Performance of RL? An Empirical Study On OpenAI Gym." submitted to ICLR 2024, under review.
>
> [4] De Cao, Nicola, Wilker Aziz, and Ivan Titov. "Editing Factual Knowledge in Language Models." *EMNLP* 2021.
>
> [5] Cao, Boxi, et al. "The Life Cycle of Knowledge in Big Language Models: A Survey." *arXiv preprint arXiv:2303.07616* (2023).
>
> `Q8: The authors mention throughout the paper (introduction and Fig. 2, for example) the potential for generalizability of the proposed method. However, this is not shown or further discussed, and due to the required preprocessing, I fail to understand how generalization to different environments/tasks can be quickly done.`
>
> We apologize if our paper did not clearly explain the generalization issue, leading to confusion. In the four stages of SAMA, namely preprocessing, task decomposition, language grounding, and self-reflection, both preprocessing and language grounding phases are task-agnostic, including the prompt engineering aspect. For task decomposition and self-reflection stages, it is only required to design a 1-shot in-context example with a total token count not exceeding $1000$, roughly equivalent to $750$ words. We believe this level of prompt engineering workload offers better generalization performance than few-shot learning algorithms on new tasks.
>
> `Q9: In section 2: "endeavoring to parse it into N distinct sub-goals g1k, · · ·, gN"; missing brackets.`
>
> We apologize for our oversight and corrected the typos in the revised version. Thank you very much for your thorough review and attention to detail!
>
> `Q10: In section 3.1: "illustrate the process in the yellow line of Figure 2," the word figure shouldn't be in yellow; same here (purple line in Figure 2) and in the others that follow in the paper.`
>
> We apologize for any inconsistency in our formatting due to the excessive use of colors. In the revised version, we've adjusted them all to the default black. Thank you very much for your valuable suggestion!

---

> ### Author Response · Authors · 2023-11-18
>
> `Q11: To extend the proposed method to other cases, would it be possible to create a task manual and state action translation for environments that do not follow the conventions presented in this paper? For instance, in more complex environments such as SMAC [2].`
>
> Thank you for your inquiry. After some discussion and analysis, we have identified several potential technical approaches to generate task manuals and carry out state/action translations in different ways. These approaches have their respective advantages and disadvantages when compared to the method employed by SAMA.
>
> Firstly, we can have language agents create task manuals incrementally by continually interacting with the environment, receiving feedback, and summarizing their experiences without relying on assumed resources like research papers or technical reports. The feasibility of this approach has already been preliminarily validated in some studies [1]. However, a limitation of this approach is that the generation of task manuals deeply intertwines with the task-solving process, and the language agents' capabilities influence the quality of these manuals in solving specific tasks, such as exploration, planning, reasoning, and induction. This deep coupling may lead to negative feedback loops, causing the quality of the manuals and the solving strategies to converge prematurely to suboptimal, or even trivial, solutions.
>
> Secondly, for state/action translation, we can use pre-trained multimodal large models to process image frames [2] or directly compare differences between adjacent frames [3,4]. This approach has also received preliminary validation through previous work. Nevertheless, it performs poorly on non-image tasks, making it less versatile than SAMA. Additionally, most pre-trained multimodal models use predominantly natural images for training, which limits their generalizability to tasks involving non-natural images, such as those found in gaming or simulation environments.
>
> We have added the above discussion to the Appendix J in the revised version.
>
> [1] Chen, Liting, et al. "Introspective Tips: Large Language Model for In-Context Decision Making." *arXiv preprint arXiv:2305.11598* (2023).
>
> [2] Wang, Zihao, et al. "JARVIS-1: Open-World Multi-task Agents with Memory-Augmented Multimodal Language Models." *arXiv preprint arXiv:2311.05997* (2023).
>
> [3] Du, Yilun, et al. "Learning universal policies via text-guided video generation." *NeurIPS* 2023.
>
> [4] Du, Yilun, et al. "Video Language Planning." *arXiv preprint arXiv:2310.10625* (2023).
>
> `Q12: In section 3.1: "For each paragraph, we filter paragraphs for relevance and retain only those deemed relevant by at least one prompt ."; how is the filtering of the relevant paragraphs done? Are they manually filtered?`
>
> We apologize for any confusion caused. Due to organization issues, we have placed the relevant details in Appendix F.1. Specifically, the filtering stage is also implemented by querying the PLM. Below, you'll find the prompts we used:
>
> 1. Would this paragraph contribute to my success in the game?
> 2. Does this paragraph provide information about the game mechanics or strategies?

---

> ### Author Response · Authors · 2023-11-18
>
> `Q13: In overcooked, despite the method being more sample efficient during training (figure 5), we can see in Figure 4-right that the testing performance of the proposed method stays below other SOTA methods. Is this because MARL might not be good enough for this environment?`
>
> Thank you very much for your question! The performance gap is not due to the limitations of MARL algorithms but rather the limited problem-solving ability of PLMs in specific, vertical-domain tasks.
>
> To be more specific, we selected MARL methods as baselines for the Overcooked-AI task. SP, PBT, FCP, and COLE are population-based MARL methods that can achieve state-of-the-art (SOTA) performance in this task. MASER, LDSA, and ROMA are subgoal-based MARL methods (as categorized in our paper) and perform significantly worse in this task. Our approach may not outperform population-based methods, but it is significantly better than classical subgoal-based MARL methods, demonstrating the advantages of semantic task decomposition.
>
> We conducted additional ablation experiments in Appendix H to understand the reasons behind SAMA's performance gap compared to population-based MARL methods in some scenarios. The experimental results suggest that the only learning stage in the SAMA process, language grounding, has satisfactory training outcomes. We therefore believe that the main reason for the gap between SAMA and SOTA methods lies in the inherent limitations of PLMs in logical reasoning and task planning capabilities. One potential approach to addressing this issue is to fine-tune open-source PLMs through lightweight methods such as LoRA [1, 2].
>
> [1] Carta, Thomas, et al. "Grounding Large Language Models in Interactive Environments with Online Reinforcement Learning." *ICML* 2023.
>
> [2] "Leveraging Large Language Models for Optimised Coordination in Textual Multi-Agent Reinforcement Learning." submitted to ICLR 2024, under review.

---

> ### Author Response · Authors · 2023-11-21
> **Have we addressed your concerns?**
>
> Thanks again for your time and effort in reviewing our paper! As the discussion period is coming to a close, we would like to know if we have resolved your concerns expressed in the original reviews. We remain open to any further feedback and are committed to making additional improvements if needed. If you find that these concerns have been resolved, we would be grateful if you would consider reflecting this in your rating of our paper :)

---

> > ### Comment · Reviewer_b4Ao · 2023-11-22
> > **Thanks for the response**
> >
> > I thank the authors for their detailed rebuttal. After carefully going through the response, I still maintain most of my concerns. I still do not see how SARMA can be widely adopted in MARL. As the authors explain, I understand the method can adopt different strategies for other cases, but I think it would be very challenging to use it in its current state in multiple other MARL problems, out of the box. I still believe that relying on these manuals (and related preprocessing) is limiting. For these reasons, I tend to keep my score.

---

> ### Author Response · Authors · 2023-11-22
> **Following discussion + Best Wishes from the authors**
>
> We regret that we were unable to reach a consensus on the applicability of SAMA in MARL. Nevertheless, we would like to express our sincere gratitude for the valuable comments and suggestions you have provided throughout the review process. They have greatly contributed to the improvement of our paper.
>
> Although you have concerns about the generalizability of SAMA, we still believe that, compared to existing language agents, there are no significant flaws in its generalization capabilities (**as shown in this [reply](https://openreview.net/forum?id=1OP4crhgkD&noteId=cc0SL0hRyR)**). Instead of relying on the data-driven, training-from-scratch paradigm in the RL, we think that language agents tend to lean more towards the realm of **software engineering**, specifically algorithmic architecture design. Thus, we believe that using the RL paradigm to assess generalization capabilities might not be the most appropriate means of evaluation.
>
> Once again, we appreciate your valuable insights and the time you have taken to review our work.
>
> ---Respect and Best Wishes from the authors

---

### Official Review · Reviewer_Um1j · 2023-10-31

**Soundness:** 3 good
**Presentation:** 3 good
**Contribution:** 3 good
**Rating:** 6
**Confidence:** 3

**Summary:**

This paper proposes the Semantically Aligned task decomposition (SAMA) framework, which aims to solve the sparse reward problem in multi-agent reinforcement learning. SAMA prompts pre-trained language models with chain-of-thought that can suggest potential goals, provide suitable goal decomposition and subgoal allocation as well as self-reflection-based replanning. Each agent's subgoal-conditioned policy is trained by the language-grounded RL method. Compared with the traditional automatic subgoal generation method, SAMA can have higher sample efficiency. This paper verifies the performance of SAMA on Overcooked and MiniRTS.

**Strengths:**

1. The paper has a clear structure, introduces the proposed method step by step, and the figures are clear and easy to understand.
2. Experimental details are given in the appendix, which makes it easy to reproduce the experimental results. Relevant prompts are also given in the appendix, making the contribution and experimental results more convincing.
3. The testbed chosen in this paper is very representative and challenging. The tasks in Overcooked and MiniRTS can be decomposed by common sense, and their status is relatively easy to translate into natural language. This allows the paper to better focus on how to decompose tasks and allocate subtasks.
4. It can be seen from the experimental results that SAMA can indeed reach or exceed the performance of existing baselines, and the sample efficiency is indeed significantly higher than other baselines.

**Weaknesses:**

1. Currently, SAMA may not be applicable to an environment where human rationality is immaterial or inexpressible in language or when state information is not inherently encoded as a natural language sequence. For example, when the state space is continuous, it is difficult for SAMA to complete the state and action translation stage.
2. PLM still has some flaws, which sometimes hinder the normal progress of the entire SAMA process. In addition, using PLM will bring additional time costs and economic costs.
3. In some scenarios (such as Coordination Ring), although SAMA learns very quickly in the early stage, the final convergence results are still not as good as some baselines.
4. Although in the task manual generation stage, SAMA automatically extracts critical information from the latex file or code through PLM, this is undoubtedly a relatively cumbersome process, so the cost of this stage cannot be ignored.

**Questions:**

1. Is the introduction of the self-reflection mechanism unfair to other baselines? Because other methods do not have this ability similar to "regret."
2. As can be seen from Figures 4 and 5, although SAMA has a very high sample efficiency in the early stages of training, it often converges to a local optimal solution. What is the reason for this result? Is it because the interval $k$ is too large? Do different $k$ values have different effects on the algorithm's performance?
3. How to ensure the accuracy of PLM in the Task Manual Generation stage or the generated code snippet for assessing sub-goal completion?
4. Is the wall time for the agent to complete a round in SAMA much different from other baselines? What is the number of tokens that need to be input to PLM in one episode?

---

> ### Author Response · Authors · 2023-11-18
>
> We greatly appreciate your taking the time to carefully review our paper and provide valuable suggestions and constructive feedback. This has significantly helped us improve the rigor and completeness of our paper. Additionally, we are grateful for your recognition of our work! Below, we have organized your questions and will address them individually.
>
> `Q1: SAMA may not apply to an environment where human rationality is immaterial or inexpressible in language or when state information is not inherently encoded as a natural language sequence. For example, when the state space is continuous, it is difficult for SAMA to complete the state and action translation stage.`
>
> We fully acknowledge that the method we propose has the limitations you mentioned. Currently, most language agents operate on the idea of using pre-trained large models to replace humans' role in algorithmic frameworks. Therefore, most of these methods inherently share similar limitations, which we have verified through a simple experiment designed in Appendix I.1.
>
> Concretely, we have modified the classic FrozenLake environment in OpenAI Gym and expanded it into a task for two agents, which can be regarded as a discrete version of the Spread task in MPE. In this task, two agents must avoid holes in the ice and reach a target location (Figure 14 Left). The difficulty with this task is that each agent's observation space contains only integers representing its coordinates and those of another agent. Besides, it contains no information and can only be remembered through continuous exploration to remember all possible situations. In this case, human commonsense cannot give optimal planning in advance.
>
> We verify the performance of SAMA and other PLM-based task planners (DEPS [1] and Plan4MC [2]) on the multi-agent version FrozenLake. Figure 14 (Right) shows the average rewards of different algorithms over $20$ game episodes. FrozenLake is a sparse reward task with a $+1$ reward only when the agent reaches the target point. As can be seen from the figure, the PLM-based task planner we selected, including SAMA, cannot solve this type of problem.
>
> However, we believe that employing techniques such as knowledge editing [3] can help alleviate this issue, and we leave this for discussion in future work. Additionally, we think the example of continuous states and actions you raised here is not entirely appropriate. Existing works [4] have directly translated the classic control tasks with continuous state and action spaces in the OpenAI Gym and used language agents to achieve performance approach the PPO algorithm.
>
> [1] Wang, Zihao, et al. "Describe, explain, plan and select: Interactive planning with large language models enables open-world multi-task agents." *NeurIPS* 2023.
>
> [2] Yuan, Haoqi, et al. "Plan4mc: Skill reinforcement learning and planning for open-world minecraft tasks." *arXiv preprint arXiv:2303.16563* (2023).
>
> [3] De Cao, Nicola, Wilker Aziz, and Ivan Titov. "Editing Factual Knowledge in Language Models." *EMNLP* 2021.
>
> [4] "Can Language Agents Approach the Performance of RL? An Empirical Study On OpenAI Gym." submitted to ICLR 2024, under review.
>
> `Q2: PLM still has some flaws, which sometimes hinder the regular progress of the entire SAMA process. In addition, using PLM will bring additional time costs and economic costs.`
>
> We fully acknowledge the limitation you mentioned regarding our proposed method. Since pretrained large models can be considered "generalists" rather than "specialists," their success rate in solving specific tasks in niche domains is inherently limited. Even with the addition of a self-reflection mechanism, there is an upper bound to the improvement it can bring. However, we believe that incorporating large models within the RL or MARL paradigms aligns with the current trends in the research community. As this paper represents an early attempt at integrating large models into MARL, we directly employed pretrained large models without any task-oriented fine-tuning. A few studies have already improved decision-making and inference capabilities for specific tasks in RL [1] or MARL [2] by fine-tuning pretrained large models. Additionally, fine-tuning an open-source "smaller" large model can significantly reduce the time and financial burdens of accessing the OpenAI API.
>
> Furthermore, whether using pretrained large models or fine-tuned open-source large models, we do not believe their use would incur "extra" time costs. The substantial improvement in sample efficiency achieved by introducing large models more than compensates for these time costs, ultimately saving considerable time.
>
> [1] Carta, Thomas, et al. "Grounding Large Language Models in Interactive Environments with Online Reinforcement Learning." *ICML* 2023.
>
> [2] "Leveraging Large Language Models for Optimised Coordination in Textual Multi-Agent Reinforcement Learning." submitted to ICLR 2024, under review.

---

> ### Author Response · Authors · 2023-11-18
>
> `Q3: In some scenarios (such as Coordination Ring), although SAMA learns very quickly in the early stage, the final convergence results are still not as good as some baselines.`
>
> Thank you very much for your careful review of our manuscript. As you mentioned, SAMA does not achieve the performance of some baselines in specific scenarios. However, it is worth emphasizing that the primary purpose of our work is not to achieve SOTA performance in test tasks but rather to focus on validating concepts or ideas. By designing a pipeline consisting of four key modules - preprocessing (including task manual generation, state, and action translation), semantic task decomposition, language-grounded MARL agent training, and self-reflection mechanism, SAMA paves the way for utilizing pre-trained large models combined with MARL to solve specialized tasks in vertical domains.
>
> At the same time, we have also demonstrated through experiments that this pipeline can approach the performance of dedicated MARL methods and show significant advantages in sample efficiency. We believe that the success of this technical route will promote the subsequent integration of language agents and MARL to solve more complex multi-agent cooperation or game problems.
>
> `Q4: Although in the task manual generation stage, SAMA automatically extracts critical information from the latex file or code through PLM, this is undoubtedly a relatively cumbersome process, so the cost of this stage cannot be ignored.`
>
> Thank you very much. Indeed, there are certain limitations due to the OpenAI API usage in SAMA, as you mentioned. However, we believe that as the capabilities of open-source pretrained large models continue to improve and the context length they can handle increases, the time and economic costs at this stage will be significantly reduced. Moreover, it has become standard practice for most existing language agents to access external knowledge bases or vector databases, so we do not consider generating task manuals or translating states and actions through papers or code a significant flaw.
>
> `Q5: Is the introduction of the self-reflection mechanism unfair to other baselines? Because other methods do not have this ability similar to "regret."`
>
> Thank you very much for your question, as it is a very intriguing one. We believe this comparison is still fair because the trial-and-error paradigm used in the training process of reinforcement learning inherently embodies a form of "regret" in a certain sense. After a failed attempt due to trial and error, the agent still has the opportunity to make another attempt. SAMA's self-reflection mechanism [1] is designed based on the inspiration from the trial-and-error paradigm used in reinforcement learning.
>
> [1] Shinn, Noah, et al. "Reflexion: Language agents with verbal reinforcement learning." *NeurIPS* 2023.
>
> `Q6: As seen from Figures 4 and 5, SAMA has a very high sample efficiency in the early stages of training, but it often converges to a local optimal solution. What is the reason for this result? Is it because the interval is too large? Do different values have different effects on the algorithm's performance?`
>
> Thank you very much for your question! We are unsure about the specific meaning of "interval" you mentioned. Figure 5 displays the learning curve of the language-grounded MARL algorithm. The PLM does not participate in the training process of this algorithm but only provides warm-start training data, including goals and subgoals represented by natural language text. The performance on the y-axis of the learning curve is obtained by testing the language-grounded MARL agents combined with the PLM at different training progressions in the environment. Of course, during the testing process, the PLM may generate new goals and subgoals that do not exist in the training data, and we will add them to the training data as well.
>
> The SAMA converges rapidly to a suboptimal result mainly due to the limitations of the PLM rather than issues arising in the training process of the language-grounded MARL algorithm (we have verified this through the ablation study in Appendix H). We hope the above analysis can help answer your questions.

---

> ### Author Response · Authors · 2023-11-18
>
> `Q7: How to ensure the accuracy of PLM in the Task Manual Generation stage or the generated code snippet for assessing sub-goal completion?`
>
> Thank you very much for your question! In the current version of SAMA, we have not designed additional mechanisms to ensure the correctness of the generated task manual and the generated reward function. From the experimental results, this does not affect SAMA's ability to approach SOTA baselines and achieve a leading sample efficiency. In Appendix H's Figure 17, we have also compared the performance impact of the human-designed reward function with that of the PLM-generated reward function. It can be seen from the figure that both methods show comparable performance levels. We plan to introduce a meta-prompt mechanism [1, 2] to adjust the output of the PLM during the task manual and reward function generation stages based on the performance of the final algorithm, thus further improving the performance of the SAMA algorithm. However, as this mechanism may bring significant framework changes to the SAMA method, and the current version of SAMA is mainly designed for proof-of-concept purposes, we plan to explore the meta-prompt mechanism in more depth in our subsequent work.
>
> We have added the above analysis to the Appendix J in the revised version.
>
> [1] Yang, Chengrun, et al. "Large language models as optimizers." *arXiv preprint arXiv:2309.03409* (2023).
>
> [2] Ma, Yecheng Jason, et al. "Eureka: Human-Level Reward Design via Coding Large Language Models." *arXiv preprint arXiv:2310.12931* (2023).
>
> `Q8: Is the wall time for the agent to complete a round in SAMA much different from other baselines? What is the number of tokens that need to be input to PLM in one episode?`
>
> Thank you very much for your question, and we apologize for not providing detailed data in our paper. We have added the following content to Appendix E of the revised version:
>
> During the testing process of SAMA, a query to the PLM is only made to generate the next goal and assign sub-goals after the language-grounded MARL agents have completed their assigned tasks. While the self-reflection mechanism may require some queries to be repeated multiple times, the number of times (averaging around 20 times for Overcooked-AI, one time for MiniRTS) the PLM is queried is relatively tiny compared to the entire episode horizon. According to our rough statistics, the wall time has increased by approximately 1-2 times compared to the baselines.
>
> The number of tokens required for each query to the PLM is similar for both Overcooked-AI and MiniRTS, at around 3,000 tokens. Therefore, for Overcooked-AI, the total number of tokens needed per episode (after convergence) is approximately 20 * 3,000 = 60,000. The cost for GPT-3.5-turbo is 0.06 dollars, while the cost for GPT-4 is 1.8 dollars. The output of the PLM is around 1,500 tokens. Thus, the total output tokens amount is approximately 30,000. Since the cost of output tokens is twice that of input tokens, the cost is consistent. For MiniRTS, the total number of tokens needed per episode is approximately 1 * 3,000 = 3,000. The cost for GPT-3.5-turbo is 0.003 dollars, while the cost for GPT-4 is 0.09 dollars. The output of the PLM is also around 1,500 tokens. Thus, the cost is also consistent with the input.
>
> We have added the above analysis to the Appendix E in the revised version.

---

> ### Author Response · Authors · 2023-11-21
> **Have we addressed your concerns?**
>
> Thanks again for your time and effort in reviewing our paper! As the discussion period is coming to a close, we would like to know if we have resolved your concerns expressed in the original reviews. We remain open to any further feedback and are committed to making additional improvements if needed. If you find that these concerns have been resolved, we would be grateful if you would consider reflecting this in your rating of our paper :)

---

> > ### Comment · Reviewer_Um1j · 2023-11-22
> > **Thanks for the response**
> >
> > Thanks to the authors for their detailed responses. The authors cleared up some misconceptions and gave more experimental details, which resolved some of my concerns. SAMA can indeed perform well in some simple multi-agent tasks. But I still think SAMA may be difficult to be applied to other multi-agent domains, such as StarCraft II or Google Research Football. So I keep my score. Nonetheless, I think the current work is a contribution to the multi-agent reinforcement learning and LLM communities.

---

> > > ### Author Response · Authors · 2023-11-22
> > > **Following discussion + Best Wishes from the authors**
> > >
> > > Thank you very much for your recognition of our work, as well as the numerous suggestions and comments you have provided. We greatly appreciate your input, which has been invaluable in improving our paper. Regarding the difficulties of applying SAMA to domains such as StarCraft II or Google Research Football, we agree that the **main obstacles stem from translating these environments and achieving effective language grounding**.
> > >
> > > As language agents **rely on natural language to serve as a unified interface**, the challenges you have mentioned are indeed unavoidable. In our future research, we plan to continue exploring MARL based on **multimodal pre-trained large models**, hoping to alleviate the inherent limitations of language agents.
> > >
> > > Once again, we deeply appreciate the time and effort you have dedicated to reviewing our work.
> > >
> > > ---Respect and Best Wishes from the authors

---

### Author Response · Authors · 2023-11-18
**Common Response (Revision Uploaded)**

We would like to express our sincere gratitude to all the reviewers for taking the time out of their busy schedules to carefully review our paper and provide a series of constructive comments and suggestions. These comments and suggestions have been enormously helpful in enhancing the rigor and comprehensiveness of our paper. We have now submitted the revised version, in which all additions and modifications are highlighted in $\textcolor{red}{red}$ (except for some typo corrections that are not highlighted). Any numbered references in our individual responses to each reviewer pertain to the revised version, rather than the original version. We look forward to hearing back from the reviewers!

---

### Meta-Review · Area_Chair_Mdji · 2023-12-05

**Metareview:**

**Summary**: This paper proposes the SAMA framework, a LLM-assisted method for generating subgoals for cooperative multi-agent RL problems with sparse reward signals. The idea is to prompt LLMs with chain-of-thought to suggest potential goals, decompose the goal into sub-goals, assign sub-goals to each agent, and replan when the pre-trained RL agents fail. SAMA has better sample efficiency than SOTA baselines on Overcooked and MiniRTS.

**Strengths**:
- SAMA reaches or exceeds baseline performance with a higher sample efficiency.
- The paper is well-written, well-organized, and clear.
- The appendix contains experimental details that make it easy to reproduce the experimental results.

**Weaknesses**:
- Limited generalization: A big limitation several reviewers (including myself) stress is the method's reliance on resources for creating an accurate task manual and state action translations, which severely reduces the applicability of the method to other MARL problems. It appears that SAMA would need to adopt different strategies for every new MARL problem, so it can't be used out of the box.
- Huge amount of preprocessing effort: Related to the above point, SAMA requires complex preprocessing, prior knowledge, and carefully engineered features to learn how to split subgoals or generate them, so despite being more sample efficient than baselines, it is unclear if all this effort is "worth it", especially if it needs to be done again in new MARL problems.
- Very complicated design: Another major concern is that the sophisticated design of SAMA may hinder its general use on other benchmarks. Additionally, presentation could be improved: the authors may want to emphasize and explain the most novel and important parts of SAMA in detail, rather than briefly mention each component within space limit.
- Limited evaluation: Most evaluations only have 2 agents, which is not very reflective or comprehensive of "multi-agent" RL problems.

Overall, this paper is a low borderline paper. The slightly positive reviews still echo some of the above weaknesses, which is why I believe none of the reviewers is a strong proponent of the paper. Due to the over-reliance on preprocessing and engineering efforts, I am inclined to reject this paper.

**Justification For Why Not Higher Score:**

The algorithm works well, but so many things need to go right for that to happen (have the right pre-processing, extract the right rules, etc.). For this reason, I think SAMA is pretty limited and can only be applied narrowly in its current state.

**Justification For Why Not Lower Score:**

N/A

---

### Decision · Program_Chairs · 2024-01-16

Reject